JCB Journal of Cell Biology

# Mechanistic basis for Sgo1-mediated centromere localization and function of the CPC

Maria Alba Abad[1]*, Tanmay Gupta[2]*, Michael A. Hadders[3], Amanda Meppelink[3], J. Pepijn Wopken[3], Elizabeth Blackburn[1], Juan Zou[1], Anjitha Gireesh[1], Lana Buzuk[1], David A. Kelly[1], Toni McHugh[1], Juri Rappsilber[1,4], Susanne M.A. Lens[3], and A. Arockia Jeyaprakash[1]

**Centromere association of the chromosomal passenger complex (CPC; Borealin-Survivin-INCENP-Aurora B) and Sgo1 is crucial for chromosome biorientation, a process essential for error-free chromosome segregation. Phosphorylated histone H3 Thr3 (H3T3ph; directly recognized by Survivin) and histone H2A Thr120 (H2AT120ph; indirectly recognized via Sgo1), together with CPC's intrinsic nucleosome-binding ability, facilitate CPC centromere recruitment. However, the molecular basis for CPC–Sgo1 binding and how their physical interaction influences CPC centromere localization are lacking. Here, using an integrative structure-function approach, we show that the "histone H3-like" Sgo1 N-terminal tail-Survivin BIR domain interaction acts as a hotspot essential for CPC–Sgo1 assembly, while downstream Sgo1 residues and Borealin contribute for high-affinity binding. Disrupting Sgo1–Survivin interaction abolished CPC–Sgo1 assembly and perturbed CPC centromere localization and function. Our findings reveal that Sgo1 and H3T3ph use the same surface on Survivin to bind CPC. Hence, it is likely that these interactions take place in a spatiotemporally restricted manner, providing a rationale for the Sgo1-mediated "kinetochore-proximal" CPC centromere pool.**

## Introduction

Equal and identical segregation of chromosomes to daughter cells during mitosis requires physical attachment of duplicated sister chromatids (via their kinetochores) to microtubules emanating from opposite spindle poles and subsequent alignment of chromosomes at the metaphase plate, a state known as biorientation (Musacchio and Desai, 2017). Chromosome biorientation is achieved and monitored by several processes including sister chromatid cohesion and quality control mechanisms known as error correction and spindle assembly checkpoint (SAC), all controlled by the spatiotemporal regulation of kinases and phosphatases (Funabiki and Wynne, 2013; Gelens et al., 2018; Saurin, 2018).

Cohesin, a ring-shaped protein complex, is a major player that mediates sister chromatid cohesion in S-phase (Haering et al., 2008; Haering et al., 2002). During prophase, the bulk of cohesin is removed from the chromosome arms (Gandhi et al., 2006; Kueng et al., 2006), while centromeric cohesin is maintained until anaphase onset, protected by Shugoshin 1 (Sgo1; Kitajima et al., 2006; Salic et al., 2004). Cdk1 phosphorylation of Sgo1 during mitosis enables the binding of the Sgo1–protein phosphatase 2 (PP2A) complex to cohesin and ensures that the two sister chromatids remain connected until anaphase onset,

when Separase cleaves the remaining centromeric cohesion, allowing the sister chromatids to separate (Kitajima et al., 2006; Liu et al., 2013b; Shintomi and Hirano, 2009; Waizenegger et al., 2000). Sgo1 localization to centromeres is crucial for its role as cohesion protector. Sgo1 has been suggested to first localize to kinetochores via the Bub1-dependent histone H2A phosphorylation at T120 (H2AT120ph) in order to then efficiently load onto centromeres to protect cohesion and prevent premature sister chromatid separation (Broad et al., 2020; Hengeveld et al., 2017; Kawashima et al., 2010; Liu et al., 2013a).

Error correction is a mechanism that destabilizes incorrect kinetochore–microtubule (KT-MT) attachments, such as syntelic (two sister kinetochores attached to microtubules from the same spindle pole) or merotelic (a single kinetochore attached to microtubules emanating from both spindle poles) attachments and stabilizes correct bipolar attachments. The chromosomal passenger complex (CPC), consisting of Aurora B kinase, inner centromere protein (INCENP), Borealin, and Survivin, is one of the key players regulating this process (Carmena et al., 2012). The CPC, via its Aurora B enzymatic core, destabilizes aberrant KT-MT attachments by phosphorylating outer kinetochore substrates such as the Knl1 complex/Mis12 complex/Ndc80

[1]Wellcome Centre for Cell Biology, University of Edinburgh, Edinburgh, UK;   [2]Early Cancer Institute, University of Cambridge Department of Oncology, Hutchison Research Centre, Cambridge Biomedical Campus, Cambridge, UK;   [3]Oncode Institute and Center for Molecular Medicine, University Medical Center Utrecht, Utrecht University, Utrecht, Netherlands;   [4]Bioanalytics, Institute of Biotechnology, Technische Universität Berlin, Berlin, Germany.

*M.A. Abad and T. Gupta contributed equally to this paper.   Correspondence to A. Arockia Jeyaprakash: jeyaprakash.arulanandam@ed.ac.uk.

complex network so that new attachments can be formed (Cheeseman et al., 2006; Cimini et al., 2006; DeLuca et al., 2006; Lampson et al., 2004; Welburn et al., 2010). Sgo1 has also been shown to regulate KT-MT attachments via PP2A-B56 recruitment that balances Aurora B activity at the centromeres (Meppelink et al., 2015). In addition to error correction, the CPC is also involved in the regulation of the SAC, a surveillance mechanism that prevents anaphase onset until all kinetochores are attached to microtubules (Foley and Kapoor, 2013; Musacchio, 2015).

During (pro)metaphase, the CPC predominantly localizes in the centromeric region between the sister kinetochores, and multiple independent studies recently suggested that the evolutionary conserved Haspin and Bub1 kinases can recruit independent pools of the CPC along the interkinetochore axis. Both recruitment pathways appear redundant for KT-MT error correction and can support faithful chromosome segregation (Bekier et al., 2015; Broad et al., 2020; Hadders et al., 2020; Liang et al., 2020). Haspin mediates phosphorylation on histone H3 Thr3 (H3T3ph), which is recognized by the BIR domain (baculovirus inhibitor of apoptosis repeat domain) of Survivin (Dai et al., 2005; Du et al., 2012; Jeyaprakash et al., 2011; Kelly et al., 2010; Niedzialkowska et al., 2012; Serena et al., 2020; Wang et al., 2010; Yamagishi et al., 2010). Bub1 phosphorylates Thr120 of Histone H2A (H2AT120ph) that is recognized by Sgo1, which in turn is suggested to interact with Borealin via its coiled-coil domain (Bonner et al., 2020; Kawashima et al., 2007; Kawashima et al., 2010; Liu et al., 2015; Tsukahara et al., 2010; Yamagishi et al., 2010). However, our earlier work showed that the histone H3-like Sgo1 N-terminal tail can also interact with the Survivin BIR domain using a binding mode that is nearly identical to that of the histone H3 tail phosphorylated at Thr3 (Jeyaprakash et al., 2011). This suggests that a direct interaction between Survivin and Sgo1 is possible. H3T3ph and H2AT120ph appear to localize to distinct regions within the mitotic centromeres, with H3T3ph localizing to the inner centromere and H2AT120ph to the KT-proximal centromere (Broad et al., 2020; Hadders et al., 2020; Liang et al., 2020; Liu et al., 2013a; Yamagishi et al., 2010). While Sgo1 is known to play a role in the recruitment of the CPC to centromeres, the structural and molecular basis for how the CPC and Sgo1 interact and how these interactions contribute to the localization and function of the specific CPC pools remain unclear. Here, we address these questions by combining biochemical, structural, biophysical, and cellular approaches.

## Results

### CPC–Sgo1 forms a robust complex in vitro

The CPC–Sgo1 interaction has been reported to be critical for sister chromatid biorientation and accurate chromosome segregation from yeast to humans (Hengeveld et al., 2017; Hindriksen et al., 2017; Peplowska et al., 2014; Tsukahara et al., 2010). However, detailed characterization of how the various CPC subunits contribute to Sgo1 binding has not yet been performed. To assess whether the CPC can directly interact with Sgo1 in vitro, we purified recombinant CPC containing

INCENP$_{1-58}$, full-length Survivin, and a stable version of Borealin lacking the first nine residues, Borealin$_{10-280}$ (CPC$_{ISB10-280}$; Fig. 1 A) and tested its interaction with recombinant Sgo1$_{1-415}$ (just lacking the HP1 binding domain and the Sgo motif) using size exclusion chromatography (SEC; Figs. 1 A and S1 A). Our data showed that Sgo1$_{1-415}$ and CPC$_{ISB10-280}$ can form a stable monodisperse complex in vitro as analyzed by SEC (Fig. 1 B). Using isothermal titration calorimetry (ITC), we assessed the binding affinity of this interaction. CPC$_{ISB10-280}$ and Sgo1$_{1-415}$ exhibited high affinity with a dissociation constant ($K_D$) in the low nanomolar range ($K_D$ = 52.83 ± 6.95 nM; Figs. 1 C and S3 D). The interaction is both enthalpically ($\Delta H$ = −6.58 ± 0.098 kcal/mol) and entropically ($-T\Delta S$ = −3.19 kcal/mol) driven. ITC data revealed a 1:1 stoichiometry for the CPC–Sgo1 complex. In agreement with this, the mass photometry data (Fig. S1, B–D) showed a major CPC$_{ISB10-280}$/Sgo1$_{1-415}$ complex species with a measured molecular weight (MW) of 193 ± 29 kD (Fig. S1 D). This is similar to the calculated MW for a 2:2 Sgo1$_{1-415}$:CPC$_{ISB10-280}$ complex (203.6 kD) and suggests that a CPC$_{ISB10-280}$ dimer (105 ± 17.5 kD; Fig. S1 B; calculated MW for a CPC$_{ISB10-280}$ dimer is 108.8 kD) binds to a Sgo1$_{1-415}$ dimer (82 ± 24 kD; Fig. S1 C; calculated MW for a Sgo1$_{1-415}$ dimer is 94.8 kD).

### Sgo1 makes multipartite interactions with CPC subunits

Previous studies have suggested that the Sgo1–CPC interaction is mediated via the N-terminal coiled-coil of Sgo1 and Borealin (Bonner et al., 2020; Tsukahara et al., 2010). However, our structural data revealed that the very N-terminus of Sgo1 can interact with the BIR domain of Survivin (Jeyaprakash et al., 2011). Together, these studies suggest that multipartite interactions between Sgo1 and different CPC subunits could facilitate CPC–Sgo1 complex formation. To gain further structural insights, we performed chemical cross-linking of the CPC$_{ISB10-280}$-Sgo1$_{1-415}$ complex using a zero-length cross-linker, 1-ethyl-3-(3-dimethylaminopropyl) carbodiimide (EDC), followed by mass spectrometry analysis (Fig. S1 E). Cross-linking-mass spectrometry (CLMS) data showed that (1) consistent with our previous observations (Jeyaprakash et al., 2011), the N-terminal region of Sgo1 (amino acids 1–34) makes extensive contacts with Survivin BIR domain (amino acids 18–89); (2) the N-terminal coiled-coil of Sgo1 (amino acids 10–120) interacts with the CPC triple helical bundle; (3) consistent with previous findings (Bonner et al., 2020), the N-terminal coiled-coil also contacts the Borealin dimerization domain; and (4) the Sgo1 region beyond the N-terminal coiled-coil region, which is predicted to be unstructured, contacts both Survivin and Borealin, with most contacts confined to the Sgo1 central region spanning amino acids 180–300 (Fig. 2, A and B). Thus, our cross-linking results suggest that Sgo1 interacts with the CPC mainly via two regions, the N-terminal coiled-coil domain and the unstructured central region (Fig. 2, A and B).

We further analyzed the contribution of different Sgo1 regions for CPC binding using a LacO-LacI tethering assay. For this, we made use of U-2 osteosarcoma (OS) cells harboring a LacO array on the short arm of chromosome 1, to which we could recruit Sgo1 fragments as LacI-GFP fusions (U-2 OS-LacO cells; Janicki et al., 2004). To exclude any contribution from H3T3ph on CPC recruitment, we made use of a Haspin CRISPR mutant

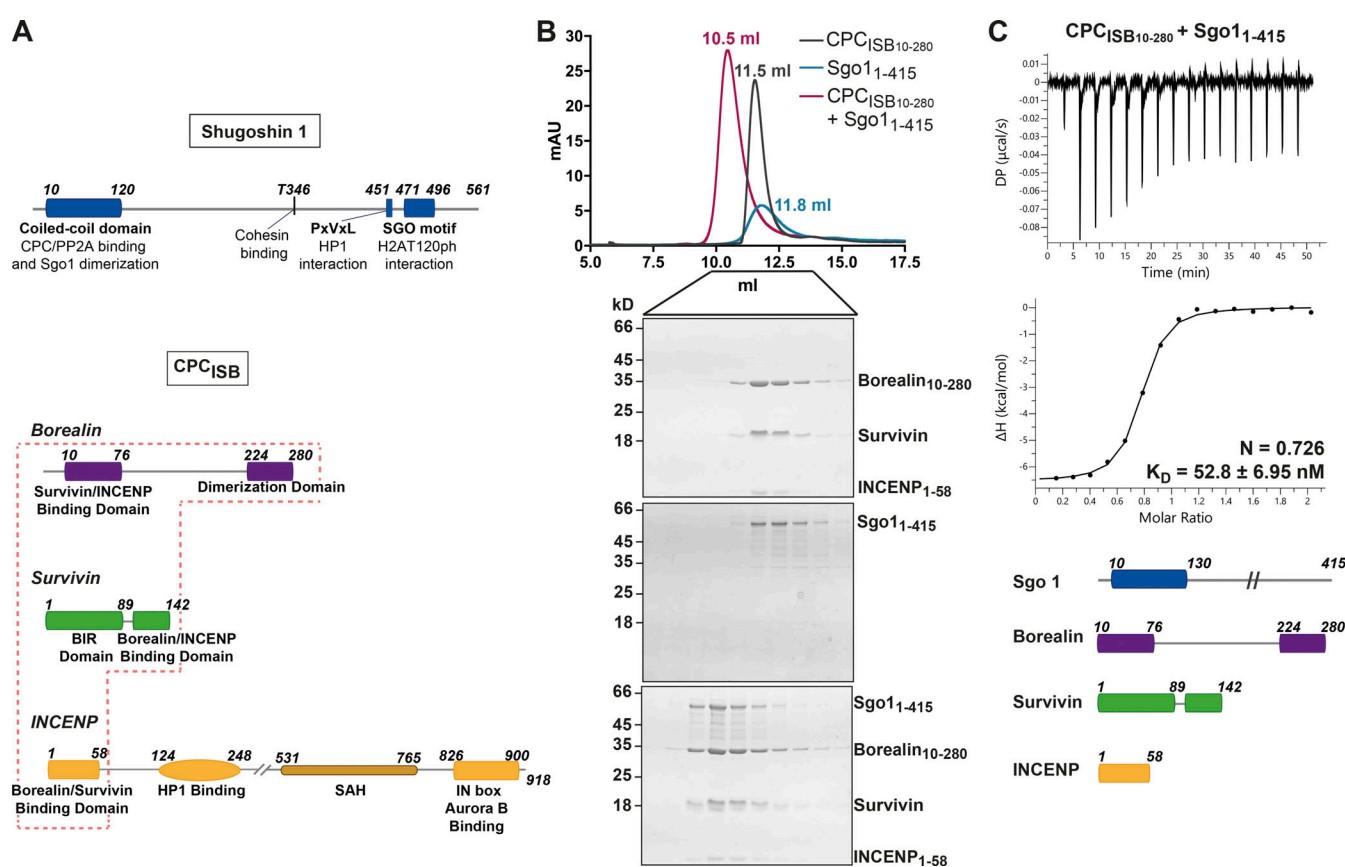

Figure 1. **CPC–Sgo1 forms a robust complex in vitro. (A)** Schematic diagram depicting the domain architecture of Sgo1 and the Borealin, Survivin, and INCENP subunits of the CPC. CPC$_{ISB}$ (INCENP$_{1–58}$, Survivin full length, and Borealin full length) is highlighted in the red box. **(B)** SEC profiles and corresponding Coomassie-stained SDS-PAGEs for the analysis of Sgo1$_{1–415}$ (blue) and CPC$_{ISB10–280}$ (INCENP$_{1–58}$, Survivin, and Borealin$_{10–280}$; dark gray) complex formation (red). A Superdex S200 10/300 GL (Cytiva) column pre-equilibrated with 25 mM Hepes, pH 8, 250 mM NaCl, 5% glycerol, and 2 mM DTT was used. Elution volumes of peak fractions are indicated above the chromatogram peaks. **(C)** Isotherms for Sgo1$_{1–415}$ interaction with CPC$_{ISB10–280}$ (40 µl of 50 µM CPC$_{ISB10–280}$ was injected into 200 µl of 5 µM Sgo1$_{1–415}$). The ITC experiment was performed with 16 × 2.5-µl injections (0.5 µl first injection), 180 s apart, at 20°C. Top panel shows the raw ITC data; bottom panel shows the integrated heat data corrected for heat of dilution and fits to a standard 1:1 binding model (Malvern Instruments MicroCal Origin software, v1.3). DP, differential power; mAU, milli absorbance units. Source data are available for this figure: SourceData F1.

(CM) cell line that displays no discernible Haspin activity (Hadders et al., 2020). Constructs containing Sgo1$_{1–130}$ and full-length Sgo1$_{1–527}$ recruited endogenous Aurora B (Fig. 2 C) and Borealin (Fig. 2 D) to the LacO foci, at comparable levels. This is in line with previous data that suggested the Sgo1 N-terminal region as a major CPC binding site (Bonner et al., 2020; Jeyaprakash et al., 2011; Tsukahara et al., 2010). Surprisingly, Sgo1$_{130–280}$ fused to LacI-GFP was also able to recruit endogenous Aurora B (Fig. 2 C) and Borealin (Fig. 2 D) to the LacO foci, although at lower levels compared with Sgo1$_{1–130}$ and Sgo1$_{1–527}$. In contrast, the Sgo1 fragments Sgo1$_{274–415}$ and Sgo1$_{415–527}$ (Sgo1$_{274–415}$-LacI-GFP and Sgo1$_{415–527}$-LacI-GFP) failed to recruit either Aurora B or Borealin (Fig. 2, C and D). Taken together, the LacO-LacI tethering data confirm that the main CPC-interacting regions of Sgo1 lie within the N-terminal coiled-coil region of Sgo1 (Sgo1$_{1–130}$) and the adjacent unstructured region (Sgo1$_{130–280}$).

**The Survivin interaction with the Sgo1 N-terminal tail is essential for CPC–Sgo1 assembly**
Our previous study identified a histone H3-like N-terminal tail in Sgo1 (Ala1-Lys2-Glu3-Arg4) that interacted with the Survivin

BIR domain with affinity similar to that of the histone H3 tail (Jeyaprakash et al., 2011). Further crystal structure analysis revealed that the mode of Sgo1 tail binding is near identical to that of the histone H3 tail with phosphorylated threonine 3 (Ala1-Arg2-Thr3ph-Lys4; Jeyaprakash et al., 2011; Fig. S1 F). However, whether the Sgo1 N-terminal tail interaction with Survivin is possible in the context of a longer Sgo1 fragment remained an open question. Here, using SEC, we confirmed that Sgo1$_{1–415}$ and Sgo1$_{1–155}$ (a shorter and a more stable fragment spanning aa residues 1–155, identified from limited proteolysis of Sgo1$_{1–415}$) can form a stable complex with Survivin (Figs. 3 A, S1 G, and S2 A and consistent with the data in Fig. 2) and Survivin$_{1–116}$ (mainly composed of the BIR domain, missing most of the C-terminal α helix; Figs. S1 I and S2 B), indicating that the Sgo1 N-terminal tail is accessible for binding Survivin BIR domain in the context of Sgo1$_{1–415}$ and Sgo1$_{1–155}$. Survivin full length and Survivin$_{1–116}$ bound Sgo1$_{1–155}$ with similar binding affinity (240 ± 46.9 nM for Survivin full length vs. 408 ± 110 nM for Survivin$_{1–116}$; Fig. S1, G and I).

Binding of the histone H3 tail by Survivin requires anchoring of the small hydrophobic side chain of H3-Ala1 in a hydrophobic pocket of the Survivin BIR domain (Du et al., 2012; Jeyaprakash

Figure 2. **Sgo1 makes multipartite interactions with CPC components. (A)** Circular view of the EDC cross-links observed between the different subunits of the CPC$_{ISB10–280}$ (INCENP$_{1–58}$ in yellow, Survivin in green, and Borealin$_{10–280}$ in purple) and Sgo1$_{1–415}$ (dark blue). For clarity, only contacts between Sgo1 and

the CPC subunits are shown. Intermolecular contacts of INCENP, Survivin, and Borealin with Sgo1 are shown as yellow, green, and purple lines, respectively. XiNet (Kolbowski et al., 2018) was used for data visualization. Autovalidation filter was used. **(B)** Cartoon representation of the crystal/nuclear magnetic resonance structures of the CPC (CPC core; PDB accession no. 2QFA; Jeyaprakash et al., 2007; Borealin dimerization domain; PDB accession no. 2KDD; Bourhis et al., 2009) and domain architecture of Sgo1 highlighting the regions involved in the CPC–Sgo1 contacts observed in A. Borealin residues in the circular view are annotated to match data deposited to the ProteomeXchange Consortium via the PRIDE repository (1–271 is equivalent to 10–280). **(C and D)** Representative immunofluorescence images (top) and quantification (bottom) for the analysis of the recruitment of endogenous Aurora B (C) and Borealin (D) to the LacO array in U-2 OS-LacO Haspin CM cells expressing different Sgo1-LacI-GFP constructs: LacI-GFP ($n$ = 22 for Aurora B; $n$ = 22 for Borealin), Sgo1$_{1-527}$-LacI-GFP ($n$ = 22 for Aurora B; $n$ = 22 for Borealin), Sgo1$_{1-130}$-LacI-GFP ($n$ = 22 for Aurora B; $n$ = 22 for Borealin), Sgo1$_{130-280}$-LacI-GFP ($n$ = 22 for Aurora B; $n$ = 22 for Borealin), Sgo1$_{274-415}$-LacI-GFP ($n$ = 22 for Aurora B; $n$ = 22 for Borealin), and Sgo1$_{415-527}$-LacI-GFP ($n$ = 22 for Aurora B; $n$ = 22 for Borealin). Representative immunofluorescence images in C and D show Aurora B and Borealin signal for the same cell, thus, DAPI and GFP in C and D are the same. The graphs show the intensities of Aurora B and Borealin over GFP (dots) and the means (red bar). Data are representative of two biological replicates. Scale bar, 5 µm. One-way ANOVA with Dunnett's multiple comparison test (****, P < 0.0001).

et al., 2011; Niedzialkowska et al., 2012; Serena et al., 2020). Likewise, anchoring of Sgo1-Ala1 side chain in the same BIR domain hydrophobic pocket is required for Survivin–Sgo1 N-terminal peptide interaction (Jeyaprakash et al., 2011). Hence, we targeted this interaction to investigate the contribution of the Sgo1 N-terminal tail for Survivin binding by mutating the first alanine after the initiator methionine to a methionine (a residue with a long side chain not compatible with the BIR domain hydrophobic pocket; Sgo1$_{Nmut}$). Remarkably, the Sgo1$_{1-155\ Nmut}$ was unable to interact with Survivin or Survivin$_{1-116}$, indicating that the Sgo1 N-terminal tail interaction with BIR domain is crucial for Survivin binding (Fig. 3 B and Fig. S1, F–J). Similarly, a Survivin BIR mutant (Survivin 3A: K62/E65/H80A) not capable of interacting with the histone H3 tail (Niedzialkowska et al., 2012; Fig. S1 F) failed to interact with Sgo1$_{1-155}$ (Fig. 3 C). Together, these data show that the Sgo1 N-terminus and histone H3 N-terminal tail use the same binding pocket in the Survivin BIR domain.

To assess the contribution of Survivin to the CPC–Sgo1 interaction, we next analyzed the binding between Sgo1$_{1-155}$ or Sgo1$_{1-415}$ and CPC containing different Survivin BIR mutants (K62A, H80A, or K62A/E65A/H80A; Fig. 3, G–J via ITC; Fig. S2 C via SEC; Fig. S3 A via LacO-LacI tethering). Interestingly, we found that while Survivin K62A and Survivin H80A retained Sgo1 binding, Survivin K62A/E65A/H80A abolished Sgo1 binding (binding affinities of 71.2 ± 23.8, 42.4 ± 10.2, and 112 ± 16.4 nM for CPC$_{ISB10-280}$, CPC$_{ISB10-280\ K62A}$, and CPC$_{ISB10-280\ H80A}$, respectively, and no measurable binding affinity for CPC$_{ISB10-280\ 3A}$). Conversely, when we mixed CPC$_{ISB10-280}$ with Sgo1$_{1-155\ Nmut}$ or Sgo1$_{1-415\ Nmut}$ and tested their interaction by SEC (Fig. 3, D–F) and ITC (Figs. S2, D and E; and S3 D), neither Sgo1$_{1-155\ Nmut}$, nor the longer Sgo1$_{1-415\ Nmut}$, which includes the second CPC interacting region (aa 130–280), were able to interact with the CPC. These data agree with the tethering assays in which Sgo1$_{1-130\ Nmut}$-LacI-GFP showed a drastic reduction in its ability to recruit Aurora B (Fig. S3 B) and Borealin (Fig. S3 C) to the LacO array compared with Sgo1$_{1-130}$-LacI-GFP. Together, our results reveal that the Survivin–Sgo1 interaction is essential for CPC binding to Sgo1 and that the Sgo1 N-terminal tail acts as a hotspot whose perturbation abolishes the ability of CPC to form a complex with Sgo1.

## Borealin and INCENP are required for a high-affinity CPC–Sgo1 interaction

To assess how the different CPC subunits contribute to the high-affinity Sgo1 interaction, we performed a series of ITC experiments with either Survivin on its own or CPC$_{ISB}$ containing different Borealin truncations. Sgo1$_{1-130}$ interacted with Survivin with mid-nanomolar affinity ($K_D$ of 255 ± 33 nM; Figs. 4 A and S3 D). This, together with our previous observation that a Sgo1 N-terminal tail peptide bound Survivin with ~1 µM affinity (Jeyaprakash et al., 2011) suggests that, although the interaction between the alanine and the Survivin BIR domain is essential for Sgo1/Survivin complex formation, the Sgo1–Survivin interaction extends beyond Sgo1 N-terminal tail. Sgo1$_{1-130}$ bound CPC$_{ISB10-280}$ with a $K_D$ of 57.4 ± 7.9 nM, an approximately five-fold higher affinity compared with the affinity for Survivin alone (Figs. 4 B and S3 D). This observation together with the CLMS analysis suggests that further interactions involving Borealin, and possibly INCENP, strengthen the affinity between the CPC and Sgo1. Consistent with our CLMS analysis (Fig. 2, A and B) and a previous study (Bonner et al., 2020), CPC$_{ISB}$ lacking the Borealin dimerization domain (CPC$_{ISB10-221}$) bound Sgo1$_{1-130}$ with a threefold lower affinity compared with the CPC$_{ISB10-280}$ ($K_D$ = 163 ± 15.9 nM vs. 57.4 ± 7.9 nM), highlighting the contribution of the Borealin dimerization domain for binding to Sgo1 (Figs. 4 C and S3 D). The measured affinity of CPC$_{ISB10-280}$ binding to the near-full-length Sgo1 (Sgo1$_{1-415}$, $K_D$ = 52.8 ± 6.95 nM; Fig. 1 C) is almost identical to that for Sgo1$_{1-130}$ (Fig. 4 B; $K_D$ = 57.4 ± 7.9 nM). This confirms that the first 130 amino acids of Sgo1 represent the main CPC-interacting region in vitro. Furthermore, the observation that the affinity goes from a micromolar range for the Sgo1 N-terminal tail (AKER peptide) with Survivin (Jeyaprakash et al., 2011) to the low nanomolar range for the CPC$_{ISB10-280}$-Sgo1$_{1-130}$ complex indicates that although the interaction between the CPC and Sgo1 depends on the Sgo1 N-terminal tail binding to Survivin, the high-affinity interaction requires Sgo1 binding to Borealin and possibly INCENP. Overall, the ITC data indicate that the interaction between the Sgo1 N-terminal tail and Survivin is electrostatically driven (Fig. S3, D and E), while the high-affinity interaction between the rest of the Sgo1 regions and the CPC is strengthened by entropic contributions that could be due to a release of water molecules associated with the surface and/or a conformational rearrangement upon binding (Fig. S3 D). These data together suggest that a weak micromolar affinity electrostatic interaction between Survivin and the Sgo1 N-terminal tail is required to establish a high-affinity CPC–Sgo1 interaction mediated by multiple interprotein contacts and hydrophobic effects.

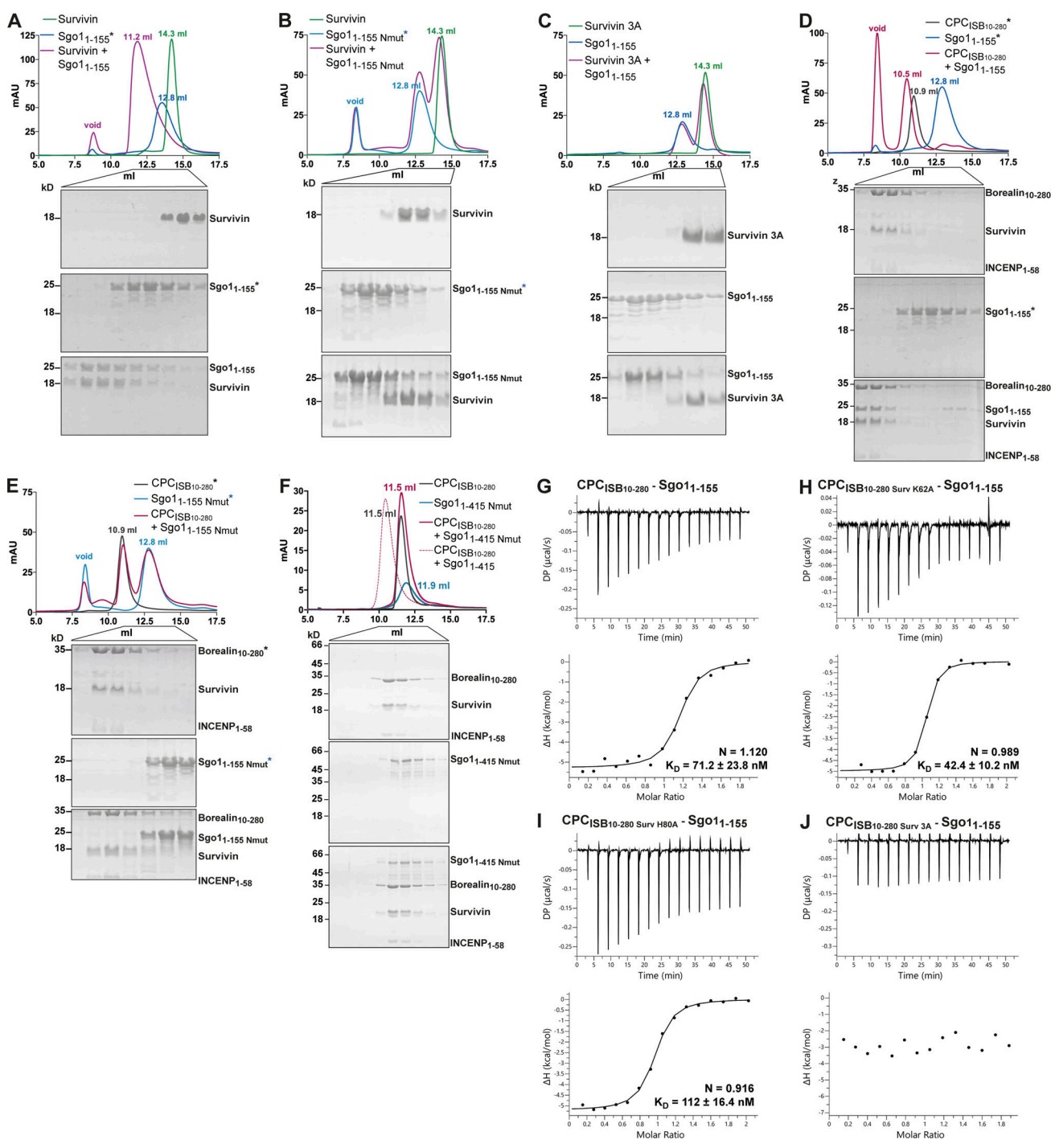

Figure 3. **Survivin interaction with Sgo1 N-terminal tail is essential for CPC–Sgo1 assembly. (A–F)** SEC profiles (top) and corresponding representative SDS-PAGEs stained with Coomassie (bottom) for the analysis of Survivin and Sgo1$_{1-155}$ interaction (A); Survivin and Sgo1$_{1-155}$ Nmut interaction (B); Survivin 3A and Sgo1$_{1-155}$ interaction (C); CPC$_{ISB10-280}$ and Sgo1$_{1-155}$ interaction (D); CPC$_{ISB10-280}$ and Sgo1$_{1-155}$ Nmut interaction (E); and CPC$_{ISB10-280}$ and Sgo1$_{1-415}$ Nmut interaction (F). A Superdex S200 10/300 GL (Cytiva) column pre-equilibrated with either 25 mM Hepes, pH 7.5, 150 mM NaCl, 5% glycerol, and 4 mM DTT (A–E) or 25 mM Hepes, pH 8, 250 mM NaCl, 5% glycerol, and 2 mM DTT (F) was used. Elution volumes are indicated on top of the chromatogram peaks. For easy direct comparison, control SDS-PAGEs and chromatograms corresponding to Sgo1$_{1-155}$, Sgo1$_{1-155}$ Nmut, and CPC$_{ISB 10-280}$ (marked with an asterisk) are shown in two different panels in A and D, B and E, and D and E, respectively. **(G–J)** Isotherms for the analyses of Sgo1$_{1-155}$ interaction with CPC$_{ISB10-280}$ (G); CPC$_{ISB10-280}$ $_{K62A}$ (H); CPC$_{ISB10-280}$ $_{H80A}$ (I); and CPC$_{ISB10-280}$ $_{3A}$ (J). The ITC experiments were performed with 16 × 2.5-μl injections of 100 μM CPC$_{ISB10-280}$ variants into 200 μl of 10 μM Sgo1$_{1-155}$ (0.5 μl first injection), 180 s apart, at 20°C. Top panels show raw ITC data; bottom panels show integrated heat data corrected for heat of dilution and fitted to a standard 1:1 binding model (Malvern Instruments MicroCal Origin software, v1.3). DP, differential power; mAU, milli absorbance units. Source data are available for this figure: SourceData F3.

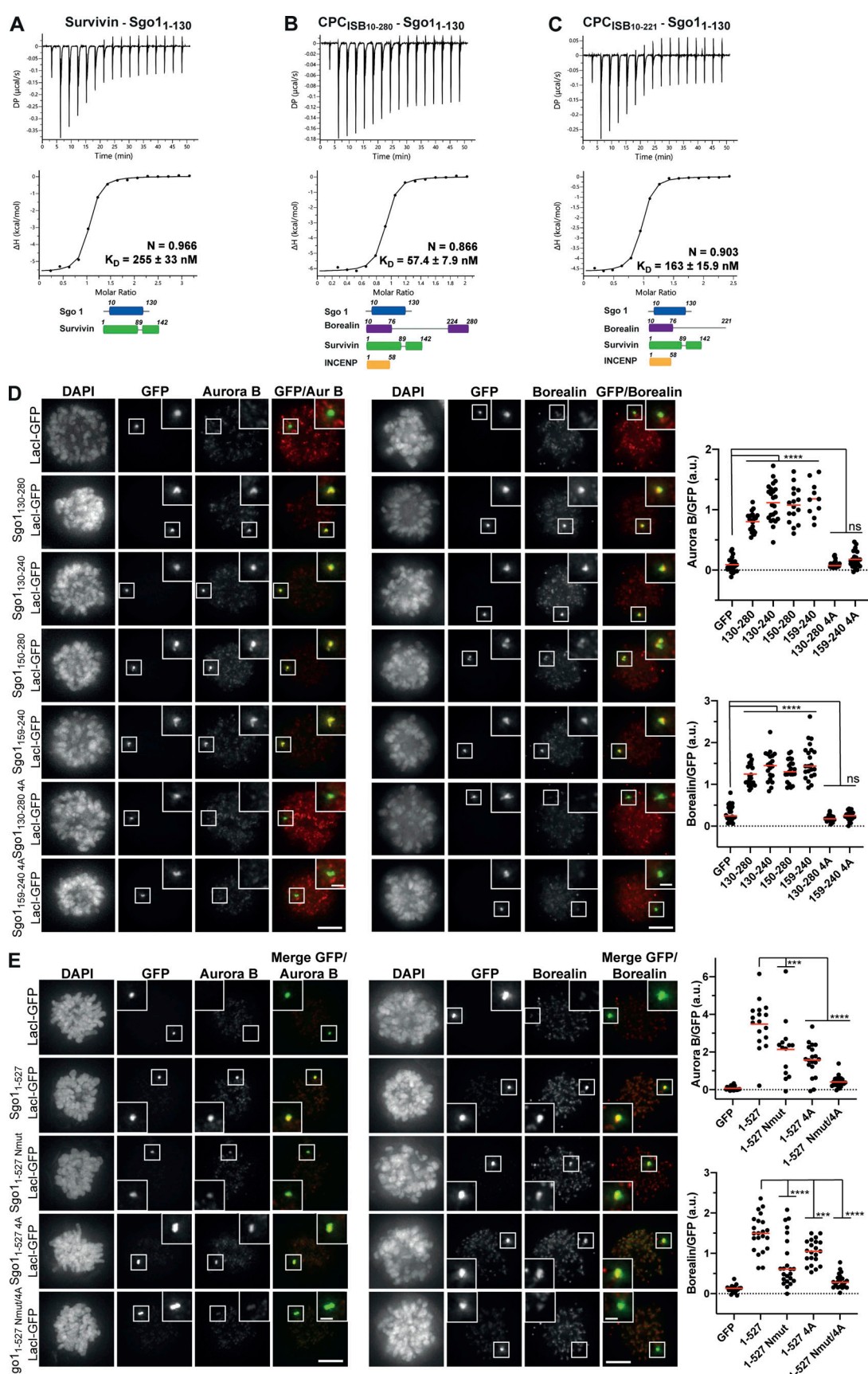

Figure 4. **Multivalent interactions between Sgo1 and different CPC components are essential for high-affinity CPC–Sgo1 binding and efficient CPC recruitment. (A–C)** Isotherms for the analyses of Survivin interaction with Sgo1$_{1–130}$ (40 µl of 312 µM Survivin was injected into 200 µl of 20 µM Sgo1$_{1–130}$; A),

CPC$_{ISB10-280}$ interaction with Sgo1$_{1-130}$ (40 µl of 100 µM CPC$_{ISB10-280}$ was injected into 200 µl of 10 µM Sgo1$_{1-130}$; B), and CPC$_{ISB10-221}$ interaction with Sgo1$_{1-130}$ (40 µl of 120 µM CPC$_{ISB10-221}$ was injected into 200 µl of 12 µM Sgo1$_{1-130}$; C). The ITC experiments were performed with 16 × 2.5-µl injections (0.5 µl first injection), 180 s apart, at 20°C. Top panels show raw ITC data; bottom panels show integrated heat data corrected for heat of dilution and fitted to a standard 1:1 binding model (Malvern Instruments MicroCal Origin software, v1.3). **(D and E)** Representative micrographs (left) and quantifications (right) for the analysis of endogenous Aurora B and Borealin recruitment to the LacO array in U-2 OS-LacO Haspin CM cells expressing different Sgo1-LacI-GFP constructs: LacI-GFP (n = 22 for Aurora B; n = 24 for Borealin), Sgo1$_{130-280}$-LacI-GFP (n = 20 for Aurora B; n = 22 for Borealin), Sgo1$_{130-240}$-LacI-GFP (n = 24 for Aurora B; n = 20 for Borealin), Sgo1$_{150-280}$-LacI-GFP (n = 16 for Aurora B; n = 21 for Borealin), Sgo1$_{159-240}$-LacI-GFP (n = 10 for Aurora B; n = 22 for Borealin), Sgo1$_{130-280\ 4A}$-LacI-GFP (n = 22 for Aurora B; n = 21 for Borealin) or Sgo1$_{159-240\ 4A}$-LacI-GFP (n = 23 for Aurora B; n = 21 for Borealin; D); LacI-GFP (n = 20 for Aurora B; n = 20 for Borealin), Sgo1$_{1-527}$-LacI-GFP (n = 17 for Aurora B; n = 22 for Borealin), Sgo1$_{1-527\ Nmut}$-LacI-GFP (n = 14 for Aurora B; n = 23 for Borealin), and Sgo1$_{1-527\ 4A}$-LacI-GFP (n = 19 for Aurora B; n = 21 for Borealin) or Sgo1$_{1-527\ Nmut/4A}$-LacI-GFP (n = 22 for Aurora B; n = 22 for Borealin; E). The graphs show the intensities of Aurora B and Borealin over GFP (dots) and the means (red bar). Data are representative of four biological replicates. Scale bar, 5 µm. One-way ANOVA with Dunnett's multiple comparison test (****, P < 0.0001; ***, P < 0.001). DP, differential power.

## Interactions involving the Sgo1 N-terminal tail and a hydrophobic stretch spanning residues 188–191 are required for efficient recruitment of the CPC

Our cross-linking and tethering data (Fig. 2) identified an additional novel CPC-interacting region of Sgo1 within aa 130–280, which is found upstream of the cohesin binding site (Fig. 1 A). To assess whether Sgo1$_{130-280}$ can form a complex with the CPC in vitro, a 1.5× molar excess of recombinant Sgo1$_{130-280}$ was mixed with CPC$_{ISB10-280}$, and the mix was analyzed by SEC (Fig. S4 A). SEC profiles and the analysis of SEC fractions showed that Sgo1$_{130-280}$ can indeed form a complex with CPC$_{ISB10-280}$. To further pinpoint the region within Sgo1$_{130-280}$ that is necessary for the interaction with the CPC, we expressed Sgo1$_{130-280}$-LacI-GFP and multiple truncations of the 130–280 fragment in U-2 OS-LacO cells and assessed CPC recruitment through immunofluorescence analysis (Fig. 4 D). A smaller fragment spanning Sgo1 amino acids 159–240 was capable of recruiting similar levels of the CPC as Sgo1$_{130-280}$-LacI-GFP (Fig. 4 D). The region between 159 and 240 contained a highly conserved stretch of hydrophobic amino acids (188–191), and mutation of these residues to alanines (V188/S189/V190/R191A: 4A; Sgo1$_{130-280\ 4A}$-LacI-GFP, Sgo1$_{159-240\ 4A}$-LacI-GFP; Fig. S1 A) completely abrogated CPC recruitment to both Sgo1$_{130-280}$ and Sgo1$_{159-240}$ to the CPC (Fig. 4 D). Interestingly, when we introduced the same 4A mutation in recombinant Sgo1$_{130-280}$ or Sgo1$_{1-415}$, Sgo1$_{130-280\ 4A}$ or Sgo1$_{1-415\ 4A}$, they still managed to interact with CPC$_{ISB10-280}$ in the SEC analysis (Fig. S4, A–C). Moreover, ITC data showed that, in vitro, Sgo1$_{1-415}$ and Sgo1$_{1-415\ 4A}$ can bind CPC$_{ISB10-280}$ with similar affinity (151 ± 35.6 vs. 112 ± 42.2 nM, respectively; Fig. S4, D and E). Considering the substoichiometric amounts of Sgo1$_{130-280}$ observed to coelute with CPC$_{ISB10-280}$ in SEC (based on the SDS-PAGE band intensities observed for the corresponding SEC fractions, Fig. S4, A and B) and that perturbing the central region interaction did not significantly reduce the measured CPC-binding affinity of Sgo1$_{1-415}$ by ITC (Fig. S4, D and E), we conclude that the Sgo1 central region does not make a significant contribution to CPC binding in vitro. However, as the same Sgo1 mutant (Sgo1$_{130-280\ 4A}$-LacI-GFP) is sufficient to perturb CPC–Sgo1 interaction in cells, we propose that Sgo1 central region requires one or more yet-unidentified post-translational modifications to facilitate its interaction with the CPC, either in the Sgo1 region and/or in the CPC.

As our analysis identified two CPC-interacting regions within Sgo1 (the N-terminal 130 aa including the N-terminal tail and the conserved coiled-coil, and the conserved hydrophobic region between aa 188 and 191), we next evaluated their contribution for CPC recruitment in the context of full-length Sgo1 using the LacO-LacI tethering assay. Consistent with our in vitro data, full-length Sgo1, harboring the N-terminal mutation (Sgo1$_{1-527\ Nmut}$-LacI-GFP), recruited less Aurora B or Borealin (Fig. 4 E) compared with the Sgo1$_{1-527}$-LacI-GFP. Similarly, the 4A mutation in the full-length context (Sgo1$_{1-527\ 4A}$-LacI-GFP) also reduced the recruitment of Aurora B and Borealin, while the double mutant (Sgo1$_{1-527\ Nmut/4A}$-LacI-GFP) showed an even stronger reduction of endogenous Aurora B and Borealin recruitment to the LacO array (Fig. 4 E). Collectively, these data demonstrate the contribution of both Sgo1 regions for CPC binding in cells.

## The Survivin interaction with the Sgo1 N-terminal tail is essential for the centromeric localization of the CPC and proper chromosome segregation

We next evaluated how the different Sgo1 regions we identified as important for the CPC–Sgo1 interaction contribute to the centromeric levels of the CPC in cells. Endogenous Sgo1 was depleted by siRNA in HeLa Kyoto cells transiently expressing either wild-type Sgo1 (Sgo1-GFP) or mutant Sgo1 (Sgo1$_{Nmut}$-GFP, Sgo1$_{4A}$-GFP, or Sgo1$_{Nmut/4A}$ double mutant), and centromeric levels of Borealin were analyzed by quantitative immunofluorescence microscopy (Fig. 5 A; Fig. S4, F and G; and Fig. S5 A). Consistent with previous observations (Broad et al., 2020; Kawashima et al., 2007; Meppelink et al., 2015; Tsukahara et al., 2010; van der Waal et al., 2012; Wang et al., 2010), depletion of Sgo1 led to a twofold reduction in the centromeric levels of Borealin. As expected, expression of wild-type Sgo1 (Sgo1-GFP) rescued the centromeric abundance of Borealin (Fig. 5 A). In line with our in vitro binding studies and cellular tethering data, expression of Sgo1 mutants (Sgo1$_{Nmut}$-GFP, Sgo1$_{4A}$-GFP, or Sgo1$_{Nmut/4A}$-GFP), aimed to perturb either the Sgo1–N-terminal tail-Survivin interaction or the Sgo1$_{188-191}$–Borealin interaction, did not rescue the centromeric levels of Borealin, demonstrating that these regions directly contribute to the efficient centromere recruitment of the CPC (Fig. 5 A).

It is known that a complete Sgo1 depletion causes a mitotic arrest due to SAC activation (Kitajima et al., 2005; Kitajima et al., 2006; McGuinness et al., 2005; Salic et al., 2004; Tang et al., 2004). In our siRNA depletion experiments, we observed ~70% reduction in Sgo1 levels (Fig. S4 F) as estimated from

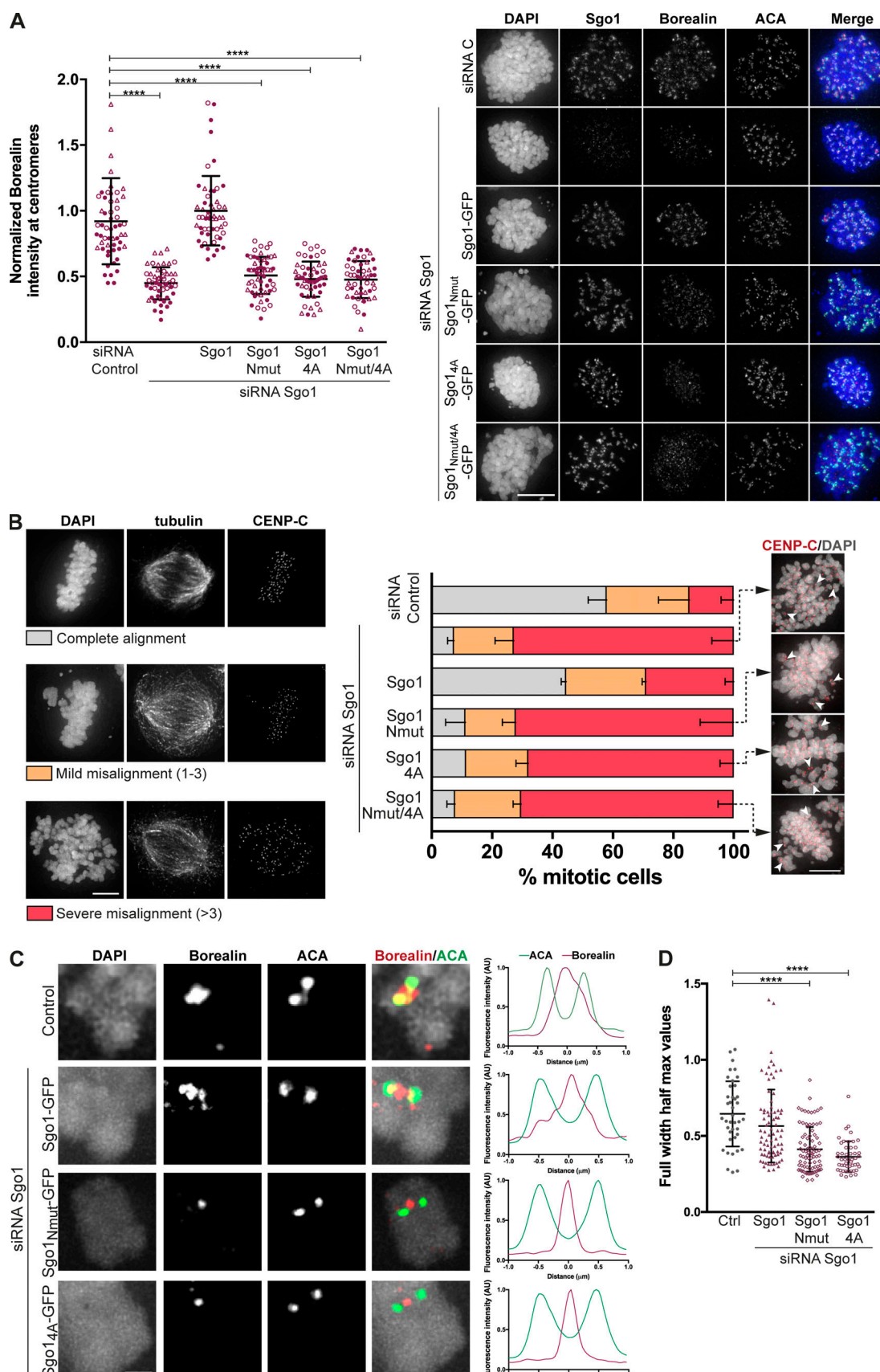

Figure 5. **CPC interaction with the Sgo1 N-terminal tail is essential for the centromeric localization of CPC and proper chromosome segregation.** **(A)** Representative micrographs of HeLa Kyoto cells transiently expressing different Sgo1-GFP constructs (Sgo1-GFP, Sgo1Nmut-GFP, Sgo14A-GFP, or Sgo1Nmut/

$_{4A}$-GFP) and depleted of endogenous Sgo1 using siRNA oligonucleotides (right). Immunofluorescence of endogenous Borealin and ACA. DAPI was used for DNA staining. Scale bar, 10 µm. Quantification of Borealin levels at the centromeres using ACA as reference channel (left). Values normalized to Sgo1 siRNA/Sgo1-GFP condition. Three independent experiments, $n \geq 50$ cells analyzed in total per treatment, mean ± SD, Kruskal–Wallis with Dunn's multiple comparisons test; ****, $P \leq 0.0001$. The values from the three independent replicates are represented in three different symbols. **(B)** Quantification of chromosome alignment of cells subjected to biorientation assay. Transfected cells were treated with 100 µM Monastrol for 16 h and released into medium containing 5 µM MG132 for 1 h. Representative examples of the alignment categories: complete alignment, mild misalignment (with one to three misaligned chromosomes), and severe misalignment (with more than three misaligned chromosomes) are found in the left panel. Representative images of the conditions expressing the three Sgo1 mutants showing pairs of CENP-C foci (red; right panel). DAPI was used to visualize DNA. Scale bar, 5 µm. Three independent experiments, $n \geq 100$ of metaphases analyzed; mean ± SD. **(C)** Line plots depicting normalized fluorescence intensity levels of Borealin and ACA, measured along a line across the two sister ACA signals of the interkinetochore axis. Scale bar, 2 µm. Left, representative images of kinetochore pairs represented in the line plots. **(D)** Quantification of the full width at half maximum for the Borealin signal obtained in the line plots. Three independent experiments, $n \geq 49$ kinetochores, mean ± SD, Kruskal–Wallis with Dunn's multiple comparisons test; ****, $P \leq 0.0001$.

immunoblotting experiments. This allowed a population of cells to progress into anaphase and let us evaluate the consequence of specifically disrupting CPC–Sgo1 interaction on chromosome segregation in these cells. Following the same experimental setup described above, we depleted endogenous Sgo1 using siRNA oligonucleotides in HeLa Kyoto cells transiently expressing wild-type Sgo1 (Sgo1-GFP) or mutant Sgo1 (Sgo1$_{Nmut}$-GFP, Sgo1$_{4A}$-GFP, or Sgo1$_{Nmut/4}$-GFP) and quantified anaphase cells showing lagging chromosomes or chromosome bridges, which are direct indicators of chromosome missegregation (Fig. S5 D). This analysis confirmed that cells expressing Sgo1$_{Nmut}$-GFP, Sgo1$_{4A}$-GFP, or Sgo1$_{Nmut/4A}$-GFP show a high percentage of cells with either lagging chromosomes or chromosome bridges (27.8 ± 5.8, 26.9 ± 3.1, and 29.3 ± 4.4%, respectively) compared to the siRNA C and Sgo1-GFP rescue (6.1 ± 1.1 and 10.3 ± 2.6%, respectively).

We further analyzed the effects of disrupting the CPC–Sgo1 interaction on chromosome biorientation. Sgo1-depleted HeLa cells transiently expressing either Sgo1-GFP or mutant Sgo1 (Sgo1$_{Nmut}$-GFP, Sgo1$_{4A}$-GFP, or Sgo1$_{Nmut/4}$-GFP) were released from a monastrol-induced mitotic arrest into a medium with MG132 for 1 h (Fig. 5 B) and 2 h (Fig. S5 B), and chromosome alignment was assessed. Under these conditions, expression of Sgo1 mutant constructs led to ~70% of the cells showing severe chromosome misalignment, comparable to the phenotype observed for Sgo1 depletion (Figs. 5 B and S5 B). Notably, unlike Sgo1 knockdown cells, Sgo1 mutant–expressing cells did not seem to experience loss of sister chromatid cohesion, because sister centromeres remained close together (Fig. 5 B). This suggests that the alignment errors observed are not due to a loss of centromeric cohesion, but a reflection of perturbed KT-MT error correction, presumably because of the reduced centromeric levels of CPC (Fig. 5 A). Finally, we also examined chromosome alignment in asynchronously growing cells transiently expressing the Sgo1 mutants (Fig. S5 C) and observed similar chromosome alignment defects as after monastrol release and MG132 treatment.

Considering the ability of Sgo1 to bind H2AT120ph and to recruit CPC to the kinetochore-proximal centromere, we analyzed the precise localization of Borealin using chromosome spreads of nocodazole-arrested HeLa cells expressing the Sgo1 mutants. Control HeLa cells or Sgo1 depletion in Sgo1-GFP–expressing HeLa cells displayed Borealin localization at the inner centromere with a small pool localized at the

kinetochore-proximal centromere (Fig. 5 C), consistent with the previously described pattern of CPC localization in unperturbed mitotic cells (Bekier et al., 2015; Hadders et al., 2020; Liang et al., 2020). In contrast, depletion of Sgo1 on Sgo1$_{Nmut}$-GFP or Sgo1$_{4A}$-GFP expressing cells, Borealin was enriched as a single focus between the two sister ACA dots, similar to the inner centromere localization previously observed for Borealin dimerization mutants that bind less well to Sgo1 (Bekier et al., 2015). Quantification of the full width at half maximum values for the Borealin intensity profiles obtained from the line plots of the chromosome spreads confirmed that rescue of Sgo1 depletion with Sgo1-GFP expression generated a broader Borealin signal at the centromere (most likely the result of the combination of inner centromere and kinetochore-proximal centromere pools), while expression of Sgo1 mutants (Sgo1$_{Nmut}$-GFP and Sgo1$_{4A}$-GFP) generated narrower Borealin profiles consistent with CPC localized at the inner centromere only (Fig. 5 D). These data reveal that the interaction of CPC with H2AT120ph-bound Sgo1 is responsible for the kinetochore-proximal centromere pool of the CPC.

## Discussion

Concentration of the CPC near centromeres during early mitosis facilitates accurate chromosome congression and segregation in many organisms (Carmena et al., 2012; Dai et al., 2005; Krenn and Musacchio, 2015; Liu et al., 2009; Tanaka et al., 2002; van der Horst and Lens, 2014; Wang et al., 2010; Wang et al., 2012; Wang et al., 2011; Welburn et al., 2010). Two histone phosphorylation marks, Haspin-mediated H3T3ph and Bub1-mediated H2AT120ph, ensure the inner and kinetochore-proximal centromere enrichment of the CPC, respectively (Broad et al., 2020; Hadders et al., 2020; Liang et al., 2020). Several independent studies have provided molecular and structural understanding of how the Survivin subunit of the CPC directly recognizes the H3T3ph mark and its flanking amino acid residues, including the free amino terminus (Du et al., 2012; Jeyaprakash et al., 2011; Kelly et al., 2010; Niedzialkowska et al., 2012; Serena et al., 2020; Wang et al., 2010). Unlike the H3T3ph mark, the H2AT120ph is indirectly recognized by the CPC via Sgo1, which is capable of directly binding H2AT120ph via its C-terminal Sgo motif (Liu et al., 2015). As far as the CPC–Sgo1 interaction is concerned, the coiled-coil (Tsukahara et al., 2010) and dimerization domains of Borealin (Bonner et al., 2020) and Survivin BIR domain

(Jeyaprakash et al., 2011) have been implicated in direct Sgo1 binding. However, whether these interactions (Borealin–Sgo1 and Survivin–Sgo1) take place in the context of the CPC and their relative contribution for CPC–Sgo1 binding and centromere localization has remained unresolved.

Here we show that Sgo1 forms a tight complex with the $CPC_{ISB10-280}$ in vitro and that the interaction between the Histone H3-like N-terminal region of Sgo1 and the BIR domain of Survivin is crucial for CPC–Sgo1 complex formation, while the interaction of ~120 amino acid residues downstream of the Sgo1 N-terminal tail with Borealin, and possibly INCENP, are required for high-affinity binding. We previously showed that the CPC binds H3T3ph nucleosomes with a $K_D$ of ~90 nM (Abad et al., 2019). This value is comparable to the measured $K_D$ for CPC–Sgo1 binding, and notably, both interactions rely on a micromolar affinity interaction involving the Survivin BIR domain and the N-terminal tails of H3 and Sgo1, respectively. In addition, we also identified a hydrophobic region in Sgo1, comprising aa 188–191, that was required for the CPC–Sgo1 interaction. Mutations of this motif in Sgo1 abrogated the interaction with the CPC as well as CPC centromere recruitment in cells. However, while a Sgo1 fragment surrounding aa 188–191 could bind the $CPC_{ISB10-280}$ in vitro, this region did not appear to further contribute to the $Sgo1_{1-415}$-$CPC_{ISB10-280}$ interaction in vitro. Therefore, we deem it likely this interaction is mediated by yet-unidentified posttranslational modification(s) on Sgo1 and/or CPC.

Several independent studies proposed that Haspin-mediated H3T3ph and Bub1-mediated H2AT120ph recruit the CPC to centromeres independently and as distinct spatial pools, inner centromere and kinetochore-proximal centromere pools, respectively (Bekier et al., 2015; Broad et al., 2020; Hadders et al., 2020; Liang et al., 2020). Our detailed molecular mapping of the CPC–Sgo1 interaction provided an excellent opportunity to test this model using Sgo1 separation-of-function mutants. Importantly, it has been previously suggested that the kinetochore proximal pool is independent of H2AT120ph/Sgo1 (Bekier et al., 2015). However, we observed that Sgo1 mutations that specifically perturb CPC binding ($Sgo1_{Nmut}$ and $Sgo1_{4A}$) mainly affect the kinetochore-proximal centromere pool of the CPC while leaving the inner centromere pool largely intact, indicating Sgo1 as a main kinetochore-proximal centromere receptor for the CPC. This is in line with the observation that inhibition of Bub1 leads to loss of kinetochore-proximal centromere CPC in Haspin KO cells (Hadders et al., 2020). Finally, the aforementioned Sgo1 mutations led to chromosome misalignment and segregation errors. These observations suggest that the H2AT120ph-mediated kinetochore-proximal centromere pool of the CPC could indeed play a role in error correction (Hadders et al., 2020), in addition to a proposed role for this pool in SAC inhibition (Liang et al., 2020). Our data will guide future research that aims to couple specific CPC functions to the distinct CPC pools.

The observation that the Sgo1 and histone H3 N-terminal tails exploit the same binding site in Survivin suggests that these interactions could be mutually exclusive and may explain why the Bub1-dependent CPC pool exists as a kinetochore-proximal

centromere pool that is spatially distinct from the Haspin-dependent inner centromere CPC pool (Broad et al., 2020; Hadders et al., 2020; Liang et al., 2020). Moreover, our observation that the H3T3ph binding deficient Survivin BIR mutants (K62A and H80A) retain Sgo1 binding is in line with previous findings from Liang et al. (2020), showing that these mutants retain their ability to form a kinetochore-proximal CPC pool. It further suggests that subtle differences in H3T3ph and Sgo1 binding mediated by Survivin BIR domain might contribute to the spatiotemporal control of the CPC pools along the intersister KT axis.

## Materials and methods

### Protein expression and purification of CPC and Sgo1

$CPC_{ISB10-280}$ and $CPC_{ISB10-221}$ were purified as previously described (Abad et al., 2019). Briefly, pRSET-His-GFP-3C-Survivin (pRSET vector from Thermo Fisher Scientific), pMCNcs-$INCENP_{1-58}$, and pETM-His-TEV-$Borealin_{10-280}$ or pETM-His-TEV-$Borealin_{10-221}$ (pETM vector, gift from C. Romier, Institute of Genetics and Molecular and Cell Biology, Strasbourg, France) were cotransformed in BL21(DE3) pLysS. Cultures were grown at 37°C until OD 0.8 and induced overnight at 18°C with 0.35 mM IPTG. Cells were lysed in lysis buffer (25 mM Hepes, pH 7.5, 500 mM NaCl, 25 mM imidazole, and 2 mM β-mercaptoethanol) and supplemented with complete EDTA-free cocktail tablets (Roche), 0.01 mg/ml DNase (Sigma-Aldrich), and 1 mM PMSF. The lysate was sonicated for 8 min and centrifuged at 58,000 $g$ for 50 min at 4°C, and the complex was purified by affinity chromatography using His Trap Column (Cytiva). The protein-bound column was washed with lysis buffer followed by a high salt buffer wash (25 mM Hepes, pH 7.5, 1 M NaCl, 25 mM imidazole, 2 mM β-mercaptoethanol, and 1 mM ATP). The complex was eluted using high imidazole buffer (25 mM Hepes, pH 7.5, 500 mM NaCl, 400 mM imidazole, and 2 mM β-mercaptoethanol), and the affinity tags were cleaved using TEV and 3C proteases while dialyzing the sample in a buffer containing 25 mM Hepes, pH 7.5, 150 mM NaCl, and 4 mM DTT at 4°C overnight. The dialyzed sample was then loaded onto a HiTrap SP HP (Cytiva) cation exchange column to separate the excess Borealin–Survivin complex and GFP from the $CPC_{ISB}$ complexes. The samples containing stoichiometric and pure $CPC_{ISB}$ complex were pooled, concentrated, and run on a Superdex 200 Increase 10/300 (Cytiva) pre-equilibrated with 25 mM Hepes, pH 8, 150 mM NaCl, 5% glycerol, and 4 mM DTT.

Sgo1 fragments ($Sgo1_{1-415}$, $Sgo1_{1-155}$, and $Sgo1_{1-130}$) were cloned in the pTYB11 vector (IMPACT system; New England Biolabs) which contains an Intein tag with an embedded chitin-binding domain. The Intein tag is a DTT-induced self-cleavable tag that allows purification of proteins with a native N-terminus, as it leaves no extra amino acids after cleavage. Cloning of the Sgo1 in the pTYB11 vector with an N-terminal Intein-tag allowed the purification of an Sgo1 with a native N-terminus, leaving the initiator methionine exposed to be excised by methionine aminopeptidases (Giglione et al., 2004). Sgo1 fragments were expressed in the BL21 (*DE3*) Gold *Escherichia coli* strain. Cells were grown at 37°C to OD 1.5 and induced

overnight at 18°C with 0.35 mM IPTG. Cells were resuspended in lysis buffer containing 20 mM Tris-HCl, pH 7.5, 500 mM NaCl, and 1 mM EDTA and supplemented with complete EDTA-free cocktail tablets (1 tablet/50 ml cells; Roche), 0.01 mg/ml DNase (Sigma-Aldrich), and 1 mM PMSF. The lysate was sonicated for 8 min and centrifuged at 58,000 $g$ for 50 min at 4°C, and the protein was batch purified using chitin beads (New England Biolabs). Protein-bound chitin beads were washed with lysis buffer and high salt buffer (20 mM Tris-HCl, pH 7.5, 1 M NaCl, 1 mM EDTA, and 1 mM ATP) and eluted with 20 mM Tris-HCl, pH 7.5, 500 mM NaCl, and 50 mM DTT overnight at room temperature. The eluted protein was then dialyzed into 20 mM Tris-HCl, pH 7.5, 150 mM NaCl, 50 mM glutamate, 50 mM arginine, and 2 mM DTT overnight at 4°C and loaded onto a HiTrap SP-HP (Cytiva) ion exchange column. The samples containing Sgo1 were pooled, concentrated, and run in a Superdex 200 Increase 10/300 column (Cytiva) pre-equilibrated with 25 mM Hepes, pH 8, 250 mM NaCl, 5% glycerol, and 2 M DTT.

Sgo1$_{130-280}$ was cloned in a pEC-S-CDF-His vector as N-terminally His-tagged. Sgo1$_{130-280}$ $_{4A}$ was generated using the Quickchange site-directed mutagenesis method (Stratagene). The vectors were transformed in BL21 Gold cells and grown and induced as described above. Cells were resuspended in lysis buffer containing 20 mM Tris-HCl, pH 8, 500 mM NaCl, 35 mM imidazole, and 2 mM β-mercaptoethanol and supplemented with complete EDTA-free cocktail tablets (1 tablet/50 ml cells; Roche), 0.01 mg/ml DNase (Sigma-Aldrich), and 1 mM PMSF. The protein was purified using a HisTrap column (Cytiva). The protein-bound column was washed with lysis buffer and high salt buffer (20 mM Tris-HCl, pH 8, 1 M NaCl, 35 mM imidazole, and 2 mM β-mercaptoethanol) and eluted with 20 mM Tris-HCl, pH 8, 200 mM NaCl, 400 mM imidazole, and 2 mM β-mercaptoethanol. The eluted protein was then dialyzed into 20 mM Tris-HCl, pH 8, 200 mM NaCl, and 1 mM DTT overnight at 4°C and loaded onto a HiTrap Q (Cytiva) ion exchange column. The samples containing Sgo1 were pooled, concentrated, and run in a Superdex 200 Increase 10/300 column (Cytiva) pre-equilibrated with 25 mM Hepes, pH 8, 150 mM NaCl, 5% glycerol, and 1 mM DTT.

### Interaction studies using SEC
All SEC experiments for the purified recombinant proteins were performed on an AKTA Pure 25 HPLC unit (Cytiva) with sample collector. For all interaction studies, a Superdex 200 10/300 GL 24 ml column (Cytiva) was used at 4°C. Before sample injection, the column was pre-equilibrated in 25 mM Hepes, pH 7.5, 150 mM NaCl, 4 mM DTT, and 5% glycerol (vol/vol) for interaction experiments involving Sgo1$_{1-155}$ or pre-equilibrated in 25 mM Hepes, pH 8, 250 mM NaCl, 2 mM DTT, and 5% glycerol (vol/vol) for interaction experiments involving Sgo1$_{1-415}$. 0.5-ml fractions were collected with a 0.2–column volume delayed fractionation setting. UV 280- and 260-nm wavelengths were monitored. A 1.5× to 2× molar excess of Sgo1 was used in all interaction studies with CPC. Proteins were mixed and incubated at 4°C for 1 h before being injected to the size exclusion column.

### Chemical cross-linking and MS analysis
Cross-linking experiments of Sgo1$_{1-415}$ and CPC$_{ISB10-280}$ were performed using EDC (Thermo Fisher Scientific) in the presence of $N$-hydroxysulfosuccinimide (Thermo Fisher Scientific). 25 µg of gel-filtrated protein complex was cross-linked with 20 µg EDC and 44 µg of $N$-hydroxysulfosuccinimide in 25 mM Hepes, pH 6.8, and 150 mM NaCl for 1 h 30 min at room temperature. The cross-linking was stopped by the addition of 100 mM Tris-HCl, and cross-linking products were briefly resolved using 4–12% Bis-Tris NuPAGE (Thermo Fisher Scientific). Bands were visualized by short Instant Blue staining (Abcam), excised, reduced with 10 mM DTT for 30 min at room temperature, alkylated with 5 mM iodoacetamide for 20 min at room temperature, and digested overnight at 37°C using 13 ng/µl trypsin (Promega). Digested peptides were loaded onto C18-Stage-tips (Rappsilber et al., 2007). Liquid chromatography with tandem mass spectrometry (LC-MS/MS) analysis was performed using an Orbitrap Fusion Lumos Tribrid Mass Spectrometer (Thermo Fisher Scientific) applying a "high-high" acquisition strategy. Peptide mixtures were injected for each mass spectrometric acquisition. Peptides were separated on a 75 µm × 50 cm PepMap EASY-Spray column (Thermo Fisher Scientific) fitted into an EASY-Spray source (Thermo Fisher Scientific), operated at 50°C column temperature. Mobile phase A consisted of water and 0.1% vol/vol formic acid. Mobile phase B consisted of 80% vol/vol acetonitrile and 0.1% vol/vol formic acid. Peptides were loaded at a flow-rate of 0.3 µl/min and eluted at 0.2 µl/min using a linear gradient going from 2% mobile phase B to 40% mobile phase B over 139 (or 109) min, followed by a linear increase from 40 to 95% mobile phase B in 11 min. The eluted peptides were directly introduced into the mass spectrometer. MS data were acquired in the data-dependent mode with the top-speed option. For each 3-s acquisition cycle, the mass spectrum was recorded in the Orbitrap with a resolution of 120,000. The ions with a precursor charge state between 3+ and 8+ were isolated and fragmented using higher-energy collisional dissociation (HCD) or electron-transfer/HCD (EThcD). The fragmentation spectra were recorded in the Orbitrap. Dynamic exclusion was enabled with single repeat count and 60-s exclusion duration.

The mass spectrometric raw files were processed into peak lists using ProteoWizard (v3.0.20338; Kessner et al., 2008), and cross-linked peptides were matched to spectra using Xi software (v1.7.6.3; Mendes et al., 2018; https://github.com/Rappsilber-Laboratory/XiSearch) with in-search assignment of mono-isotopic peaks (Lenz et al., 2018). Search parameters were MS accuracy, 3 ppm; MS/MS accuracy, 10 ppm; enzyme, trypsin; cross-linker, EDC; max missed cleavages, 4; missing mono-isotopic peaks, 2; fixed modification, carbamidomethylation on cysteine; variable modifications, oxidation on methionine; and fragments b and y type ions (HCD) or b, c, y, and z type ions (EThcD) with loss of $H_2O$, $NH_3$, and $CH_3SOH$. 1% on link level false discovery rate was estimated based on the number of decoy identification using XiFDR (Fischer and Rappsilber, 2017). The MS proteomics data have been deposited to the ProteomeXchange Consortium via the PRIDE (Perez-Riverol et al., 2019) partner repository. Data are available via ProteomeXchange with identifier PXD028433.

## ITC

ITC experiments were performed using a MicroCal Auto-iTC200 (Malvern Instruments). A total of 40 µl of 50–375 µM (monomer concentration) Survivin/CPC complexes was injected into 200 µl of 5–20 µM (monomer concentration) hSgo1 constructs in 16 aliquots (1 × 0.5 µl and 15 × 2.5 µl), 180 s between injections, reference power 3 µcal/s$^{-1}$, syringe spin 750 rpm, and filter period 5 s. Control titrations were performed in which the injectant was added to buffer without protein or buffer was injected into the protein. Titrations were carried out at 20°C, except for the analysis of the Survivin/Sgo1$_{AKER}$ interaction, which was performed at 10°C. The heat of reaction was corrected for the heat of dilution and analyzed using the MicroCal ITC software v1.30 (Malvern Instruments). All experiments were carried out in 50 mM Hepes, pH 8, 150 mM NaCl, 5% (vol/vol) glycerol, and 1 mM tris(2-carboxyethyl)phosphine (TCEP), except the experiment to assess the binding affinity between CPC$_{10–280}$ and Sgo1$_{1–415}$ or Sgo1$_{1–415}$ 4A (Fig. S4, D and E) that was carried out at 50 mM Hepes, pH 8, 250 mM NaCl, 5% (vol/vol) glycerol, 1 mM TCEP, and 0.005% Tween.

## Mass photometry

High-precision microscope coverslips (no. 1.5, 24 × 50 mm) were cleaned with Milli-Q water, 100% isopropanol, Milli-Q water, and dried. Silicone gaskets (103250; Grace BioLabs) were placed on the coverslips. Samples were cross-linked with 0.01% glutaraldehyde for 5 min at 4°C and quenched by addition of 50 mM Tris-HCl, pH 7.5, for 1 h at 4°C. Immediately before mass photometry measurements, samples were diluted to 100 nM in buffer containing 25 mM Hepes, pH 8, 250 mM NaCl, and 2 mM DTT. For each acquisition, 20 nM of diluted protein was measured following manufacturer's instructions. All data was acquired using a One$^{MP}$ mass photometer instrument (Refeyn) and AcquireMP software (Refeyn, v2.4.1). Videos were recorded in the regular field of view using default settings. Data was analyzed using Discover$^{MP}$ software (Refeyn, v2.4.2).

## Tethering assays

The LacO tethering assays were performed essentially as described before (Hadders et al., 2020). U-2 OS LacO Haspin CM cells (Hadders et al., 2020) were seeded on glass coverslips and directly transduced with recombinant baculovirus expressing LacI-GFP fusion proteins. After ~4–6 h, S-trityl-L-cysteine (20 µM) was added and left to incubate overnight. The next morning cells were fixed in 4% PFA for 15 min and permeabilized with ice-cold methanol. Before staining, cells were blocked in PBS supplemented with 0.01% Tween20 (PBST) and 3% BSA for 30 min followed by staining with primary antibodies in PBST + 3% BSA for 2–4 h. Coverslips were then washed three times with PBST followed by staining with secondary antibodies and DAPI (1 µg/ml) for 1 h. After another three washes with PBST, coverslips were mounted using Prolong Diamond. Cells were imaged on a DeltaVision system. The following antibodies were used for indirect immunofluorescence: anti–Aurora B (mouse monoclonal; 1:1,000; 611083; BD Transductions), anti-Borealin (1:1,000; rabbit polyclonal; a kind gift from Dr. S. Wheatley, School of Life Sciences, Medical School, Queen's Medical Centre, University of Nottingham, Nottingham, UK), and GFP-Booster (1:1,000; alpaca monoclonal; gba488; Chromotek). The secondary antibodies used were goat anti-mouse IgG Alexa Fluor 568 conjugate (1:500; A-11031; Thermo Fisher Scientific), goat anti-mouse IgG Alexa Fluor 647 conjugate (1:500; A-1103121236; Thermo Fisher Scientific), goat anti-rabbit IgG Alexa Fluor 568 conjugate (1:500; A-11036; Thermo Fisher Scientific), goat anti-rabbit IgG Alexa Fluor 647 conjugate (1:500; A-21245; Thermo Fisher Scientific), and goat anti-chicken IgY Alexa Fluor 568 conjugate (1:500; A-11041; Thermo Fisher Scientific).

## Rescue experiments and immunofluorescence microscopy

Sgo1 was cloned in the pCDNA3-GFP vector (6D; a gift from Scott Gradia, California Institute for Quantitative Biosciences, University of California, Berkeley, Berkeley, CA; plasmid #30127; Addgene; http://n2t.net/addgene:30127; RRID: Addgene_30127). Mutations of Sgo1 were obtained using the Quikchange site-directed mutagenesis method (Stratagene). Transient DNA transfection (700 ng) was performed using jetPRIME (Polyplus Transfection) according to the manufacturer's instructions. Lipofectamine RNAimax was used for depletion of endogenous Sgo1 using the following oligonucleotide: 5′-UGCACCAUGC-CAAUAAdTdT-3′ (40 pmol). Luciferase targeting was used as a control (5′-CGUACGCGGAAUACUUCGAdTdT-3′; Elbashir et al., 2001). All siRNA oligonucleotides were purchased from Qiagen. Cells were plated in glass coverslips, transfected with jetPRIME (DNA transient transfection) 16 h after plating, and transfected with Lipofectamine siRNA max (siRNA oligonucleotides) 24 h after the first transfection. For centromeric quantification of the Borealin signal, HeLa Kyoto cells were synchronized with 50 ng/ml nocodazole for 16 h, 8 h after siRNA transfection. A minimum of 50 cells per condition were quantified. The acquired images were processed by constrained iterative deconvolution using SoftWoRx 3.6 software package (Applied Precision), and the centromere intensity of Borealin was quantified using an ImageJ plugin (https://doi.org/10.5281/zenodo.5145584). Briefly, the plugin quantifies the mean fluorescence signal of Borealin and Sgo1 in a 3-pixel-wide ring immediately outside the centromere, defined by the ACA staining. For background subtraction, a selected area within the cytoplasm signal was selected. To compare data from different replicates, values obtained after background correction were averaged and normalized to the mean of Borealin intensity in the Sgo1-GFP rescue condition. Statistical significance of the difference between normalized intensities at the centromere region was established by a Kruskal–Wallis test with Dunn's multiple comparisons test using Prism 7.0.

Quantification of anaphases displaying chromosome bridges or lagging chromosomes was performed 24 h after HeLa Kyoto cells were transfected with the siRNA oligonucleotides. For the Monastrol assay, HeLa Kyoto cells were synchronized with 100 µM Monastrol for 16 h and released into 5 µM MG132 for 1 or 2 h. Observed metaphases were classified as complete alignment, mild misalignment (one to three unaligned chromosomes), and severe misalignment (more than three unaligned chromosomes). Quantification of chromosome alignment errors in unperturbed asynchronous cells was performed as described above. The experiments were performed in triplicate, and a minimum of 85 cells per condition were quantified.

For the chromosome spreads, 8 h after siRNA oligonucleotide transfection, HeLa cells were treated with 50 ng/ml Nocodazole. 16 h after Nocodazole treatment, cells were collected by mitotic shake-off and incubated in hypotonic buffer (75 mM KCl) at 37°C for 10 min. After attachment to glass coverslips using Cytospin at 1,800 rpm for 5 min, chromosome spreads were extracted with ice-cold PBS/0.2% Triton X-100 for 4 min and fixed with 4% PFA. The immunofluorescence was performed as described below. Three replicates were performed, and a minimum of 49 kinetochores were analyzed. The centroids of kinetochores were detected in ImageJ using Speckle TrackerJ (Smith et al., 2011) software. A custom ImageJ script (https://doi.org/10.5281/zenodo.5235670) was then used to assign kinetochore pairs as closest neighbors, with a maximum separation of 1.5 μm. The fluorescence intensities along 2-μm line regions of interest through the centroids and centered on the midpoint of the pair were taken in both the channels. Full width at half maximum values for the Borealin line plots were calculated by linear interpolation using a combination of the point-slope formula and the slope formula. Statistical significance of the difference between the full width at half maximum values between different Sgo1 constructs was established by a Kruskal–Wallis test with Dunn's multiple comparisons test using Prism 7.0.

In all cases, cells were fixed in 4% PFA 48 h after DNA transfection and 24 h after oligonucleotide transfection. Cells were then permeabilized with permeabilization buffer (0.2% Triton X-100 in 1× PBS) for 10 min, blocked with 3% BSA in permeabilization buffer for 1 h, and incubated with primary and secondary antibodies in blocking buffer for 1 h each.

All experiments were performed in triplicate.

The following antibodies were used for indirect immunofluorescence: rabbit anti-Sgo1 (1:300; kind gift from Ana Losada, Spanish National Cancer Research Centre, Madrid, Spain; Serrano et al., 2009), mouse anti-Borealin (1:500; 147-3; MBL), mouse anti-tubulin (1:2,000; B512; T5168; Sigma-Aldrich), rabbit anti–CENP-C (1:400; kind gift from William C. Earnshaw, Wellcome Centre for Cell Biology, University of Edinburgh, Edinburgh, UK), and human anti-ACA (1:300; 15-235; Antibodies Inc.). The secondary antibodies used were FITC-conjugated AffiniPure donkey anti-rabbit IgG, TRITC-conjugated AffiniPure goat anti-rabbit IgG, TRITC-conjugated AffiniPure donkey anti-mouse, Cy5-conjugated AffiniPure donkey anti-human, and Cy5-conjugated AffiniPure donkey anti-mouse (1:300; 711-095-152, 111-025-006, 715-025-150, 709-175-149, and 715-175-151, respectively; Jackson ImmunoResearch). Vectashield Antifade Medium with DAPI (Vectashield Laboratories) was used as mounting medium. Imaging was performed at room temperature using a wide-field DeltaVision Elite (Applied Precision) microscope with Photometrics Cool Snap HP camera and 100× NA 1.4 Plan Apochromat objective with oil immersion (refractive index = 1.514) using SoftWoRx 3.6 (Applied Precision) software. Shown images are deconvolved and maximum-intensity projections.

### Western blot

To study Sgo1 levels after siRNA oligo treatment and to test the expression levels of each of the Sgo1-GFP constructs, HeLa Kyoto cells were transfected in 12-well dishes as described above for the rescue experiments and fluorescence microscopy, lysed in 1× Laemmli buffer, boiled for 5 min, and analyzed by SDS-PAGE followed by Western blotting. The antibodies used for the immunoblot were rabbit anti-Sgo1 antibody (1:1,000; a gift from Ana Losada's laboratory, Spanish National Cancer Research Centre, Madrid, Spain; Serrano et al., 2009), mouse anti-tubulin (1:10,000; ab18251; Abcam), and rabbit anti-GFP (1:1,000; Abcam). Secondary antibodies used were goat anti-mouse 680, donkey anti-rabbit 800, and donkey anti-mouse 800 (LI-COR) at 1:2,000 dilution. Immunoblots were imaged using the Odyssey CLx system, and band intensities were quantified using ImageJ, uncalibrated OD values. Values were then corrected by the corresponding tubulin levels (loading control) and normalized to siRNA control values. Three experimental replicates were analyzed.

### Statistical methods

In the graphs corresponding to the Sgo1 siRNA and rescue experiments with Sgo1-GFP WT and mutants, mean ± SD was plotted. Data derived from the different conditions were compared using either a Kruskal–Wallis test with Dunn's multiple comparisons test, a $\chi^2$ test, or a Student's two-tailed unpaired $t$ test using Prism 7.0. When parametric tests were used, normality was tested using a Shapiro–Wilk normality test using Prism 7.0. The tethering assays were analyzed using a one-way ANOVA with Dunnett's multiple comparisons test. Data were considered statistically different at $P \leq 0.05$ with a single asterisk, at $P \leq 0.01$ with two asterisks, at $P \leq 0.001$ with three asterisks, and at $P \leq 0.0001$ with four asterisks.

### Online supplemental material

Fig. S1 shows the sequence alignment of Sgo1 orthologues and the mass photometry histograms and kernel density estimates that support Fig. 1. Fig. S1 also shows the crosslinking SDS-PAGE that supports Fig. 2 and a cartoon representation of the Survivin-Sgo1$_{AKER}$ structure and Survivin/Sgo1 isotherms that support Fig. 3. Fig. S2 shows SEC profiles and ITC isotherms that highlight the importance of the BIR domain of Survivin and the N-terminus of Sgo1 for CPC/Sgo1 interaction, supporting Fig. 3. Fig. S3 shows the in vivo LacO-LacI tethering assays and the ITC data that support the contribution of the BIR domain of Survivin and the N-terminus of Sgo1 for CPC/Sgo1 interaction, supporting Fig. 3. Fig. S4 shows SEC profiles and ITC isotherms corresponding to the Sgo1$_{4A}$ mutant, supporting Fig. 4. Fig. S4 also shows the Western blots corresponding to Sgo1 depletion and Sgo1 transient expression, supporting Fig. 5. Fig. S5 shows centromere localization of Sgo1 constructs (right) and quantification of Sgo1 intensities at centromeres (left), supporting Fig. 5. It also shows the quantification of chromosome alignment and segregation defects observed for Sgo1 mutants, supporting Fig. 5.

## Acknowledgments

We thank the staff of the Edinburgh Protein Production Facility and the Centre for Optical Instrumentation Laboratory for their help. We also thank Ana Losada (Chromosome Dynamics Group,

Molecular Oncology Programme, Spanish National Cancer Research Centre, Madrid, Spain) for kindly providing the Sgo1 antibody and William C. Earnshaw (Wellcome Centre for Cell Biology, University of Edinburgh, Edinburgh, UK) for kindly providing the CENP-C antibody. We thank Refeyn Ltd. (Oxford, UK), especially Akhila Bettadapur and Tomás de Garay, for their help with the mass photometry experiments. We thank Cristina Ferrás and Marco Cruz (Instituto de Biologia Molecular e Celular, Universidade do Porto, Portugal) for sharing with us the Sgo1 siRNA oligonucleotide sequence. We thank Bethan Medina-Pritchard for critical reading of the manuscript.

The Wellcome Trust generously supported this work through a Career Development and Enhancement Grant (095822) and Senior Research Fellowship (202811) to A.A. Jeyaprakash and a Centre Core Grant (203149) and a Core Grant (109916/Z/15/Z) to the Edinburgh Protein Production Facility. This study was also supported by a research grant from the Dutch Cancer Society (KWF grant 10366) to S.M.A. Lens. Moreover, the Lens lab is part of Oncode Institute, which is partly financed by the Dutch Cancer Society. Tanmay Gupta was funded by the Darwin Trust of Edinburgh.

The authors declare no competing financial interests.

Author contributions: A.A. Jeyaprakash conceived the project. M.A. Abad, T. Gupta, M.A. Hadders, A. Meppelink, J.P. Wopken, E. Blackburn, D.A. Kelly, T. McHugh, J. Rappsilber, and S.M.A. Lens designed the experiments. M.A. Abad, T. Gupta, E. Blackburn, J. Zou, A. Gireesh, and L. Buzuk performed the biochemical, structural, and biophysical characterization. M.A. Hadders, A. Meppelink, and J.P. Wopken performed the tethering assays. M.A. Abad performed the functional assays. M.A. Abad, M.A. Hadders, E. Blackburn, S.M.A. Lens, and A.A. Jeyaprakash wrote the manuscript.

Submitted: 30 August 2021

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

# Supplemental material

Abad et al.

Molecular basis for CPC–Sgo1 interaction

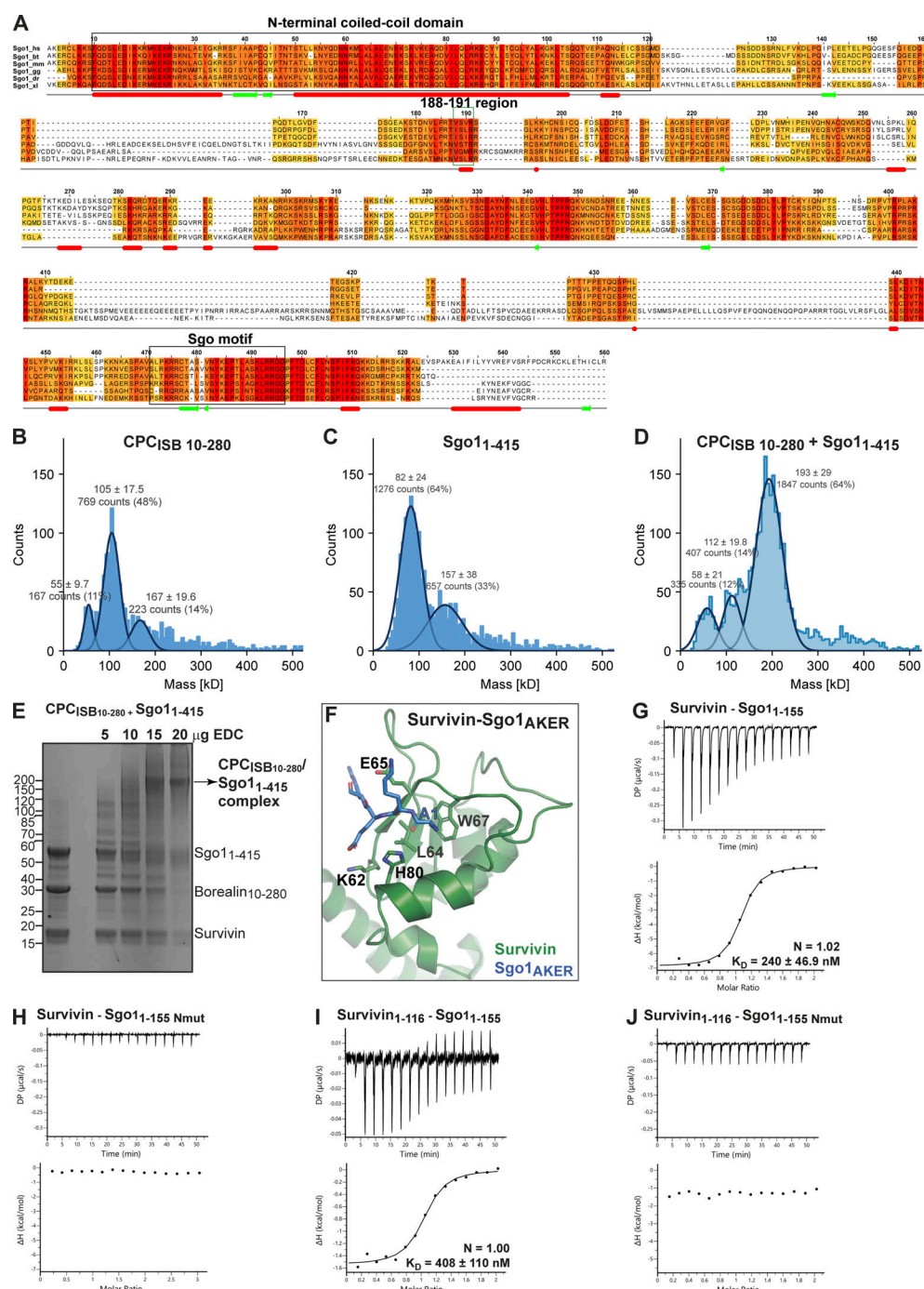

Figure S1. **CPC and Sgo1 interact in vitro. (A)** Sequence alignment of Sgo1 orthologues from *Homo sapiens* (hs), *Bos taurus* (bt), *Mus musculus* (mm), *Gallus gallus* (gg), *Danio rerio* (dr), and *Xenopus laevis* (xl). The conservation score is mapped from red (highly conserved) to yellow (poorly conserved). Predicted secondary structure elements are shown below the sequence alignment. Multiple sequence alignment was performed with Clustal Omega (EMBL-EBI) and edited with Jalview 2.11.0 (Waterhouse et al., 2009). Highlighted with boxes are the N-terminal coiled-coil domain of Sgo1, the highly conserved 188–191 region, and the Sgo motif. The N-terminal AKER motif of Sgo1 is well conserved in most higher vertebrates. **(B–D)** Resulting mass photometry histograms and kernel density estimates for $CPC_{ISB10-280}$ (B), $Sgo1_{1-415}$ (C), and $CPC_{ISB10-280}$ /$Sgo1_{1-415}$ complex (D). All samples were cross-linked with 0.01% glutaraldehyde for 5 min at 4°C. Mean ± SD. **(E)** Representative SDS-PAGE analysis of $CPC_{ISB10-280}$ cross-linked with $Sgo1_{1-415}$ using EDC chemical cross-linker. **(F)** Close-up of the crystal structure of Survivin bound to a peptide comprising the four first amino acid residues of Sgo1 (AKER peptide; PDB accession no. 4A0I; Jeyaprakash et al., 2011). $Sgo1_{Nmut}$ disrupts the interaction between the first amino acid of Sgo1 and the shallow hydrophobic pocket of Survivin. Mutation of amino acids Lys62, Glu65, and His80 in the Survivin BIR domain to alanine disrupt the crucial interactions with the AKER N-terminal tail of Sgo1. **(G and H)** Isotherms for the analyses of Survivin interaction with $Sgo1_{1-155}$ (G) and $Sgo1_{1-155\ Nmut}$ (H). **(I and J)** Isotherms for the analyses of $Survivin_{1-116}$ interaction with $Sgo1_{1-155}$ (I) and $Sgo1_{1-155\ Nmut}$ (J). The ITC experiments were performed with 16 × 2.5-µl injections of 200 µM Survivin or $Survivin_{1-116}$ into 200 µl of 20 µM $Sgo1_{1-155}$ or $Sgo1_{1-155\ Nmut}$ (0.5 µl first injection), 180 s apart, at 20°C. Top panels show raw ITC data; bottom panels show integrated heat data corrected for heat of dilution and fitted to a standard 1:1 binding model (Malvern Instruments MicroCal Origin software, v1.3). DP, differential power. Source data are available for this figure: SourceData FS1.

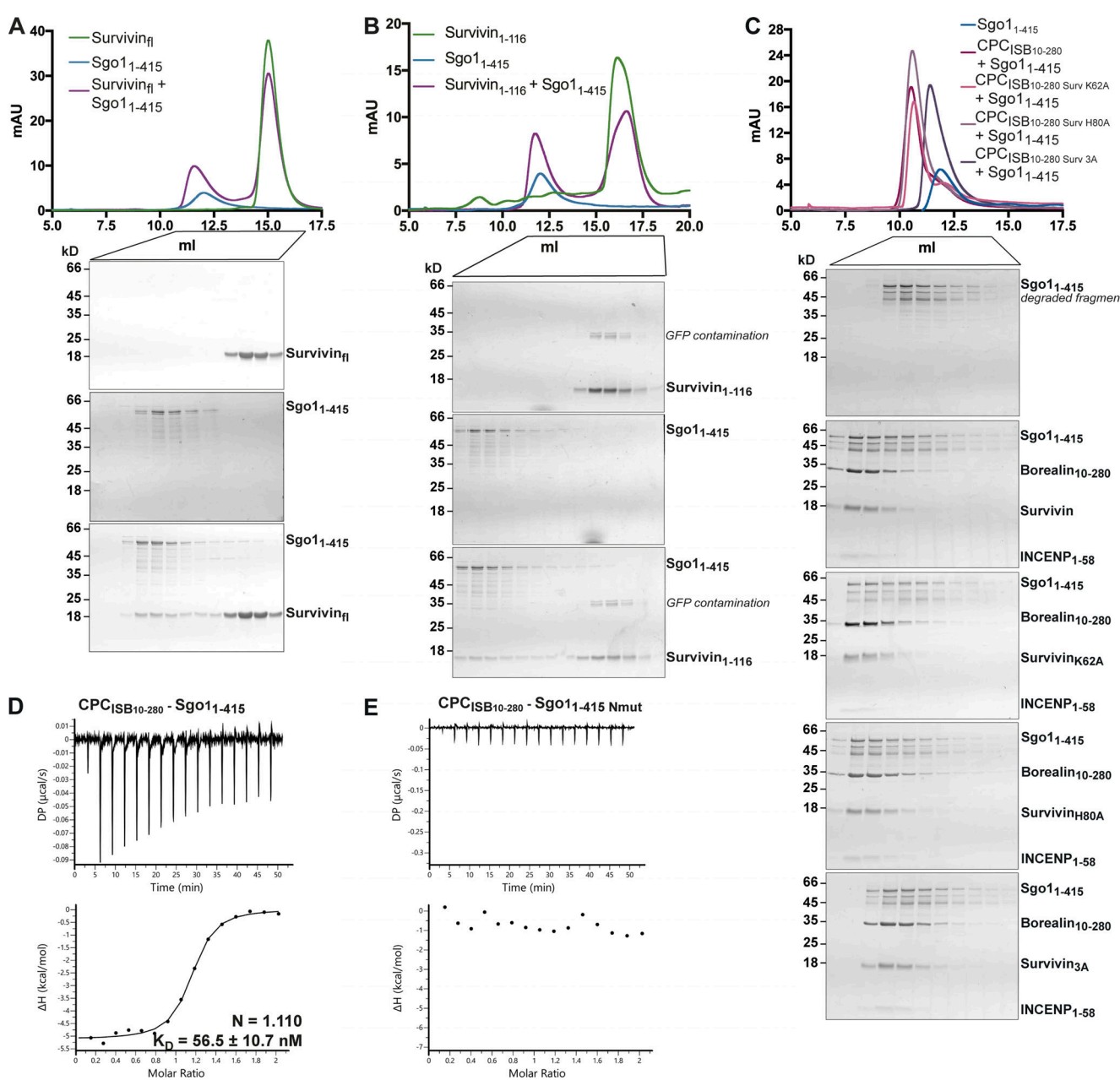

Figure S2. **Sgo1–Survivin interaction is essential for CPC–Sgo1 complex formation. (A–C)** SEC profiles (top) and corresponding representative SDS-PAGEs stained with Coomassie (bottom) for the analysis of Survivin and Sgo1$_{1-415}$ interaction (A), Survivin$_{1-116}$ and Sgo1$_{1-415}$ interaction (B), and CPC$_{ISB10-280}$ containing different Survivin mutants and Sgo1$_{1-415}$ interaction (C). For easy direct comparison, control SDS-PAGE and chromatogram corresponding to Sgo1$_{1-415}$ are shown in Figs. S2 A and S4 C. A Superdex S200 10/300 GL (Cytiva) column pre-equilibrated with 25 mM Hepes, pH 8, 250 mM NaCl, 5% glycerol, and 2 mM DTT was used. **(D and E)** Isotherms for the analyses of CPC$_{ISB\ 10-280}$ interaction with Sgo1$_{1-415}$ (D) and CPC$_{ISB\ 10-280}$ interaction with Sgo1$_{1-415\ Nmut}$ (E). 40 μl of 50 μM CPC$_{ISB\ 10-280}$ was injected into 200 μl of 5 μM Sgo1$_{1-415}$ or Sgo1$_{1-415\ Nmut}$. The ITC experiments were performed with 16 × 2.5-μl injections (0.5 μl first injection), 180 s apart, at 20°C. Top panels show raw ITC data; bottom panels show integrated heat data corrected for heat of dilution and fitted to a standard 1:1 binding model (Malvern Instruments MicroCal Origin software, v1.3). DP, differential power; mAU, milli absorbance units. Source data are available for this figure: SourceData FS2.

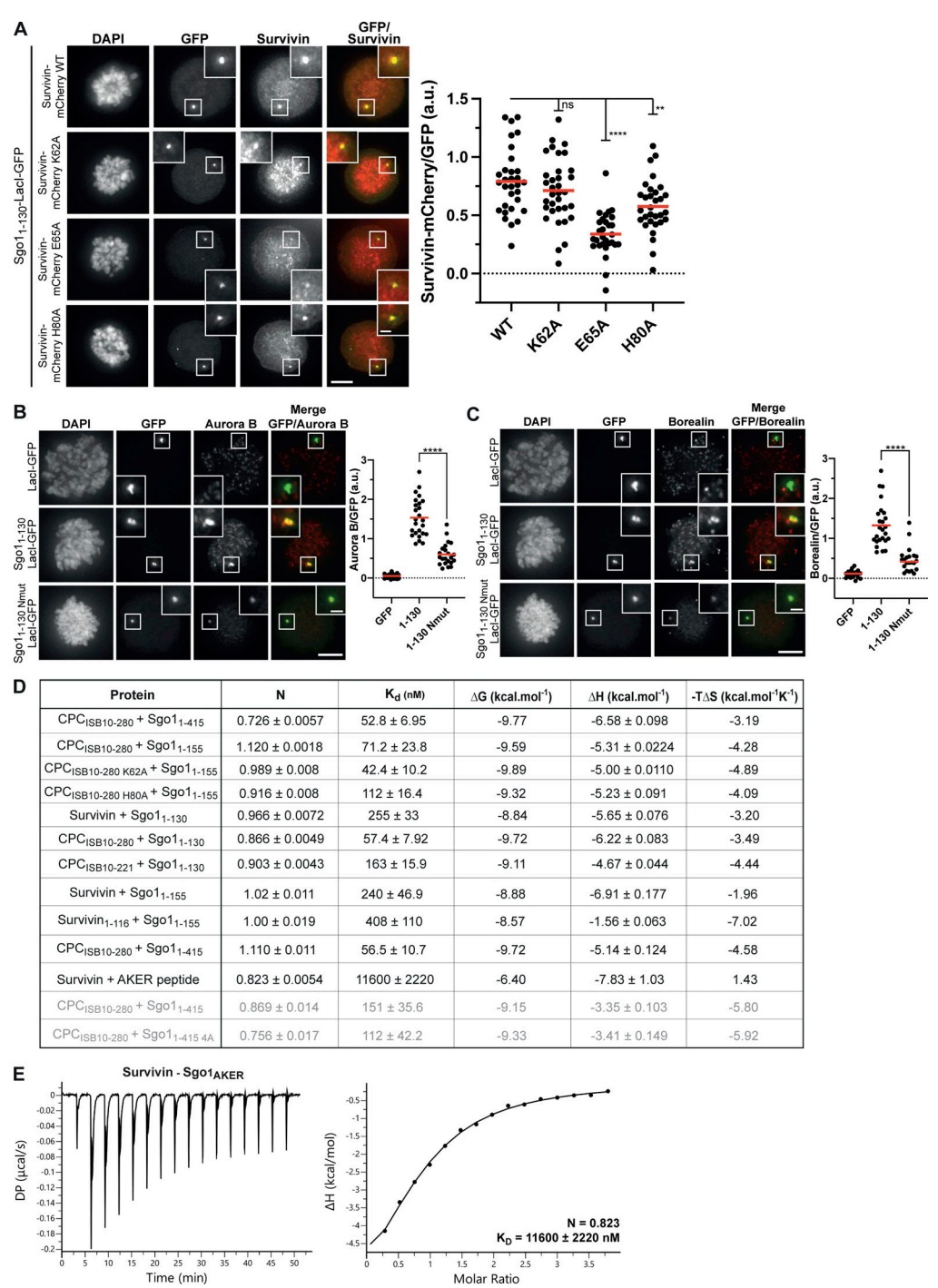

**D**

| Protein | N | $K_d$ (nM) | $\Delta G$ (kcal.mol⁻¹) | $\Delta H$ (kcal.mol⁻¹) | $-T\Delta S$ (kcal.mol⁻¹K⁻¹) |
|---|---|---|---|---|---|
| CPC$_{ISB10-280}$ + Sgo1$_{1-415}$ | 0.726 ± 0.0057 | 52.8 ± 6.95 | -9.77 | -6.58 ± 0.098 | -3.19 |
| CPC$_{ISB10-280}$ + Sgo1$_{1-155}$ | 1.120 ± 0.0018 | 71.2 ± 23.8 | -9.59 | -5.31 ± 0.0224 | -4.28 |
| CPC$_{ISB10-280}$ K62A + Sgo1$_{1-155}$ | 0.989 ± 0.008 | 42.4 ± 10.2 | -9.89 | -5.00 ± 0.0110 | -4.89 |
| CPC$_{ISB10-280}$ H80A + Sgo1$_{1-155}$ | 0.916 ± 0.008 | 112 ± 16.4 | -9.32 | -5.23 ± 0.091 | -4.09 |
| Survivin + Sgo1$_{1-130}$ | 0.966 ± 0.0072 | 255 ± 33 | -8.84 | -5.65 ± 0.076 | -3.20 |
| CPC$_{ISB10-280}$ + Sgo1$_{1-130}$ | 0.866 ± 0.0049 | 57.4 ± 7.92 | -9.72 | -6.22 ± 0.083 | -3.49 |
| CPC$_{ISB10-221}$ + Sgo1$_{1-130}$ | 0.903 ± 0.0043 | 163 ± 15.9 | -9.11 | -4.67 ± 0.044 | -4.44 |
| Survivin + Sgo1$_{1-155}$ | 1.02 ± 0.011 | 240 ± 46.9 | -8.88 | -6.91 ± 0.177 | -1.96 |
| Survivin$_{1-116}$ + Sgo1$_{1-155}$ | 1.00 ± 0.019 | 408 ± 110 | -8.57 | -1.56 ± 0.063 | -7.02 |
| CPC$_{ISB10-280}$ + Sgo1$_{1-415}$ | 1.110 ± 0.011 | 56.5 ± 10.7 | -9.72 | -5.14 ± 0.124 | -4.58 |
| Survivin + AKER peptide | 0.823 ± 0.0054 | 11600 ± 2220 | -6.40 | -7.83 ± 1.03 | 1.43 |
| CPC$_{ISB10-280}$ + Sgo1$_{1-415}$ | 0.869 ± 0.014 | 151 ± 35.6 | -9.15 | -3.35 ± 0.103 | -5.80 |
| CPC$_{ISB10-280}$ + Sgo1$_{1-415\,4A}$ | 0.756 ± 0.017 | 112 ± 42.2 | -9.33 | -3.41 ± 0.149 | -5.92 |

Figure S3. **Sgo1 N-terminal tail is crucial for CPC–Sgo1 interaction. (A)** Representative micrographs (left) and quantifications (right) for the analysis of Survivin-mCherry WT (*n* = 31), K62A (*n* = 34), E65A (*n* = 29), or H80A (*n* = 31) recruitment to the LacO array in U-2 OS-LacO Haspin CM cells expressing Sgo1$_{1–130}$-LacI-GFP. One-way ANOVA with Dunnett's multiple comparison test (****, P < 0.0001; **, P = 0.0026). Scale bar, 5 μm. **(B and C)** Representative micrographs (left) and quantifications (right) for the analysis of endogenous Aurora B (B) and Borealin (C) recruitment to the LacO array in U-2 OS-LacO Haspin CM expressing different Sgo1-LacI-GFP constructs: LacI-GFP (*n* = 21 for Aurora B; *n* = 21 for Borealin), Sgo1$_{1–130}$-LacI-GFP (*n* = 25 for Aurora B; *n* = 25 for Borealin), or Sgo1$_{1–130\ Nmut}$-LacI-GFP (*n* = 22 for Aurora B; *n* = 22 for Borealin). Representative immunofluorescence images in B and C show Aurora B and Borealin signal for the same cell; thus, DAPI and GFP in B and C are the same. The graphs show the intensities of Survivin-mCherry, Aurora B, or Borealin over GFP (dots) and the means (red bar). Data are representative of two or five biological replicates. Scale bar, 5 μm. One-way ANOVA with Dunnett's multiple comparison test (****, P < 0.0001). **(D)** Table including the ITC thermodynamic parameters for the different ITC experiments. All ITC experiments were performed using a buffer composed of 50 mM Hepes, 150 mM NaCl, 5% glycerol, and 1 mM TCEP, pH 8, except the two runs that are shown in light gray that were performed using a buffer composed of 50 mM Hepes, 250 mM NaCl, 5% glycerol, 0.005% Tween, and 1 mM TCEP, pH 8. **(E)** Isotherms for the analyses of Survivin interaction with Sgo1$_{AKER}$ peptide (40 μl of 375 μM Survivin was injected into 200 μl of 20 μM Sgo1$_{AKER}$). The ITC experiment was performed with 16 × 2.5-μl injections (0.5 μl first injection), 180 s apart, at 10°C. Left panel shows raw ITC data; right panel shows integrated heat data corrected for heat of ligand dilution and fitted to a standard 1:1 binding model (Malvern Instruments MicroCal Origin software, v1.3). DP, differential power.

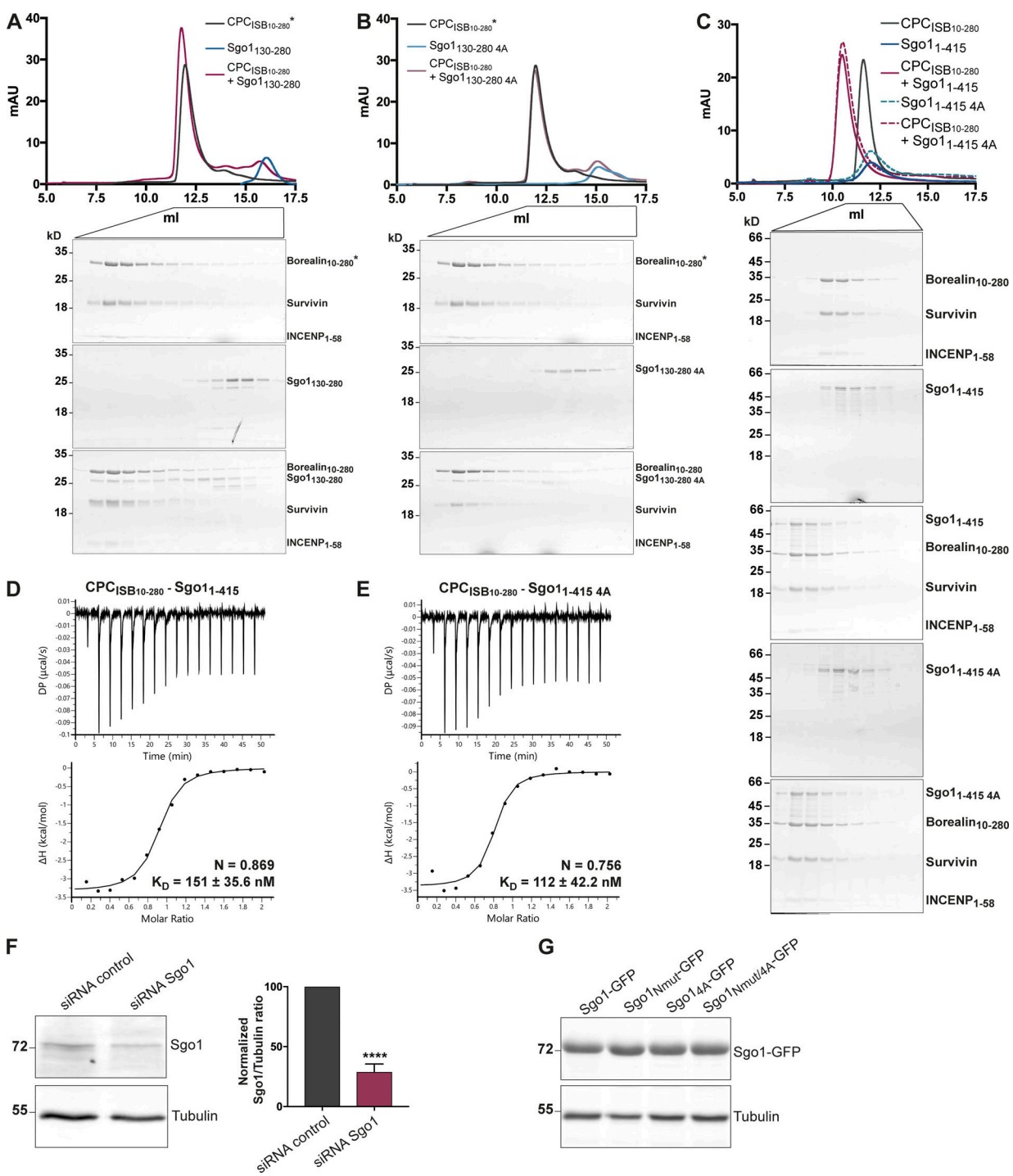

Figure S4. **Sgo1 central region does not significantly contribute to CPC binding in vitro. (A–C)** SEC profiles (top) and corresponding representative SDS-PAGEs stained with Coomassie (bottom) for the analysis of $CPC_{ISB10-280}$ and $Sgo1_{130-280}$ interaction (A), $CPC_{ISB10-280}$ and $Sgo1_{130-280\ 4A}$ interaction (B), and $CPC_{ISB10-280}$ and $Sgo1_{1-415}$ or $Sgo1_{1-415\ 4A}$ interaction (C). For easy direct comparison, control SDS-PAGE and chromatogram corresponding to $Sgo1_{1-415}$ are shown in Figs. S2 A and S4 C. A Superdex S200 10/300 GL (Cytiva) column pre-equilibrated with 25 mM Hepes, pH 8, 150 mM NaCl, 5% glycerol, and 2 mM DTT was used for A and B and 25 mM Hepes, pH 8, 250 mM NaCl, 5% glycerol, and 2 mM DTT was used for C. For easy comparison, control SDS-PAGEs and chromatograms corresponding to $CPC_{ISB\ 10-280}$ (marked with an asterisk) are shown in two different panels, A and B. **(D and E)** Isotherms for the analyses of $CPC_{ISB\ 10-280}$ interaction with $Sgo1_{1-415}$ (D) and $CPC_{ISB\ 10-280}$ interaction with $Sgo1_{1-415\ 4A}$ (E). 40 µl of 100 µM $CPC_{ISB\ 10-280}$ was injected into 200 µl of 10 µM $Sgo1_{1-415}$ or $Sgo1_{1-415\ 4A}$. The ITC experiments were performed with 16 × 2.5-µl injections (0.5 µl first injection), 180 s apart, at 20°C. Top panels show raw ITC data; bottom panels show integrated heat data corrected for heat of dilution and fitted to a standard 1:1 binding model (Malvern Instruments MicroCal Origin software, v1.3). **(F)** Representative immunoblot for the analysis of Sgo1 levels upon Sgo1 depletion using siRNA oligonucleotides. Quantification of Sgo1/Tubulin ratio using uncalibrated OD values (normalized Sgo1/Tubulin ratio for siRNA Sgo1 is 28.8 ± 6.6). Three independent experiments; mean ± SD; unpaired two-sided *t* test; ****, P ≤ 0.0001. **(G)** Representative immunoblot of Sgo1-GFP constructs (Sgo1-GFP, $Sgo1_{Nmut}$-GFP, $Sgo1_{4A}$-GFP, or $Sgo1_{Nmut/4A}$-GFP) showing comparable expression levels. DP, differential power; mAU, milli absorbance units. Source data are available for this figure: SourceData FS4.

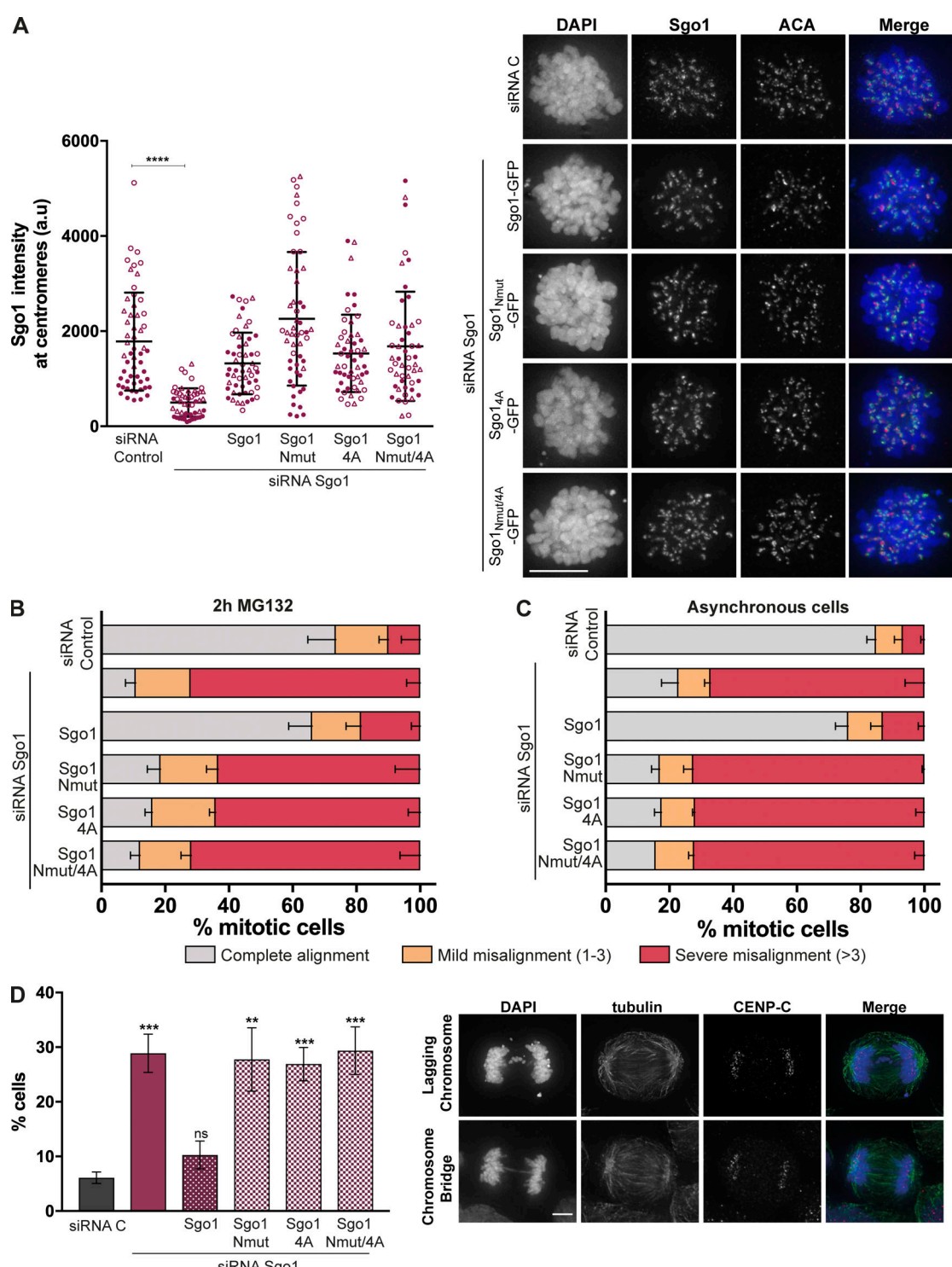

Figure S5. **CPC interaction with Sgo1 N-terminal tail is crucial for accurate chromosome segregation. (A)** Quantification of Sgo1 intensity at centromeres (left) and representative micrographs showing the localization of the transiently expressed Sgo1 mutant in comparison to the endogenous Sgo1 localization (right). Three independent experiments, $n \geq 50$ cells analyzed in total per treatment, mean ± SD, Kruskal–Wallis with Dunn's multiple comparisons test; ****, $P \leq 0.0001$. The values from the three independent replicates are represented in three different symbols. Scale bar, 10 µm. **(B and C)** Quantification of chromosome alignment of cells subjected to biorientation assay. Transfected cells were treated with 100 µM Monastrol for 16 h and released into a medium containing 5 µM MG132 for 2 h (B) or left as unperturbed asynchronous cultures (C). Observed metaphases were classified as complete alignment, mild misalignment (with one to three misaligned chromosomes), and severe misalignment (with more than three misaligned chromosomes). Three independent experiments, $n \geq 100$ of metaphases analyzed; mean ± SD. **(D)** Quantification of anaphase cells with lagging chromosomes or chromosome bridges for the siRNA-rescue assay of the Sgo1-GFP constructs: Sgo1-GFP, Sgo1$_{Nmut-GFP}$, Sgo1$_{4A-GFP}$, or Sgo1$_{Nmut/4A-GFP}$. Right: Representative examples of lagging chromosomes and chromosome bridges quantified. Three independent experiments, $n \geq 300$ of anaphases analyzed; mean ± SD; $\chi^2$ test for differences between the indicated groups and the control, for % complete alignment; **, $P \leq 0.01$; ***, $P \leq 0.001$). Scale bar, 10 µm.

