## [Peer Review File · The Journal of Cell Biology]

Mechanistic Basis for Sgo1-Mediated Centromere Localisation and Function of the CPC

Maria Alba Abad, Tanmay Gupta, Michael Hadders, Amanda Meppelink, Pepijn Wopken, Elizabeth Blackburn, Juan Zou, Anjitha Gireesh, Lana Buzuk, David Kelly, Toni McHugh, Juri Rappsilber, Susanne Lens, and A. Arockia Jeyaprakash

Corresponding Author(s): A. Arockia Jeyaprakash, University of Edinburgh

Review Timeline:

Submission Date:	2021-08-30
Editorial Decision:	2021-10-08
Revision Received:	2022-04-08
Editorial Decision:	2022-05-19
Revision Received:	2022-05-28

Monitoring Editor: Arshad Desai

Scientific Editor: Lucia Morgado-Palacin

Transaction Report:

DOI: <https://doi.org/10.1083/jcb.202108156>

October 8, 2021

Re: JCB manuscript #202108156

Dr. A. Arockia Jeyaprakash
University of Edinburgh
Wellcome Trust Centre for Cell Biology
Max Born Crescent
Edinburgh EH9 3BF
United Kingdom

Dear Dr. Jeyaprakash,

Thank you for submitting your manuscript entitled "Molecular Basis for CPC-Sgo1 Interaction: Implications for CPC Centromere Localisation and Function of the CPC". The manuscript was assessed by three expert reviewers, whose comments are appended to this letter. Based on their feedback, we are interested in receiving a thorough revision that addresses their specific and constructive comments on the work.

All three reviewers appreciated the rigorous biochemical reconstitutions, complemented by the *in vivo* analysis, to address the mechanism of CPC-Sgo1 interaction. While the reviewers find your analysis compelling, they raise a number of points that will need to be addressed in your revision. Specifically, reviewers have a number of concerns about the analysis of the central unstructured region of Sgo1 and its contribution to the binding affinity between Sgo1 and Survivin (major comments from Rev 1 & Rev 2). The reviewers suggest experiments to strengthen analysis of this region to the interaction *in vitro*, which would also help clarify apparent inconsistencies between *in vitro* binding affinity and *in vivo* tethering analysis. This is a central point to focus on in your revision, given that prior work had already established interaction of the histone-like N-terminus of Sgo1 with Survivin. A second major point relates to the *in vivo* significance of the interactions described (see point 5 from Rev 3). Our sense is that your efforts on the first major focus of the revision will help improve this aspect of the manuscript. Finally, while the binding mode of the histone-like sequence of Sgo1 is similar to that of the histone H3 tail, reviewers urge caution on emphasizing mutual exclusivity as no direct experimental tests, such as competitive binding, are performed. In addition to these issues, reviewers have provided numerous detailed and constructive comments that you will need to address. Please note that not all experimental requests must be met and it is acceptable to address specific points through appropriate text revisions and discussion. If discussion of a revision plan would be helpful to you and your colleagues, please reach out to the journal office.

GENERAL GUIDELINES:

Text limits: Character count for an Article is < 40,000, not including spaces. Count includes title page, abstract, introduction, results, discussion, acknowledgments, and figure legends. Count does not include materials and methods, references, tables, or supplemental legends.

Figures: Articles may have up to 10 main text figures. Figures must be prepared according to the policies outlined in our Instructions to Authors, under Data Presentation, <https://jcb.rupress.org/site/misc/ifora.xhtml>. All figures in accepted manuscripts will be screened prior to publication.

*****IMPORTANT:** It is JCB policy that if requested, original data images must be made available. Failure to provide original images upon request will result in unavoidable delays in publication. Please ensure that you have access to all original microscopy and blot data images before submitting your revision.

*****Please note that JCB now requires authors to submit Source Data used to generate figures containing gels and Western blots with all revised manuscripts. This Source Data consists of fully uncropped and unprocessed images for each gel/blot displayed in the main and supplemental figures. Since your paper includes cropped gel and/or blot images, please be sure to provide one Source Data file for each figure that contains gels and/or blots along with your revised manuscript files. File names for Source Data figures should be alphanumeric without any spaces or special characters (i.e., SourceDataF#, where F# refers to the associated main figure number or SourceDataFS# for those associated with Supplementary figures). The lanes of the gels/blots should be labeled as they are in the associated figure, the place where cropping was applied should be marked (with a box), and molecular weight/size standards should be labeled wherever possible. Source Data files will be made available to reviewers during evaluation of revised manuscripts and, if your paper is eventually published in JCB, the files will be directly linked to specific figures in the published article.**

Supplemental information: There are strict limits on the allowable amount of supplemental data. Articles may have up to 5 supplemental figures. Up to 10 supplemental videos or flash animations are allowed. A summary of all supplemental material should appear at the end of the Materials and methods section.

As you may know, the typical timeframe for revisions is three to four months. However, we at JCB realize that the implementation of social distancing and shelter in place measures that limit spread of COVID-19 also pose challenges to scientific researchers. Lab closures especially are preventing scientists from conducting experiments to further their research. Therefore, JCB has waived the revision time limit. We recommend that you reach out to the editors once your lab has reopened to decide on an appropriate time frame for resubmission. Please note that papers are generally considered through only one revision cycle, so any revised manuscript will likely be either accepted or rejected.

Thank you for this interesting contribution to Journal of Cell Biology. You can contact us at the journal office with any questions, cellbio@rockefeller.edu.

Sincerely,

Arshad Desai
Monitoring Editor
Journal of Cell Biology

Lucia Morgado-Palacin, PhD
Scientific Editor
Journal of Cell Biology

Reviewer #1 (Comments to the Authors (Required)):

The chromosomal passenger complex (CPC), comprising Survivin, Borealin, INCENP, and its catalytic subunit, the Aurora B kinase, plays key roles in facilitating the establishment of proper bi-orientation. The CPC is present in distinct pools at the inner centromere as well near kinetochores. Recruitment of the CPC to these sites has been proposed to occur through recognition of phosphorylated histone H3T3 (catalyzed by Haspin) by the BIR domain of Survivin, and through interaction of the CPC with Sgo1 (whose localization depends on H2AT120 phosphorylation by Bub1). Although a recent work reported interaction between the dimerization domain of Borealin to the coiled-coil region of Sgo1, it remains largely unknown how CPC interacts with Sgo1.

In this study, Abad et al. report another CPC recruitment pathway involving Sgo1 and Survivin using a combination of biochemical and cell biological approaches. They demonstrate that the Sgo1 forms a complex with CPCISB10-280 in vitro and that the N-terminal tail of Sgo1 is essential for this interaction. The authors corroborate these findings in vivo using both LacI-LacO tethering and rescue experiments. Moreover, the authors investigate another potential Survivin-Sgo1 interaction, identified by CLMS analysis, involving an unstructured central region of Sgo1. Overall, the data on the Sgo1 N-terminal tail interaction with Survivin is robust and interesting. However, I feel that this manuscript has several issues in its current form.

Major comments:

- I'm not convinced that Sgo1 central unstructured region (aa130-280) interacts with CPCISB10-280: 1) As the authors point out, Sgo11-415 Nmut clearly fails to interact with CPCISB10-280 despite having an intact central region. 2) Experiments in Figure S3A/B lack proper controls. The authors need to show the gel filtration profile and SDS-PAGE gel for the CPC ISB10-280 sample as well as an SDS-PAGE gel for Sgo1 130-280 for 11-12.5ml region in Figure S3A. I do not understand why the peak heights drop substantially in the CPCISB10-280 + Sgo1130-280 compared to the Sgo1130-280 solo run. Furthermore, to assess the contribution of the central region to CPC binding in vitro, the authors should test the interaction between Sgo11-415 4A with CPCISB10-280 and compare the binding affinity with wild-type Sgo11-415 (Figure 1C).

- Because both Borealin (via its dimerization domain) and Survivin (via its BIR domain) interact with Sgo1, I wonder whether these interactions act redundantly or synergistically in mediating CPC recruitment. For example, it would be interesting to test the interaction between full-length Sgo1 and CPCISB containing Survivin or Borealin single and double mutants. If not addressed experimentally, these questions should at least be discussed.

- The top gel images (CPC ISB 10-280) in Figure 3D and 3E look identical to me (with different magnification), while the gel filtration profiles are different. Can the authors provide an explanation for this?

Minor comments:

- Introduction could be improved in several places. For example, 'CPC function relies on its ability to localise correctly during mitosis': there is evidence that centromere localization of CPC is not essential at least in some organism/context, which could be mentioned to provide a balanced view. 'While CPC-Sgo1 functional interdependency is well established': this point was not very clear to me from reading the introduction.

- Results section 1 line 3: 'However, the molecular basis of this interaction has not been characterized'. This is too strong a claim in my opinion. Bonner et al. 2020 performed biochemical characterization of the CPC-Sgo1 interaction (Borealin-Sgo1). Perhaps the authors could be more accurate in describing the premise of the study (which is to investigate the direct interaction between Sgo1 and Survivin).

- Section 2: Has the Survivin 3A BIR mutant been characterized previously? If yes, the authors should reference that study. If no, they should either demonstrate experimentally that this mutant is not capable of interacting with the histone H3 tail or mention this point in the manuscript.

- Section 3 paragraph 3: 'Consistent with our data (Fig. 1B), the shorter Sgo11-155 fragment can also form a complex with CPCISB10-280 in vitro.' I can see how this result is consistent with the CLMS data and associated Lacl/LacO tethering experiments shown in Figure 2. Perhaps cite Fig. 2 rather than Fig. 1B?

- Results section 4 line 5: '...Sgo1 N-terminal tail peptide bound Survivin with ~1 μ M' should be ~1 μ M.

- Results section 5 line 3: Why did the authors choose to use a 1.5x molar excess of Sgo1130-280? In general, it is not mentioned anywhere in the manuscript what concentrations of recombinant protein were used for their SEC based interaction assays. Apart for the construct mentioned above, were equimolar concentrations used? Also, for how long and at which temperature were proteins mixed before running on the SEC column? These details need to be included in the revised manuscript.

- Figure 5A: I wonder how comparable the individual replicates are to each other. Presumably, the data from the three replicates were pooled to generate figure 5A? I am also curious about whether the mutant Sgo1 constructs localize efficiently to centromeres (the assumption is that centromeric enrichment of the different mutants are comparable). It is not clear to judge it from the images shown in Figure 5A.

- Figure 5C: Can the authors show more example images?

- I can see that AKER in the N-terminus of Sgo1 is not very well conserved (Figure S1A). Can the authors mention this in the text?

- I would suggest labelling the histograms in Figure S1B-D (i.e. CPCISB10-280, Sgo11-415, CPCISB10-280 + Sgo11-415).

- Figure S3C: The quantification suggests that Sgo1 levels are decreased more than two-fold upon RNAi induction. However, by looking at the Western Blot, it appears that a maximum of ~30% reduction is achieved. The figure legend only mentions that 'uncalibrated OD values' were used. Could the authors provide more details on how this was quantified in the Methods section?

- Quantification of Lacl/LacO tethering experiments (Figure 2C-D, 4D-E, S2A-B): Statistics are missing. Also, how do the replicates compare with one another?

- Figure 3: Sgo11-155 is marked with an asterisk in 3A, 3B, 3D, 3E and labelled hSgo11-155 in 3C. What is the purpose of this?

3B: Sgo11-155 Nmut shows a larger void peak compared to the Sgo11-155. Is this construct less stable or does it tend to aggregate more? 3D-F: In some gels, Borealin is labelled with '10-end' instead of '10-280'.

- Figure 4: In 4D, the red line (showing mean) is not visible in the '130-280 4A' condition. 4E: There seems to be a discrepancy between Aurora B and Borealin in terms of their recruitment to Sgo1 1-527 Nmut vs. Sgo1 1-527 4A. The data on Borealin recruitment (lower graph) fits better with the in vitro interaction assays. Is this discrepancy also represented in the other replicates?

- ITC experiments: Any replicates?

Reviewer #2 (Comments to the Authors (Required)):

The Chromosome Passenger Complex (CPC) and Shugoshin (Sgo1) play important roles for accurate chromosome segregation, which include the sister chromatid cohesion, the error collection of kinetochore-microtubule attachments, and the spindle assembly checkpoint. Components of CPC and Sgo1 function in the centromere region, and their localization mechanisms are characterized in cell biological methods. These studies suggested that CPC binds to Sgo1. However, it is not entirely clear about molecular basis how CPC binds to Sgo1 and how CPC and Sgo1 localize to centromeres. In this paper, authors used elegant biochemistry to clarify molecular basis for the CPC-Sgo1 interaction. They showed that multiple domains are involved in this interaction and especially, the Sgo1 N-terminal tail binds to Survivin, a component of CPC. In addition, they

found that Sgo1 hydrophobic patch in the aa 188-191 region is also critical for the CPC-Sgo1 interaction. Based additional cell biology observations, authors demonstrates that N-terminal and aa 188-191 regions of Sgo1 are critical for CPC centromere localization and accurate chromosome segregation. As the sequence of N-terminal tail of Sgo1 is identical to that of histone H3 N-terminal tail, authors proposed that the Sgo1 mediates localization of a CPC pool to 'kinetochore proximal centromere' using H3 and Sgo1 competitive binding to CPC.

Overall, this is a nice piece of work in the kinetochore studies. While there are many studies using cell biological methods for characterization on CPC and Sgo1, biochemical analyses on CPC and Sgo1 were not well performed. From this viewpoint, authors in this paper used elegant biochemistry to show clear interaction of the Sgo1 N-terminal tail with the reconstituted CPC or recombinant Survivin. They also used cell biology to confirm their own biochemical findings. Therefore, this reviewer basically supports publication of this work in JCB. However, authors should address several concerns before final acceptance.

1. Concerning ITC data in Fig 1 and Fig 4, authors should explain in more detail. Their data indicate that Kds of Sgo1₁₋₁₃₀-CPC (57.4 nM) and Sgo1₁₋₄₁₅-CPC (52.8 nM) are similar. However, authors demonstrate that the Sgo1 aa 188-191 region largely contributes to CPC binding. Why does Sgo1₁₋₁₃₀ lacking the important middle region (including aa 188-191) bind to CPC well? Authors should mention about it.
2. Author demonstrate that the Sgo1 4A mutant does not properly bind to CPC in Figure 3 SEC analyses. However, Sgo1₁₃₀₋₂₈₀ recruits Aurora B and Borealin using LacO-LacI experiments, suggesting that Sgo1₁₃₀₋₂₈₀ binds to CPC without the Sgo1 N-terminal region. Please explain these data in more detail, in the revised text.
3. SEC data in Figure 3 are really nice. However, we are curious how Kd values by ITC analysis of Sgo1_{Nmut}-CPC (or Survivin) is changed. Some ITC analyses using Sgo1_{Nmut} and 4A mutants should be performed. These will improve total quality of this paper.
4. In Figure 5C data, authors showed that kinetochore pool of CPC reduced in cells expressing Sgo1_{Nmut} or Sgo1_{4A} mutants. Using these data, authors mentioned that "the interaction of CPC with H2AT120ph-bound Sgo1 is responsible for the kinetochore proximal centromere pool of the CPC". But, I feel that Sgo1, which does not bind to H2AT120ph, should also reduce binding to Sgo1_{Nmut} or Sgo1_{4A} mutants. Why does not CPC signal in inner centromere change? Please explain this observation in the revised text.

Minor comments

1. In abstract authors described "Our findings provide evidence that CPC binding to Sgo1 and histone H3 N-terminal tail are mutually exclusive..." But, I think that they did not perform competitive binding assay etc, and there are no direct evidences for competitive binding of Sgo1 and H3 to CPC, Therefore, they should revise this sentence (They should be tone down for this description).
2. In p4, Knl1 form a subcomplex with Zwint1, and author should describe "Knl1 complex" instead of "Knl1".

Reviewer #3 (Comments to the Authors (Required)):

Aurora B-containing chromosomal passenger complex (CPC) plays a critical role in chromosome alignment in mitosis. The CPC predominantly localizes to centromeres during prometaphase and metaphase, which promotes the fidelity of mitotic chromosome segregation. Centromeric localization of the CPC requires binding to its Survivin subunit to the H3-T3ph mark generated by Haspin and the H2A-T120ph mark generated by Bub1. The H2A-T120ph mark promotes centromeric localization of the CPC partly via the interaction between its Borealin subunit and H2A-T120ph-bound shugoshin protein 1 (Sgo1). The molecular basis and functional significance for the Sgo1-CPC interaction remains unclear.

In this study, the authors used an integrative structure-function approach to show that the histone H3-like Sgo1 N-terminal tail interacts with Survivin acting as a hot-spot for CPC-Sgo1 assembly, while downstream Sgo1 residues bind Borealin to allow high affinity interaction. Disruption of the Sgo1 N-terminal tail-Survivin interaction abolished CPC-Sgo1 assembly in vitro and perturbed centromere localisation and function of CPC. This study provides evidence that CPC binding to Sgo1 and histone H3 N-terminal tail are mutually exclusive, as well as a rationale for the Sgo1-mediated kinetochore proximal centromere pool of CPC. This is an interesting manuscript that contains an impressive amount of work. The majority of the data are of high quality, and the manuscript is clearly written. However, my major concern is whether Sgo1 N-terminal tail containing the H3-ARpTK like AKER motif binds to Survivin BIR domain in cells, and whether this interaction is functionally important.

Major points:

1. The model for mutually exclusive binding of the CPC to Sgo1 and histone H3 N-terminal tail requires more robust test.
2. In Figure 1, it is worth testing whether recombinant proteins of Sgo1 (1-415) can bind Survivin and Survivin BIR domain in vitro?
3. In Figure 2, it is worth testing whether Sgo1 (1-130)-LacI-GFP can recruit exogenously expressed Survivin and Survivin BIR domain using the LacO-LacI tethering assay.
4. In Figure 3, it is worth examining whether the K62A or H80A mutant of recombinant Sgo1 (1-155) can bind recombinant Survivin or Survivin BIR domain in vitro? According to data shown in TABLE 1 of the Niedzialkowska E et al. study (MBoC, 2012), mutation of H80 to alanine abolished the detectable binding of Survivin to unphosphorylated H3 peptides as measured by

ITC. The crystal structure of Survivin bound to the Sgo1 peptide showed that Sgo1-Glu3 fits into the S3 pocket, with the carboxylate group interacting with K62 of Survivin (Jeyaprakash A et al., *Structure*, 2011). Niedzialkowska E et al. (*MBoC*, 2012) was unable to detect any binding of the Survivin-K62A mutant to H3T3ph-modified peptides using the structural and biochemical analysis.

5. Based on data shown Figures 3E, 3F, S2A and S2B, the authors claim that the Survivin-Sgo1 interaction is essential for the CPC binding to Sgo1 and that the Sgo1 N-terminal tail acts as a hot-spot, whose perturbation abolishes the ability of CPC to form a complex with Sgo1. Besides, based on data shown in Figures 4A-4C and S2C and S2D, the authors suggest that a weak micromolar affinity long-range electrostatic interaction between Survivin and the Sgo1 N-terminal tail is required to establish a high-affinity CPC-Sgo1 interaction mediated by multiple inter-protein contacts and hydrophobic effects. In the Liang C et al. study (*J Cell Biol*, 2020), a pool of Aurora B at the kinetochore-proximal centromere region was observed in mitotic HeLa cells in which Haspin-mediated H3-T3ph was chemically inhibited a small-molecule inhibitor (Figures 3B and 3C). Similar results were observed in mitotic HeLa cells in which endogenous Survivin was stably replaced with the H3-binding deficient mutant of Survivin-K62A or Survivin-H80A (Figure 3A). Moreover, when Bub1-mediated H2ApT120 was abolished by treatment with a chemical inhibitor BAY 1816032, Aurora B was no longer enriched at centromeres in Survivin-K62A or H80A mutant cells and Haspin inhibitor-treated HeLa cells (Figures 3A-3D). Thus, disrupting the H3-T3ph-Survivin interaction reveals a pool of Aurora B that is present at the KT-proximal centromere, presumably through interaction between H2A-T120ph-bound Sgo1 and the regulatory subunits of the CPC. Based on the literature (Tsukahara T et al., *Nature* 2010; Bonner M et al., *MBoC*, 2020), the H2ApT120-dependent localization of Aurora B at the KT-proximal centromere is presumably due to binding of Sgo1 to Borealin. If the K62A or H80A mutant of Survivin is incapable of binding Sgo1 in vitro, then the binding of Sgo1 N-terminal tail to Survivin BIR domain is not necessary for the localization of the CPC at the KT-proximal centromeres in cells.

6. In Figure 4, it is worth comparing the affinity for the binding of Survivin (wild-type or BIR domain alone) to Sgo1 and Sgo1-A1M (i.e., Nmut).

7. In Figures 5A, S3C and S3D, endogenous Sgo1 was depleted by siRNA in HeLa Kyoto cells expressing either wild type Sgo1 (Sgo1-GFP) or mutant Sgo1 (Sgo1Nmut-GFP, Sgo14A-GFP or Sgo1Nmut/4A double mutant) and centromeric levels of Borealin were analyzed. It is unclear whether these exogenous Sgo1-GFP proteins were stably expressed or transiently expressed, and whether endogenous Sgo1 was depleted to similar extent in cells expressing these Sgo1-GFP proteins.

8. In Figure S3E, siRNA-mediated knockdown of Sgo1 in HeLa Kyoto cells strongly increased the percentage of anaphase cells with lagging chromosomes. In general, Sgo1 knockdown results in premature loss of sister chromatid cohesion and spindle checkpoint activation, leading to mitotic arrest and subsequent cell death or mitotic slippage. I wonder how easy it was for the authors to find anaphase cells in which Sgo1 was depleted.

9. In Figure 5B, the authors showed that Sgo1-depleted HeLa cells expressing either Sgo1-GFP or mutant Sgo1 (Sgo1Nmut-GFP, Sgo14A-GFP or Sgo1Nmut/4-GFP) were released from a monastrol-induced mitotic arrest into medium with MG132 and chromosome alignment was assessed. Expression of Sgo1 mutants led to around 70% of cells with severe chromosome misalignment, comparable to the phenotype observed for Sgo1 depletion (Figure 5B). According to the Materials and Methods section, for the Monastrol assay, HeLa Kyoto cells were synchronised with 100 μ M Monastrol for 16 h and released into 5 μ M MG132 for 1 h. In general, after monastrol washout, HeLa cells took around 2 h to complete chromosome alignment. I therefore would recommend the authors to repeat this experiment and allow 2 h for cells to maximally complete chromosome alignment, then determine where cells expressing the Survivin mutants are intrinsically defective, or are only delayed, in chromosome alignment.

10. Based on data shown in Figure 5B, the authors conclude that the Survivin interaction with the Sgo1 N-terminal tail is essential for proper chromosome segregation. It is therefore worth examining chromosome alignment in unperturbed (i.e., without monastrol treatment and release) mitotic cells expressing these Survivin proteins (WT and mutants).

11. In Figure 5C, I would suggest to include IF pictures showing the localization of Borealin in control HeLa cells.

Minor points:

1. In Figure 5A legend, what does "Three independent experiments, n {greater than or equal to} 50" mean?

Dear Arshad,

Thank you for giving us the opportunity to answer the reviewer's comments. We are glad that all the reviewers appreciated the rigorousness and importance of our work aiming to understand the molecular basis for CPC-Sgo1 interaction.

Please find below the point-by-point response to the reviewer's comments.

Reviewer #1 (Comments to the Authors (Required)):

The chromosomal passenger complex (CPC), comprising Survivin, Borealin, INCENP, and its catalytic subunit, the Aurora B kinase, plays key roles in facilitating the establishment of proper bi-orientation. The CPC is present in distinct pools at the inner centromere as well near kinetochores. Recruitment of the CPC to these sites has been proposed to occur through recognition of phosphorylated histone H3T3 (catalyzed by Haspin) by the BIR domain of Survivin, and through interaction of the CPC with Sgo1 (whose localization depends on H2AT120 phosphorylation by Bub1). Although a recent work reported interaction between the dimerization domain of Borealin to the coiled-coil region of Sgo1, it remains largely unknown how CPC interacts with Sgo1.

In this study, Abad et al. report another CPC recruitment pathway involving Sgo1 and Survivin using a combination of biochemical and cell biological approaches. They demonstrate that the Sgo1 forms a complex with CPCISB10-280 *in vitro* and that the N-terminal tail of Sgo1 is essential for this interaction. The authors corroborate these findings *in vivo* using both LacI-LacO tethering and rescue experiments. Moreover, the authors investigate another potential Survivin-Sgo1 interaction, identified by CLMS analysis, involving an unstructured central region of Sgo1. Overall, the data on the Sgo1 N-terminal tail interaction with Survivin is robust and interesting. However, I feel that this manuscript has several issues in its current form.

Major comments:

- I'm not convinced that Sgo1 central unstructured region (aa130-280) interacts with CPCISB10-280: 1) As the authors point out, Sgo11-415 Nmut clearly fails to interact with CPCISB10-280 despite having an intact central region. 2) Experiments in Figure S3A/B lack proper controls. The authors need to show the gel filtration profile and SDS-PAGE gel for the CPC ISB10-280 sample as well as an SDS-PAGE gel for Sgo1 130-280 for 11-12.5ml region in Figure S3A. I do not understand why the peak heights drop substantially in the CPCISB10-280 + Sgo1130-280 compared to the Sgo1130-280 solo run. Furthermore, to assess the contribution of the central region

to CPC binding *in vitro*, the authors should test the interaction between Sgo1¹⁻⁴¹⁵ 4A with CPC_{ISB10-280} and compare the binding affinity with wild-type Sgo1¹⁻⁴¹⁵ (Figure 1C).

We acknowledge the concern raised by the reviewers. We have now repeated the Size Exclusion Chromatography (SEC) experiments (evaluating the interaction of CPC_{ISB10-280} with Sgo1₁₃₀₋₂₈₀) in Figure S3A/B (new Figure S4A/B), including the controls (individual SEC runs of Sgo1₁₃₀₋₂₈₀ and CPC_{ISB10-280} and SDS-PAGE gels showing the corresponding fractions). These experiments, in agreement with our original conclusion, suggest that Sgo1₁₃₀₋₂₈₀ has the ability to weakly associate with CPC_{ISB10-280}: Sgo1₁₃₀₋₂₈₀ co-elutes with CPC_{ISB10-280}, but with sub-stoichiometric band intensity as compared to that of CPC subunits. To assess the relative contribution of this interaction in the presence of the Sgo1₁₋₁₅₅ region, we performed ITC experiments and compared the CPC_{ISB10-280} binding affinities of Sgo1₁₋₄₁₅ and Sgo1₁₋₄₁₅ 4A. The corresponding K_D values are comparable: 151 ± 35.6 nM and 112 ± 42.2 nM, respectively, new Fig. S4D/E. In agreement with this, Sgo1₁₋₄₁₅ 4A forms a robust complex with CPC_{ISB10-280} in SEC (new Fig. S4C). However, the effects of mutating this region of Sgo1 in cells is substantial:

- 1) Reduced CPC recruitment when tethered onto LacO locus (Fig. 4E)
- 2) Reduced CPC recruitment to centromeres (siRNA/rescue) (Fig. 5A)
- 3) Incomplete rescue of chromosome alignment (siRNA/rescue) (Fig. 5B)

This suggests that the binding affinity of this region is most likely (strongly) enhanced by posttranslational modifications in mitotic cells, something that is not recapitulated in the *in vitro* assays. Exploring the post-translational regulation of this binding region is beyond the scope of this manuscript.

We have now explicitly discussed this on page 11 (line 24) and reads as follows: *Because the Sgo1 central region does not make a significant contribution to CPC binding in vitro, we consider it likely that this region in Sgo1 requires a yet unidentified post-translational modification/s to facilitate its interaction with the CPC, either in the Sgo1 region and/or in the CPC.*

- Because both Borealin (via its dimerization domain) and Survivin (via its BIR domain) interact with Sgo1, I wonder whether these interactions act redundantly or synergistically in mediating CPC recruitment. For example, it would be interesting to test the interaction between full-length Sgo1 and CPC_{ISB} containing Survivin or Borealin single and double mutants. If not addressed experimentally, these questions should at least be discussed.

We thank the reviewer for their suggestion. We have now tested the interaction of Sgo1₁₋₄₁₅ with CPC_{ISB10-280} containing the Survivin 3A mutations in the BIR domain (CPC_{ISB10-280} Surv 3A) by SEC. The CPC_{ISB10-280} Surv 3A severely disrupted Sgo1₁₋₄₁₅ binding in SEC (new Fig. S2C), suggesting the essential requirement of Sgo1 N-terminal tail binding to Survivin BIR domain. However, the CPC_{ISB} lacking the Borealin dimerization domain (CPC_{ISB10-221}) showed a 3-fold reduced affinity for Sgo1 (Fig. 4B

and Fig. 4C), highlighting the contribution of Borealin dimerization domain for enhanced affinity for Sgo1. This agrees with the original text on page 10.

- The top gel images (CPC_{ISB 10-280}) in Figure 3D and 3E look identical to me (with different magnification), while the gel filtration profiles are different. Can the authors provide an explanation for this?

We thank the reviewer for bringing this up and we apologise that we did not explain this in the original version. For easy direct comparison, the SEC profiles and corresponding SDS-PAGEs of control SEC runs are redundantly shown along with the mutant data that we compare with. For example, SEC profile and corresponding SDS-PAGE of CPC_{ISB10-280} are shown in both Fig. 3D and 3E. We would like to note that both, control experiments with wild type proteins and experiments with mutant proteins, were carried out at identical experimental conditions. We have now marked these gels and chromatograms with asterisks and explained this clearly in the figure legend of Fig. 3 as follows: *For easy direct comparison, control SDS-PAGEs and chromatograms corresponding to Sgo1₁₋₁₅₅, Sgo1₁₋₁₅₅ Nmut and CPC_{ISB 10-280} (marked with an asterisk) are shown in two different panels Fig. 3A and 3D, Fig. 3B and 3E, and Fig. 3D and 3E, respectively.*

Minor comments:

- Introduction could be improved in several places. For example, 'CPC function relies on its ability to localise correctly during mitosis': there is evidence that centromere localization of CPC is not essential at least in some organism/context, which could be mentioned to provide a balanced view. 'While CPC-Sgo1 functional interdependency is well established': this point was not very clear to me from reading the introduction.

We thank the reviewer for the suggestion. We agree with the reviewer that in some organisms centromeric localisation of CPC does not appear to be essential, so we have now removed that sentence from the introduction.

Regarding the second point raised, we have now clarified the statement as follows: *While Sgo1 is known to play a role in the recruitment of the CPC to centromeres, the structural and molecular basis for how the CPC and Sgo1 interact and how these interactions contribute to the localisation and function of the specific CPC pools remain unclear.* (page 5, line 6).

- Results section 1 line 3: 'However, the molecular basis of this interaction has not been characterized'. This is too strong a claim in my opinion. Bonner et al. 2020 performed biochemical characterization of the CPC-Sgo1 interaction (Borealin-Sgo1). Perhaps the authors could be more accurate in describing the premise of the study (which is to investigate the direct interaction between Sgo1 and Survivin).

We have now rephrased this sentence as follows: *However, detailed characterisation of how the various CPC subunits contribute to Sgo1 binding has not yet been performed.* (page 5, line 15).

- Section 2: Has the Survivin 3A BIR mutant been characterized previously? If yes, the authors should reference that study. If no, they should either demonstrate experimentally that this mutant is not capable of interacting with the histone H3 tail or mention this point in the manuscript.

Previous structural work from Jeyaprakash et al., 2011, Du et al., 2012 and Niedzialkowska et al., 2012 described the importance of the Survivin BIR residues K62, H80 and E65 for H3T3ph interaction. In addition, Niedzialkowska et al., 2012 showed that K62A, H80A and E65A mutants abolished Survivin's ability to bind H3T3ph peptide using ITC. We have now cited Niedzialkowska et al., 2012 in the sentence where we introduce the Survivin 3A mutant (p.9, line 2).

- Section 3 paragraph 3: 'Consistent with our data (Fig. 1B), the shorter Sgo11-155 fragment can also form a complex with CPCISB10-280 in vitro.' I can see how this result is consistent with the CLMS data and associated Lacl/LacO tethering experiments shown in Figure 2. Perhaps cite Fig. 2 rather than Fig. 1B?

We thank the reviewer for this. We have now cited the CLMS data appropriately on page 8 and line 11.

- Results section 4 line 5: '...Sgo1 N-terminal tail peptide bound Survivin with $\sim 1 \mu\text{M}$ ' should be $\sim 1 \mu\text{M}$.

We thank the reviewer for bringing this typo to our attention. We have now changed the $\sim 1 \mu\text{M}$ to $1 \mu\text{M}$.

- Results section 5 line 3: Why did the authors choose to use a 1.5x molar excess of Sgo1130-280? In general, it is not mentioned anywhere in the manuscript what concentrations of recombinant protein were used for their SEC based interaction assays. Apart for the construct mentioned above, were equimolar concentrations used? Also, for how long and at which temperature were proteins mixed before running on the SEC column? These details need to be included in the revised manuscript.

A 1.5-2x molar excess of Sgo1 was used in all the interaction experiments with CPC to ensure the formation of a stoichiometric CPC-Sgo1 complex. We have now described this in the materials and methods section: *A 1.5-2x molar excess of Sgo1 was used in all interaction studies with CPC. Proteins were mixed and incubated at 4 °C for 1 h before being injected to the size exclusion column.*

- Figure 5A: I wonder how comparable the individual replicates are to each other. Presumably, the data from the three replicates were pooled to generate figure 5A? I am also curious about whether the mutant Sgo1 constructs localize efficiently to centromeres (the assumption is that centromeric enrichment of the different mutants are comparable). It is not clear to judge it from the images shown in Figure 5A.

We confirm that the data shown in Figure 5A is a pool of three replicates. As you can see below the three individual replicates are consistent and comparable. We have now changed Fig. 5A for new Fig. 5A, where the values of the three independent replicates are represented using three different symbols.

Regarding the query on the centromere enrichment of different Sgo1 mutants: We confirm that the centromeric enrichment of mutants is comparable to that of endogenous Sgo1. We have now included a new figure (new Fig. S5A) where we have quantified the centromeric levels of the different Sgo1 constructs (WT, Nmut, 4A and Nmut/4A) compared to the endogenous Sgo1 levels and the siRNA Sgo1 condition (left panel). We also created a panel to show some more representative images on the localisation of the different Sgo1 constructs (right panel).

- Figure 5C: Can the authors show more example images?

We thank the reviewer for this suggestion. We have now prepared a new figure with more examples for Fig. 5C (please see below) for the purpose of this rebuttal. Unfortunately, we could not include this panel to the supplementary figures due to space constraints.

- I can see that AKER in the N-terminus of Sgo1 is not very well conserved (Figure S1A). Can the authors mention this in the text?

The N-terminal AKER motif, although not conserved in all the organisms analysed, is well conserved in most higher vertebrates. We have now included a sentence reflecting this in the figure legend for Fig. S1A. (page 22, line 1).

- I would suggest labelling the histograms in Figure S1B-D (i.e. CPCISB10-280, Sgo11-415, CPCISB10-280 + Sgo11-415).

We have now labelled Fig. S1B-D as suggested.

- Figure S3C: The quantification suggests that Sgo1 levels are decreased more than two-fold upon RNAi induction. However, by looking at the Western Blot, it appears that a maximum of ~30% reduction is achieved. The figure legend only mentions that 'uncalibrated OD values' were used. Could the authors provide more details on how this was quantified in the Methods section?

Our quantifications show a 70% reduction in Sgo1 levels (Normalised Sgo1/Tubulin ratio for siRNA Sgo1 is 28.8 ± 6.6). We have now added this value to the figure legend and expanded the materials and methods section to explain how the quantifications were performed: *Immunoblots were imaged using the Odyssey CLx system and band intensities were quantified using ImageJ, uncalibrated OD. Values were then corrected by the corresponding tubulin levels (loading control) and normalised to siRNA control values. Three experimental replicates were analysed.* (p. 35, line 4).

- Quantification of LacI/LacO tethering experiments (Figure 2C-D, 4D-E, S2A-B): Statistics are missing. Also, how do the replicates compare with one another?

We thank the reviewer for this comment. We have now included the statistics for Figs. 2C, 2D, 4D, 4E and new S3A, B and C that show the data for one of the replicates. We can confirm that that the replicates are consistent and comparable. Please find below the data for all the replicates. We couldn't add this to the supplementary figures of the paper due to space limitations.

C Replicate for Fig. S3A

D Replicates for Fig. S3B and S3C

- Figure 3: Sgo11-155 is marked with an asterisk in 3A, 3B, 3D, 3E and labelled hSgo11-155 in 3C. What is the purpose of this? 3B: Sgo11-155 Nmut shows a larger void peak compared to the Sgo11-155. Is this construct less stable or does it tend to aggregate more? 3D-F: In some gels, Borealin is labelled with '10-end' instead of '10-280'.

As outlined in our response to the third point of the major comments, the asterisks denote the SEC chromatogram and SDS-PAGE of control experiments that are redundantly used to directly compare the corresponding data for the mutant proteins. This is now explained in the figure legend of Figure 3 as follows: For easy direct comparison, control SDS-PAGEs and chromatograms corresponding to Sgo1₁₋₁₅₅, Sgo1_{1-155 Nmut} and CPC₁₀₋₂₈₀ (marked with an asterisk) are shown in two different panels Fig. 3A and 3D, Fig. 3B and 3E, and Fig. 3D and 3E, respectively.

The void we observe in some Sgo1 SEC runs is due to the DNA contamination that co-purifies with Sgo1. In recent purification efforts (for example in Sgo1₁₋₄₁₅ shown in Fig. 1B) we have overcome this issue by optimising the ion exchange conditions.

We thank the reviewer for noticing this. We have now consistently labelled Borealin lacking the first 9 amino acids as Borealin₁₀₋₂₈₀.

- Figure 4: In 4D, the red line (showing mean) is not visible in the '130-280 4A' condition. 4E: There seems to be a discrepancy between Aurora B and Borealin in terms of their recruitment to Sgo1 1-527 Nmut vs. Sgo1 1-527 4A. The data on Borealin recruitment (lower graph) fits better with the in vitro interaction assays. Is this discrepancy also represented in the other replicates?

We thank the reviewer for bringing this up to our notice. We have now corrected this and show the red line (mean) in the '130-280 4A' condition.

In Fig 4E we indeed observe somewhat lower levels of Borealin recruitment for Sgo1_{Nmut} compared to Sgo1_{4A}, in contrast to Aurora B where Nmut and 4A are more comparable. We mainly attribute this to experimental variation based off of several observations. First, it should be noted that these stainings are not of the same cells so it is unclear what the Aurora B levels are in the cells with more strongly reduced Borealin. Second, in the Borealin graph it can be seen that the spread of the Nmut cells is quite high, with a larger number of cells present in the lower population. For the Aurora B staining you can see a similar spread but with lower levels of the lower population. The reason behind the different distributions in these experiments is unclear but repeat experiments do not show this (See replicate experiments above). Importantly, the replicate experiments consistently show a (comparable) reduction of both Aurora B and Borealin recruitment to Sgo1_{Nmut} and Sgo1_{4A} and an even stronger reduction in the double mutant Sgo1_{Nmut/4A}.

- ITC experiments: Any replicates?

ITC has the advantage over many biochemical techniques in that it determines the stoichiometry (N) of the interaction. N is an extremely sensitive metric to assess reagent and data quality. This reduces the requirement for multiple replicates. In the data presented, a stoichiometry of 1 indicates pure, well characterised, and active components in the protein:protein complex. The K_D is much less sensitive to errors in concentration. In addition, we perform blocks of experiments where we have a common sample, for example Sgo1₁₋₁₅₅ in the ITC cell, and perform 5 experiments with different species as titrants. We see consistent stoichiometries of 1 in these experimental blocks. Thus, providing good confidence in comparing K_D between different complexes that share common components.

Where there was enough material or/and where the data quality was not satisfactory, experiments were repeated. All the replicates produced comparable K_D values, please see the table below for some examples.

Syringe	Cell	N (sites)	K _D (nM)	ΔG (kcal/mol)	ΔH (kcal/mol)	-TΔS (kcal/mol)	Figure	
CPC _{ISB10-280}	Sgo1 ₁₋₄₁₅	1.110 ± 0.011	56.5 ± 10.7	-9.72	-5.14 ± 0.124	-4.58	Fig S2D	
CPC _{ISB10-280}	Sgo1 ₁₋₄₁₅	0.726 ± 0.006	52.8 ± 6.95	-9.77	-6.58 ± 0.098	-3.19		
CPC _{ISB10-280}	Sgo1 ₁₋₁₅₅	1.120 ± 0.018	71.2 ± 23.8	-9.59	-5.31 ± 0.224	-4.28	Fig 3G	
CPC _{ISB10-280}	Sgo1 ₁₋₁₅₅	0.463 ± 0.0088	74.2 ± 24.1	-9.56	-3.61 ± 0.120	-5.95		
CPC _{ISB10-280 3A}	Sgo1 ₁₋₁₅₅	No binding						Fig3J
CPC _{ISB10-280 3A}	Sgo1 ₁₋₁₅₅	No binding						
Survivin	Sgo1 ₁₋₁₅₅	1.02 ± 0.011	240 ± 46.9	-8.88	-6.91 ± 0.177	-1.96	Fig S1G	
Survivin	Sgo1 ₁₋₁₅₅	0.715 ± 0.011	259 ± 57.3	-8.84	-7.26 ± 0.197	-1.58		
Survivin ₁₋₁₁₆	Sgo1 ₁₋₁₅₅	1.000 ± 0.019	408 ± 110	-8.57	-1.56 ± 0.063	-7.02	Fig S1I	
Survivin ₁₋₁₁₆	Sgo1 ₁₋₁₅₅	0.771 ± 0.024	219 ± 83.6	-8.93	-2.11 ± 0.111	-6.82		
Survivin ₁₋₁₁₆	Sgo1 _{1-155 A1M}	No binding						Fig S1J
Survivin ₁₋₁₁₆	Sgo1 _{1-155 A1M}	No binding						

Reviewer #2 (Comments to the Authors (Required)):

The Chromosome Passenger Complex (CPC) and Shugoshin (Sgo1) play important roles for accurate chromosome segregation, which include the sister chromatid cohesion, the error collection of kinetochore-microtubule attachments, and the spindle assembly checkpoint. Components of CPC and Sgo1 function in the centromere region, and their localization mechanisms are characterized in cell biological methods. These studies suggested that CPC binds to Sgo1. However, it is not entirely clear about molecular basis how CPC binds to Sgo1 and how CPC and Sgo1 localize to centromeres. In this paper, authors used elegant biochemistry to clarify molecular basis for the CPC-Sgo1 interaction. They showed that multiple domains are involved in this interaction and especially, the Sgo1 N-terminal tail binds to Survivin, a component of CPC. In addition, they found that Sgo1 hydrophobic patch in the aa 188-191 region is also critical for the CPC-Sgo1 interaction. Based additional cell biology observations, authors demonstrates that N-terminal and aa 188-191 regions of Sgo1 are critical for CPC centromere localization and accurate chromosome segregation. As the sequence of N-terminal tail of Sgo1 is identical to that of histone H3 N-terminal tail, authors proposed that the Sgo1 mediates localization of a CPC pool to 'kinetochore proximal centromere' using H3 and Sgo1 competitive binding to CPC.

Overall, this is a nice piece of work in the kinetochore studies. While there are many studies using cell biological methods for characterization on CPC and Sgo1, biochemical analyses on CPC and Sgo1 were not well performed. From this viewpoint, authors in this paper used elegant biochemistry to show clear interaction of the Sgo1 N-terminal tail with the reconstituted CPC or recombinant Survivin. They also used cell biology to confirm their own biochemical findings. Therefore, this reviewer basically supports publication of this work in JCB. However, authors should address several concerns before final acceptance.

1. Concerning ITC data in Fig 1 and Fig 4, authors should explain in more detail. Their data indicate that K_D s of Sgo1₁₋₁₃₀-CPC (57.4 nM) and Sgo1₁₋₄₁₅-CPC (52.8 nM) are similar. However, authors demonstrate that the Sgo1 aa 188-191 region largely contributes to CPC binding. Why does Sgo1₁₋₁₃₀ lacking the important middle region (including aa 188-191) bind to CPC well? Authors should mention about it.

We thank the reviewer for raising this concern. As discussed in our response to major point number 1 of Reviewer 1, we have now performed several additional ITC (Fig. S4D and S4E) and SEC (Fig. S4C) experiments with Sgo1₁₋₄₁₅ and Sgo1_{1-415 4A}. Although, Sgo1₁₃₀₋₂₈₀ shows weak ability to bind CPC in SEC, the K_D values measured for Sgo1₁₋₁₃₀, Sgo1₁₋₁₅₅ and Sgo1₁₋₄₁₅ to bind CPC_{ISB10-280} are near identical (57.4 ± 7.92 nM, 71.2 ± 23.8 nM and 52.8 ± 6.95 nM or 56.5 ± 10.7 nM, respectively; new Fig. S3D). Moreover, Sgo1_{1-415 4A} (harbouring mutations within the Sgo1 central region) binds CPC_{ISB10-280} with affinity similar to that of Sgo1₁₋₄₁₅.

Considering that the effects of mutating this region of Sgo1 in cells is substantial:

- 1) Reduced CPC recruitment when tethered onto LacO locus (Fig. 4E)
- 2) Reduced CPC recruitment to centromeres (siRNA/rescue) (Fig. 5A)
- 3) Incomplete rescue of chromosome alignment (siRNA/rescue) (Fig. 5B)

This suggests that the binding affinity of this region is most likely (strongly) enhanced by posttranslational modifications in mitotic cells, something that is not recapitulated in the *in vitro* assays. Exploring the post-translational regulation of this binding regions is beyond the scope of this manuscript.

We have now explicitly discussed this on page 11 (line 24) and reads as follows: *Because the Sgo1 central region does not make a significant contribution to CPC binding in vitro, we consider it likely that this region in Sgo1 requires a yet unidentified post-translational modification/s to facilitate its interaction with the CPC, either in the Sgo1 region and/or in the CPC.*

2. Author demonstrate that the Sgo1 4A mutant does not properly bind to CPC in Figure 3 SEC analyses. However, Sgo1₁₃₀₋₂₈₀ recruits Aurora B and Borealin using LacO-LacI experiments, suggesting that Sgo1₁₃₀₋₂₈₀ binds to CPC without the Sgo1 N-terminal region. Please explain these data in more detail, in the revised text.

We agree with the reviewer that in cells Sgo1 N-terminal region and Sgo1 central region appear to interact with CPC independently. However, as discussed above, contribution of the Sgo1 central region for CPC binding seems insignificant *in vitro*. It is possible that a post translational modification on CPC or/and Sgo1 might modulate CPC-Sgo1 interaction in a yet unidentified molecular pathway.

We have now explicitly discussed this on page 11 (line 24) and reads as follows: *Because the Sgo1 central region does not make a significant contribution to CPC binding in vitro, we consider it likely that this region in Sgo1 requires a yet unidentified post-*

translational modification/s to facilitate its interaction with the CPC, either in the Sgo1 region and/or in the CPC.

3. SEC data in Figure 3 are really nice. However, we are curious how Kd values by ITC analysis of Sgo_Nmut-CPC (or Survivin) is changed. Some ITC analyses using Sgo_Nmut and 4A mutants should be performed. These will improve total quality of this paper.

We thank the reviewer for this suggestion. We have now performed ITC experiments to measure the CPC_{ISB10-280} binding affinity of Sgo1₁₋₄₁₅, Sgo1_{1-415 Nmut} or Sgo1_{1-415 4A}. In agreement with the SEC analyses, Sgo1_{1-415 Nmut} did not show detectable CPC binding (new Fig. S2D and S2E), while Sgo1_{1-415 4A} bound CPC_{ISB 10-280} with affinity comparable to that of Sgo1_{1-415 WT} (151 ± 35.6 for WT vs 112 ± 42.2 for 4A mutant) (new Fig. S4D and S4E).

4. In Figure 5C data, authors showed that kinetochore pool of CPC reduced in cells expressing Sgo1_Nmut or Sgo1_4A mutants. Using these data, authors mentioned that "the interaction of CPC with H2AT120ph-bound Sgo1 is responsible for the kinetochore proximal centromere pool of the CPC". But, I feel that Sgo1, which does not bind to H2AT120ph, should also reduce binding to Sgo1_Nmut or Sgo1_4A mutants. Why does not CPC signal in inner centromere change? Please explain this observation in the revised text.

Our data suggest that the CPC-Sgo1 interaction mainly contributes to the kinetochore-proximal centromere localisation of the CPC and less to the inner centromere pool of the CPC. We appreciate that Sgo1 associates with the *inner centromere* via its interaction with cohesin and we deem it likely that cohesin associated Sgo1 does not interfere with the CPC's ability to interact with H3T3ph and hence does not perturb the inner centromere CPC pool. In addition, the levels of H3T3ph are likely to exceed the levels of Sgo1 at the inner centromere and hence the contribution of cohesin-associated Sgo1 to CPC inner centromere recruitment is predicted to be minor.

It is important to note that, to our knowledge, there is no direct evidence that Sgo1 is required specifically for recruitment of the CPC to the inner centromere. Based on the data presented here, we propose that Sgo1 forms the CPC receptor at kinetochore proximal centromeres while H3T3ph forms the main CPC receptor at the inner centromere. We now explicitly mention this in p.17, line 1.

Minor comments

1. In abstract authors described "Our findings provide evidence that CPC binding to Sgo1 and histone H3 N-terminal tail are mutually exclusive..." But, I think that they did not perform competitive binding assay etc, and there are no direct evidences for competitive binding of Sgo1 and H3 to CPC, Therefore, they should revise this sentence (They should be tone down for this description).

We agree with the reviewer. We have now revised the statement in the abstract as follows: *Our findings reveal that Sgo1 and H3T3ph use the same surface on Survivin to bind CPC. Hence, it is likely that these interactions take place in a spatio-temporal restricted manner, providing a rationale for the Sgo1-mediated 'kinetochore proximal' CPC centromere pool.*

We have also revised a related sentence p.17, line 9: *The observation that the Sgo1 and histone H3 N-terminal tails exploit the same binding site in Survivin, suggests that these interactions could be mutually exclusive and may explain why the Bub1-dependent CPC pool exists as a kinetochore-proximal centromere pool that is spatially distinct from the Haspin-dependent inner centromere CPC pool.*

2. In p4, Knl1 form a subcomplex with Zwint1, and author should describe "Knl1 complex" instead of "Knl1".

We thank the reviewer for the valid suggestion. We have now corrected this as follows: *The CPC, via its Aurora B enzymatic core, destabilises aberrant KT-MT attachments by phosphorylating outer kinetochore substrates such as the Knl1 complex/Mis12 complex/Ndc80 complex (KMN) network so that new attachments can be formed (Cheeseman et al., 2006; Cimini et al., 2006; DeLuca et al., 2006; Lampson et al., 2004; Welburn et al., 2010) (p. 4, line 6).*

Reviewer #3 (Comments to the Authors (Required)):

Aurora B-containing chromosomal passenger complex (CPC) plays a critical role in chromosome alignment in mitosis. The CPC predominantly localizes to centromeres during prometaphase and metaphase, which promotes the fidelity of mitotic chromosome segregation. Centromeric localization of the CPC requires binding to its Survivin subunit to the H3-T3ph mark generated by Haspin and the H2A-T120ph mark generated by Bub1. The H2A-T120ph mark promotes centromeric localization of the CPC partly via the interaction between its Borealin subunit and H2A-T120ph-bound shugoshin protein 1 (Sgo1). The molecular basis and functional significance for the Sgo1-CPC interaction remains unclear.

In this study, the authors used an integrative structure-function approach to show that the histone H3-like Sgo1 N-terminal tail interacts with Survivin acting as a hot-spot for CPC-Sgo1 assembly, while downstream Sgo1 residues bind Borealin to allow high affinity interaction. Disruption of the Sgo1 N-terminal tail-Survivin interaction abolished CPC-Sgo1 assembly in vitro and perturbed centromere localisation and function of CPC. This study provides evidence that CPC binding to Sgo1 and histone H3 N-terminal tail are mutually exclusive, as well as a rationale for the Sgo1-mediated kinetochore proximal centromere pool of CPC. This is an interesting manuscript that contains an impressive amount of work. The majority of the data are of high quality, and the manuscript is clearly written. However, my major concern is whether Sgo1 N-terminal tail containing the H3-ARpTK like AKER motif binds to

Survivin BIR domain in cells, and whether this interaction is functionally important.

Major points:

1. The model for mutually exclusive binding of the CPC to Sgo1 and histone H3 N-terminal tail requires more robust test.

We acknowledge the concern raised by this reviewer and agree that a careful characterisation of CPC binding to Sgo1 and Histone H3 N-terminal tail phosphorylated at Thr3 (or/and nucleosomes containing H3T3ph) is required to understand possible competitive binding. However, such an analysis is beyond the scope of the current manuscript. Hence, as discussed in our response to minor point 1 of Reviewer 2, we have now revised the statement in the abstract as follows: *Our findings reveal that Sgo1 and H3T3ph use the same surface on Survivin to bind CPC. Hence, it is likely that these interactions take place in a spatio-temporal restricted manner, providing a rationale for the Sgo1-mediated 'kinetochore proximal' CPC centromere pool.*

We have also revised a related sentence p.17, line 9: *The observation that the Sgo1 and histone H3 N-terminal tails exploit the same binding site in Survivin, suggests that these interactions could be mutually exclusive and may explain why the Bub1-dependent CPC pool exists as a kinetochore-proximal centromere pool that is spatially distinct from the Haspin-dependent inner centromere CPC pool (Broad et al., 2020; Hadders et al., 2020; Liang et al., 2019).*

2. In Figure 1, it is worth testing whether recombinant proteins of Sgo1 (1-415) can bind Survivin and Survivin BIR domain *in vitro*?

We have now tested whether Sgo1 can bind Survivin fl and Survivin₁₋₁₁₆ (composed of mainly the BIR domain) *in vitro* using SEC and ITC (new Fig. S2A/B and new Fig. S1G and S1I, respectively). Our new data shows that both Survivin fl and Survivin₁₋₁₁₆, can interact with Sgo1 in SEC and their binding affinities for Sgo1 as measured by ITC are comparable (240 ± 46.9 nM for Survivin fl vs 408 ± 110 nM for Survivin₁₋₁₁₆). We do note that binding affinity for Survivin fl is slightly stronger than that of Survivin₁₋₁₁₆, possibly due to the additional interaction involving residues downstream of Sgo1 N-terminal tail and Survivin C-helix. This data indicates that the interaction between Sgo1 and Survivin is mainly mediated via the BIR domain.

The contribution of Survivin BIR domain for Sgo1 binding is further strengthened by the observation that the CPC_{ISB10-280} containing Survivin mutant harbouring mutations in the BIR domain (3A) (new Fig. 3J and new Fig. S2C) failed to interact with either Sgo1₁₋₁₅₅ or Sgo1₁₋₄₁₅.

3. In Figure 2, it is worth testing whether Sgo1 (1-130)-LacI-GFP can recruit

exogenously expressed Survivin and Survivin BIR domain using the LacO-LacI tethering assay.

We have now performed the LacO-LacI tethering assay to test whether Sgo1₁₋₁₃₀-LacI-GFP can recruit Survivin-mCherry (new Fig. S3A). Our data shows that indeed, Sgo1₁₋₁₃₀ can recruit Survivin to the LacO array. Importantly, mutations in the BIR domain, known to abrogate BIR domain function reduced Survivin recruitment highlighting the importance of the BIR domain for the interaction with Sgo1₁₋₁₃₀-LacI-GFP. We could not perform the same experiment with Survivin₁₋₁₁₆ because this construct contains a small part of the C-terminal alpha helix (98-116) that could still bind to Borealin (Bourhis et al., 2007 – crystal structure of Survivin 1-120 and Borealin 20-78). Retention of aa residues 98-116 is crucial as removing these will expose a large hydrophobic surface of the BIR domain that might lead to aggregation or/and non-specific interactions. However, as discussed in our response to major point 2 of this reviewer, we did perform these experiments *in vitro*, and show that Survivin₁₋₁₁₆ can bind to Sgo1 as efficiently as the Survivin fl (Fig. S2A and S2B; Fig. S1G and S1I).

4. In Figure 3, it is worth examining whether the K62A or H80A mutant of recombinant Sgo1 (1-155) can bind recombinant Survivin or Survivin BIR domain *in vitro*? According to data shown in TABLE 1 of the Niedzialkowska E et al. study (MBoC, 2012), mutation of H80 to alanine abolished the detectable binding of Survivin to unphosphorylated H3 peptides as measured by ITC. The crystal structure of Survivin bound to the Sgo1 peptide showed that Sgo1-Glu3 fits into the S3 pocket, with the carboxylate group interacting with K62 of Survivin (Jeyaprakash A et al., Structure, 2011). Niedzialkowska E et al. (MBoC, 2012) was unable to detect any binding of the Survivin-K62A mutant to H3T3ph-modified peptides using the structural and biochemical analysis.

We thank the reviewer for suggesting these experiments. We have now assessed the binding of Survivin K62A and H80A to Sgo1₁₋₁₅₅ and Sgo1₁₋₄₁₅ using ITC and SEC (new Fig. 3G-J and new Fig. S2C). Both experiments show that CPC containing Survivin K62A or H80A, can still form a complex with Sgo1 with a similar binding affinity as compared with CPC containing Survivin WT (42.4 ± 10.2 nM for K62A and 112 ± 16.4 nM for H80A compared to 71.2 ± 23.8 for WT). In contrast, CPC containing a triple mutant of Survivin (K62A, E65A, H80A) failed to bind Sgo1 in SEC and ITC. Furthermore, while overexpression of Survivin-mCherry harbouring the single K62A and H80A mutations displayed slightly reduced recruitment to Sgo1₁₋₁₃₀-LacI-GFP foci, recruitment of Survivin E65A was strongly reduced (Fig. S3A). Taken together, these data show 1) the importance of the BIR domain for Sgo1 binding and 2) Survivin K62A and Survivin H80A, the mutants that failed to interact with H3T3ph (Niedzialkowska et al., 2012), retains Sgo1 binding and explain why the CPC still localises to the kinetochore proximal centromeres in the Survivin K62A and Survivin H80A cell lines from the Wang lab (Liang et al., 2019, see also point 5). It seems the latter two mutations simply have

a smaller if not no effect on Sgo1 binding compared to H3T3ph. This is not unreasonable as Survivin-Sgo1 and Survivin-H3T3ph interactions are subtly different due to sequence differences between Sgo1 and H3, while Sgo1 also harnesses several additional regions in the Sgo1-CPC interaction.

5. Based on data shown Figures 3E, 3F, S2A and S2B, the authors claim that the Survivin-Sgo1 interaction is essential for the CPC binding to Sgo1 and that the Sgo1 N-terminal tail acts as a hot-spot, whose perturbation abolishes the ability of CPC to form a complex with Sgo1. Besides, based on data shown in Figures 4A-4C and S2C and S2D, the authors suggest that a weak micromolar affinity long-range electrostatic interaction between Survivin and the Sgo1 N-terminal tail is required to establish a high-affinity CPC-Sgo1 interaction mediated by multiple inter-protein contacts and hydrophobic effects. In the Liang C et al. study (J Cell Biol, 2020), a pool of Aurora B at the kinetochore-proximal centromere region was observed in mitotic HeLa cells in which Haspin-mediated H3-T3ph was chemically inhibited a small-molecule inhibitor (Figures 3B and 3C). Similar results were observed in mitotic HeLa cells in which endogenous Survivin was stably replaced with the H3-binding deficient mutant of Survivin-K62A or Survivin-H80A (Figure 3A). Moreover, when Bub1-mediated H2A_{pT120} was abolished by treatment with a chemical inhibitor BAY 1816032, Aurora B was no longer enriched at centromeres in Survivin-K62A or H80A mutant cells and Haspin inhibitor-treated HeLa cells (Figures 3A-3D). Thus, disrupting the H3-T3ph-Survivin interaction reveals a pool of Aurora B that is present at the KT-proximal centromere, presumably through interaction between H2A-T120ph-bound Sgo1 and the regulatory subunits of the CPC. Based on the literature (Tsukahara T et al., Nature 2010; Bonner M et al., MBoC, 2020), the H2A_{pT120}-dependent localization of Aurora B at the KT-proximal centromere is presumably due to binding of Sgo1 to Borealin. If the K62A or H80A mutant of Survivin is incapable of binding Sgo1 in vitro, then the binding of Sgo1 N-terminal tail to Survivin BIR domain is not necessary for the localization of the CPC at the KT-proximal centromeres in cells.

We thank the reviewer for raising this important point. As discussed in our response to the major point 4 of this reviewer, our new ITC and SEC data shown in figures, Fig. 3G-J and new Fig. S2C, show that CPC containing Survivin K62A or H80A mutations, can still form a stable complex with Sgo1 with binding affinities similar to that of CPC containing Survivin WT (42.4 ± 10.2 nM for K62A and 112 ± 16.4 nM for H80A compared to 71.2 ± 23.8 for WT). This was also confirmed in cells, as the K62A and H80A Survivin mutants were both recruited to LacI-Sgo1₁₋₁₃₀ (Fig. S3A). However, recruitment of Survivin E65A was strongly reduced, in line with the ITC and SEC data for Survivin-3A, again highlighting the importance of the BIR domain in the CPC-Sgo1 interaction. These results provide a plausible explanation for why Liang et al 2019 observed kinetochore-proximal centromere localisation of Aurora B in mitotic HeLa cells where endogenous Survivin was stably replaced with either Survivin-K62A or Survivin-H80A.

6. In Figure 4, it is worth comparing the affinity for the binding of Survivin (wild-type or BIR domain alone) to Sgo1 and Sgo1-A1M (i.e., Nmut).

We have now compared the affinity for the binding of Survivin fl and Survivin₁₋₁₁₆ with Sgo1₁₋₁₅₅ or Sgo1₁₋₁₅₅ Nmut (new Fig. S1G-J) by ITC. Our data shows that both Survivin constructs can bind Sgo1₁₋₁₅₅ with similar affinity (240 ± 46.9 nM for Survivin fl vs 408 ± 110 nM for Survivin₁₋₁₁₆). Moreover, we observed that none of the Survivin constructs could interact with Sgo1₁₋₁₅₅ Nmut, indicating that the interaction between the BIR domain of Survivin and Sgo1 is mediated via the N-terminal tail of Sgo1.

7. In Figures 5A, S3C and S3D, endogenous Sgo1 was depleted by siRNA in HeLa Kyoto cells expressing either wild type Sgo1 (Sgo1-GFP) or mutant Sgo1 (Sgo1Nmut-GFP, Sgo14A-GFP or Sgo1Nmut/4A double mutant) and centromeric levels of Borealin were analyzed. It is unclear whether these exogenous Sgo1-GFP proteins were stably expressed or transiently expressed, and whether endogenous Sgo1 was depleted to similar extent in cells expressing these Sgo1-GFP proteins.

Exogenous Sgo1-GFP constructs were transiently expressed. We apologise that this information was not clearly discussed in the original version. We have now explicitly mentioned this in the Materials and Methods section and in the results section where we discuss the rescue experiments and fluorescence microscopy (page 12, page 13, and page 14; Figure legend for Fig 5 and S5 and materials and methods).

The assessment of endogenous Sgo1 depletion levels (siRNA Sgo1) could not be tested in the Sgo1-GFP overexpression conditions due to technical difficulties with the western blots: The Sgo1 antibody we used to detect endogenous Sgo1 levels by western blot is not very sensitive and the endogenous Sgo1 protein is running too close to the over-expressed Sgo1-GFP (possibly due to the differences in pI between the endogenous Sgo1 and the Sgo1-GFP). The effects we observe in centromeric CPC levels, chromosome alignment and chromosome segregation defects (Figure 5) suggest that the endogenous Sgo1 depletion levels in these experiments were comparable to what we observe in new Figure S4F.

8. In Figure S3E, siRNA-mediated knockdown of Sgo1 in HeLa Kyoto cells strongly increased the percentage of anaphase cells with lagging chromosomes. In general, Sgo1 knockdown results in premature loss of sister chromatid cohesion and spindle checkpoint activation, leading to mitotic arrest and subsequent cell death or mitotic slippage. I wonder how easy it was for the authors to find anaphase cells in which Sgo1 was depleted.

We thank the reviewer for raising this concern. According to our immunoblotting based estimation, we achieved ~ 70 % reduction in Sgo1 levels. This allowed us to evaluate chromosome segregation defects in the population of cells that has partial Sgo1 depletion and that progresses into anaphase. To clarify this we have now added

a paragraph to the main text (page 13, line 11) as follows: *In our siRNA depletion experiments, we observed ~ 70 % reduction in Sgo1 levels (Fig. S4F) as estimated from immunoblotting experiments. This allowed a population of cells to progress into anaphase and let us evaluate the consequence of specifically disrupting CPC-Sgo1 interaction on chromosome segregation in these cells.*

9. In Figure 5B, the authors showed that Sgo1-depleted HeLa cells expressing either Sgo1-GFP or mutant Sgo1 (Sgo1Nmut-GFP, Sgo14A-GFP or Sgo1Nmut/4-GFP) were released from a monastrol-induced mitotic arrest into medium with MG132 and chromosome alignment was assessed. Expression of Sgo1 mutants led to around 70% of cells with severe chromosome misalignment, comparable to the phenotype observed for Sgo1 depletion (Figure 5B). According to the Materials and Methods section, for the Monastrol assay, HeLa Kyoto cells were synchronised with 100 μ M Monastrol for 16 h and released into 5 μ M MG132 for 1 h. In general, after monastrol washout, HeLa cells took around 2 h to complete chromosome alignment. I therefore would recommend the authors to repeat this experiment and allow 2 h for cells to maximally complete chromosome alignment, then determine where cells expressing the Survivin mutants are intrinsically defective, or are only delayed, in chromosome alignment.

We thank the reviewer for their suggestion. We have now assessed chromosome alignment by releasing monastrol-arrested cells into medium with 5 μ M MG132 for 2h (new Fig. S5B). We observe that the mutants still do not rescue the Sgo1 depletion, which indicates that the Sgo1 mutants not only cause a delay, but result in chromosome alignment defects.

We have changed the text in page 13 line 24: *Sgo1-depleted HeLa cells transiently expressing either Sgo1-GFP or mutant Sgo1 (Sgo1^{Nmut}-GFP, Sgo1^{4A}-GFP or Sgo1^{Nmut/4}-GFP) were released from a monastrol-induced mitotic arrest into a medium with MG132 for 1h (Fig. 5B) and 2 h (Fig. S5B) and chromosome alignment was assessed. Under these conditions, expression of Sgo1 mutant constructs led to around 70 % of the cells showing severe chromosome misalignment, comparable to the phenotype observed for Sgo1 depletion (Fig. 5B and S5B).*

10. Based on data shown in Figure 5B, the authors conclude that the Survivin interaction with the Sgo1 N-terminal tail is essential for proper chromosome segregation. It is therefore worth examining chromosome alignment in unperturbed (i.e., without monastrol treatment and release) mitotic cells expressing these Survivin proteins (WT and mutants).

We have now assessed chromosome alignment in unperturbed asynchronous cells (new Fig. S5C) and have added a sentence to the main text as follows:

Finally, we also examined chromosome alignment in asynchronously growing cells transiently expressing the Sgo1 mutants (Fig. S5C) and observed similar chromosome alignment defects as after monastrol release and MG132 treatment. (page 14, line 9)

11. In Figure 5C, I would suggest to include IF pictures showing the localization of Borealin in control HeLa cells.

We thank the reviewer for their suggestion. We have now performed the chromosome spread and linescan experiment with control HeLa cells to show the localisation of Borealin in unperturbed cells. Our data indicates that Borealin is mainly localised at the inner centromere with a pool localised at the kinetochore proximal centromere (new Fig. 5C). We have now revised a related sentence in the main text as follows (page 14, line 14): *Control HeLa cells or Sgo1 depletion in Sgo1-GFP expressing HeLa cells, displayed Borealin localisation at the inner centromere with a small pool localised at the kinetochore proximal centromere (Fig. 5C), consistent with the previously described pattern of CPC localisation in unperturbed mitotic cells (Bekier et al., 2015; Hadders et al., 2020; Liang et al., 2019).*

Minor points:

1. In Figure 5A legend, what does "Three independent experiments, n {greater than or equal to} 50" mean?

We apologise that this was not clear. We have now added more information in our statement as follows: Three independent experiments, $n \geq 50$ cells analysed in total per treatment, mean \pm SD, Mann–Whitney test; ****, $P \leq 0.0001$.

May 19, 2022

RE: JCB Manuscript #202108156R

Dr. A. Arockia Jeyaprakash
University of Edinburgh
Wellcome Trust Centre for Cell Biology
Max Born Crescent
Edinburgh EH9 3BF
United Kingdom

Dear Dr. Jeyaprakash:

Thank you for submitting your revised manuscript entitled "Molecular Basis for CPC-Sgo1 Interaction: Implications for Centromere Localisation and Function of the CPC". The reviewers have now assessed your revised manuscript and are satisfied with revisions. Thus we would be happy to publish your paper in JCB pending final text changes to address the remaining issue of reviewer #2 and pending revisions necessary to meet our formatting guidelines (see details below).

To avoid unnecessary delays in the acceptance and publication of your paper, please read the following information carefully. Please go through all the formatting points paying special attention to those marked with asterisks.

A. MANUSCRIPT ORGANIZATION AND FORMATTING:

1) Text limits: Character count for Articles and Tools is < 40,000, not including spaces. Count includes title page, abstract, introduction, results, discussion, and acknowledgments. Count does not include materials and methods, figure legends, references, tables, or supplemental legends.

2) Figures limits: Articles and Tools may have up to 10 main text figures. Please note that main text figures should be provided as individual, editable files.

3) Figure formatting:

Molecular weight or nucleic acid size markers must be included on all gel electrophoresis.

***** Scale bars must be present on all microscopy images, including inset magnifications. Please include scale bars in main Figs 2C-D (inset magnifications), 4D-E (inset magnifications) and supplemental Figs 3A-C (inset magnifications), 5A.**

Also, please avoid pairing red and green for images and graphs to ensure legibility for color-blind readers. If red and green are paired for images, please ensure that the particular red and green hues used in micrographs are distinctive with any of the colorblind types. If not, please modify colors accordingly or provide separate images of the individual channels.

4) Statistical analysis:

Error bars on graphic representations of numerical data must be clearly described in the figure legend.

***** The number of independent data points (n) represented in a graph must be indicated in the legend. Please, include N in main Figs 2C-D, 3D-E and supplemental Fig 3A-C, 5A and indicate whether N refers to technical or biological replicates (i.e. number of independent experiments, number of cells, samples or animals analyzed). Please, indicate where N in main Fig 5B and supplemental Fig 5C-D refers to the number of cells/metaphases analyzed.**

***** Statistical methods should be explained in full in the materials and methods in a separate section.**

For figures presenting pooled data the statistical measure should be defined in the figure legends.

***** Please also be sure to indicate the statistical tests used in each of your experiments (both in the figure legend itself and in a**

separate methods section) as well as the parameters of the test (for example, if you ran a t-test, please indicate if it was one- or two-sided, etc.). Please indicate in figure legends in supplemental Fig. 3A the statistical test used in your experiment, and in supplemental Fig. 5D the parameters of the t-test used (i.e. one- or two-sided).

*** As you used parametric tests in your study (i.e. t-tests), you should have first determined whether the data was normally distributed before selecting that test. In the stats section of the methods, please indicate how you tested for normality. If you did not test for normality, you must state something to the effect that "Data distribution was assumed to be normal but this was not formally tested."

5) Abstract and title:

The abstract should be no longer than 160 words and should communicate the significance of the paper for a general audience.

*** The title should be less than 100 characters including spaces. Make the title concise but accessible to a general readership. We feel that your current title sounds a bit like a front matter piece, so we would suggest something along the following lines "Mechanistic basis of an Sgo1-survivin interaction contributing to CPC centromere localization and function".

6) Materials and methods: Should be comprehensive and not simply reference a previous publication for details on how an experiment was performed. Please provide full descriptions (at least in brief) in the text for readers who may not have access to referenced manuscripts. The text should not refer to methods "...as previously described." Also, the materials and methods should be included with the main manuscript text and not in the supplementary materials.

7) Please be sure to provide the sequences for all of your primers/oligos and RNAi constructs in the materials and methods.

*** You must also indicate in the methods the source, species, and catalog numbers (where appropriate) for all of your antibodies. Please include species for all of your antibodies.

8) Microscope image acquisition:

The following information must be provided about the acquisition and processing of images:

- a. Make and model of microscope
- b. Type, magnification, and numerical aperture of the objective lenses
- c. Temperature
- *** d. imaging medium
- e. Fluorochromes
- f. Camera make and model
- g. Acquisition software
- h. Any software used for image processing subsequent to data acquisition. Please include details and types of operations involved (e.g., type of deconvolution, 3D reconstitutions, surface or volume rendering, gamma adjustments, etc.).

10) Supplemental materials:

There are strict limits on the allowable amount of supplemental data. Articles/Tools may have up to 5 supplemental figures. There is no limit for supplemental tables.

Please note that supplemental figures and tables should be provided as individual, editable files.

A summary of all supplemental material should appear at the end of the Materials and Methods section (please see any recent JCB paper for an example of this summary).

11) eTOC summary:

A ~40-50 word summary that describes the context and significance of the findings for a general readership should be included on the title page. The statement should be written in the present tense and refer to the work in the third person. It should begin with "First author name(s) et al..." to match our preferred style.

12) Conflict of interest statement:

JCB requires inclusion of a statement in the acknowledgements regarding competing financial interests. If no competing financial interests exist, please include the following statement: "The authors declare no competing financial interests."

13) A separate author contribution section is required following the Acknowledgments in all research manuscripts.

All authors should be mentioned and designated by their first and middle initials and full surnames and the CRediT nomenclature should be used (<https://casrai.org/credit/>).

14) ORCID IDs: ORCID IDs are unique identifiers allowing researchers to create a record of their various scholarly contributions in a single place. At resubmission of your final files, please consider providing an ORCID ID for as many contributing authors as possible.

15) Materials and data sharing: As a condition of publication, authors must make protocols and unique materials (including, but not limited to, cloned DNAs; antibodies; bacterial, animal, or plant cells; and viruses) described in our published articles freely available upon request by researchers, who may use them in their own laboratory only. All materials must be made available on request and without undue delay.

All datasets included in the manuscript must be available from the date of online publication, and the source code for all custom computational methods, apart from commercial software programs, must be made available either in a publicly available database or as supplemental materials hosted on the journal website. Numerous resources exist for data storage and sharing (see Data Deposition: <https://rupress.org/jcb/pages/data-deposition>), and you should choose the most appropriate venue based on your data type and/or community standard. If no appropriate specific database exists, please deposit your data to an appropriate publicly available database.

16) Please note that JCB now requires authors to submit Source Data used to generate figures containing gels and Western blots with all revised manuscripts. This Source Data consists of fully uncropped and unprocessed images for each gel/blot displayed in the main and supplemental figures. The Source Data files will be directly linked to specific figures in the published article.

If your paper includes cropped gel and/or blot images, please be sure to provide one Source Data file for each figure that contains gels and/or blots along with your revised manuscript files. File names for Source Data figures should be alphanumeric without any spaces or special characters (i.e., SourceDataF#, where F# refers to the associated main figure number or SourceDataFS# for those associated with Supplementary figures). The lanes of the gels/blots should be labeled as they are in the associated figure, the place where cropping was applied should be marked (with a box), and molecular weight/size standards should be labeled wherever possible.

B. FINAL FILES:

****The license to publish form must be signed before your manuscript can be sent to production. A link to the electronic license to publish form will be sent to the corresponding author only. Please take a moment to check your funder requirements before choosing the appropriate license.****

Thank you for your attention to these final processing requirements. Please revise and format the manuscript and upload materials within 7 days. Please let us know if any complication preventing you from meeting this deadline arises and we can work with you to determine a suitable revision period.

Please contact the journal office with any questions, cellbio@rockefeller.edu.

Thank you for this interesting contribution, we look forward to publishing your paper in Journal of Cell Biology.

Sincerely,

Arshad Desai
Monitoring Editor
Journal of Cell Biology

Lucia Morgado-Palacin, PhD
Scientific Editor
Journal of Cell Biology

Reviewer #1 (Comments to the Authors (Required)):

In the revised manuscript, the authors have addressed all of my major concerns by providing additional experimental data. I am happy to support its publication in the Journal of Cell Biology.

Reviewer #2 (Comments to the Authors (Required)):

Jeyaprakash and his colleague have extensively revised this MS. They performed a lot of ITC experiments, which make this MS clearer. Then, this MS would be acceptable. However, it is still unclear about contribution of the 188-191 region of Sgo1 for interaction with CPC. Based on ITC results and LacO-LacI assays, authors concluded that the middle region of Sgo1 does not strongly contribute to CPC binding in vitro and posttranscriptional modification might be involved in the Sgo1-CPC interaction in vivo. While posttranscriptional modification of Sgo1 might be possible, authors clearly demonstrated that Sgo130-280 bound to CPC and Sgo130-280_4A did not (Figure S4). These data indicate that the Sgo130-280 region does bind CPC in vitro and 188-191 residues are involved in this interaction. It seems odd to explain posttranscriptional modification for these results. I trust biochemical data. Conformational changes of Sgo1 might happen, and then middle region does not appear to contribute to CPC binding. In any case, it would be better to slightly modify this point.

Reviewer #3 (Comments to the Authors (Required)):

The authors dissected the molecular basis for the interaction between Sgo1 and the CPC1, two critical regulators of mitotic chromosome segregation. Biochemical data in this study, which provide a rationale for the kinetochore-proximal centromere pool of the CPC, are strongly, whereas the intracellular evidence supporting the functional importance of this pool of the CPC for chromosome segregation is relatively weak. Nevertheless, the authors have addressed most of my concerns. I therefore suggest for publication in JCB.

Dear Arshad and Lucia,

27 May 2022

We have now revised the manuscript and figures as outlined below to meet the formatting guidelines of the Journal.

A. MANUSCRIPT ORGANIZATION AND FORMATTING:

Full guidelines are available on our Instructions for Authors page, <https://jcb.rupress.org/submission-guidelines#revised>. **Submission of a paper that does not conform to JCB guidelines will delay the acceptance of your manuscript.**

1) Text limits: Character count for Articles and Tools is < 40,000, not including spaces. Count includes title page, abstract, introduction, results, discussion, and acknowledgments. Count does not include materials and methods, figure legends, references, tables, or supplemental legends.

The character count has been considered and we are below 40,000 characters without spaces.

2) Figures limits: Articles and Tools may have up to 10 main text figures. Please note that main text figures should be provided as individual, editable files.

The figure limit has been considered and we have 5 main figures. The main text figures have been provided as editable .ai files.

3) Figure formatting:

Molecular weight or nucleic acid size markers must be included on all gel electrophoresis.

All figures have been checked to ensure all molecular weight markers are included in the SDS-PAGEs.

*** Scale bars must be present on all microscopy images, including inset magnifications. Please include scale bars in main Figs 2C-D (inset magnifications), 4D-E (inset magnifications) and supplemental Figs 3A-C (inset magnifications), 5A.

All inset magnifications have now been included.

Also, please avoid pairing red and green for images and graphs to ensure legibility for color-blind readers. If red and green are paired for images, please ensure that the particular red and green hues used in micrographs are distinctive with any of the colorblind types. If not, please modify colors accordingly or provide separate images of the individual channels.

We had already provided separate images of the individual channels in all our panels, ensuring legibility for color-blind readers.

4) Statistical analysis:

Error bars on graphic representations of numerical data must be clearly described in the figure legend.

*** The number of independent data points (n) represented in a graph must be

indicated in the legend. Please, include N in main Figs 2C-D, 4D-E and supplemental Fig 3A-C, 5A and indicate whether N refers to technical or biological replicates (i.e. number of independent experiments, number of cells, samples or animals analyzed). Please, indicate where N in main Fig 5B and supplemental Fig 5C-D refers to the number of cells/metaphases analyzed.

These changes have been implemented.

*** Statistical methods should be explained in full in the materials and methods in a separate section.

A Statistical methods section has been now included in the materials and methods section explaining all statistical analyses.

For figures presenting pooled data the statistical measure should be defined in the figure legends.

*** Please also be sure to indicate the statistical tests used in each of your experiments (both in the figure legend itself and in a separate methods section) as well as the parameters of the test (for example, if you ran a t-test, please indicate if it was one- or two-sided, etc.). Please indicate in figure legends in supplemental Fig. 3A the statistical test used in your experiment, and in supplemental Fig. 5D the parameters of the t-test used (i.e. one- or two-sided).

All statistical tests have now been indicated in the figure legends.

*** As you used parametric tests in your study (i.e. t-tests), you should have first determined whether the data was normally distributed before selecting that test. In the stats section of the methods, please indicate how you tested for normality. If you did not test for normality, you must state something to the effect that "Data distribution was assumed to be normal but this was not formally tested."

A sentence has been included in the Statistical methods section of the Materials and Methods indicating how we tested for normality: 'When parametric tests were used, normality was tested using a Shapiro-Wilk normality test using Prism 7.0.' (p. 36)

5) Abstract and title:

The abstract should be no longer than 160 words and should communicate the significance of the paper for a general audience.

The abstract is 159 words.

*** The title should be less than 100 characters including spaces. Make the title concise but accessible to a general readership. We feel that your current title sounds a bit like a front matter piece, so we would suggest something along the following lines "Mechanistic basis of an Sgo1-survivin interaction contributing to CPC centromere localization and function".

We have now changed the title to: Mechanistic Basis for Sgo1-Mediated Centromere Localisation and Function of the CPC

6) Materials and methods: Should be comprehensive and not simply reference a previous publication for details on how an experiment was performed. Please provide full descriptions (at least in brief) in the text for readers who may not have access to referenced manuscripts. The text should not refer to methods "...as

previously described." Also, the materials and methods should be included with the main manuscript text and not in the supplementary materials.

7) Please be sure to provide the sequences for all of your primers/oligos and RNAi constructs in the materials and methods.

*** You must also indicate in the methods the source, species, and catalog numbers (where appropriate) for all of your antibodies. Please include species for all of your antibodies.

We have now included species for all our antibodies in the corresponding Materials and Methods section.

8) Microscope image acquisition:

The following information must be provided about the acquisition and processing of images:

a. Make and model of microscope

b. Type, magnification, and numerical aperture of the objective lenses

c. Temperature

*** d. imaging medium The imaging medium has now been included in the corresponding Materials and Methods section (p.35).

e. Fluorochromes

f. Camera make and model

g. Acquisition software

h. Any software used for image processing subsequent to data acquisition. Please include details and types of operations involved (e.g., type of deconvolution, 3D reconstitutions, surface or volume rendering, gamma adjustments, etc.).

10) Supplemental materials:

There are strict limits on the allowable amount of supplemental data. Articles/Tools may have up to 5 supplemental figures. There is no limit for supplemental tables.

We already had 5 supplemental figures in our previous version of the manuscript.

Please note that supplemental figures and tables should be provided as individual, editable files.

A summary of all supplemental material should appear at the end of the Materials and Methods section (please see any recent JCB paper for an example of this summary).

We already had written the Online Supplemental Material section at the end of the Materials and Methods in our previous version of the manuscript.

11) eTOC summary:

A ~40-50 word summary that describes the context and significance of the findings

for a general readership should be included on the title page. The statement should be written in the present tense and refer to the work in the third person. It should begin with "First author name(s) et al..." to match our preferred style.

We had already included the Summary section in our previous version of the manuscript (45 words), which is retained in the revised version.

12) Conflict of interest statement:

JCB requires inclusion of a statement in the acknowledgements regarding competing financial interests. If no competing financial interests exist, please include the following statement: "The authors declare no competing financial interests."

We had already included the conflict of interest statement in our previous version of the manuscript, which is retained in the revised version.

13) A separate author contribution section is required following the Acknowledgments in all research manuscripts.

This section has now been updated including the new author.

All authors should be mentioned and designated by their first and middle initials and full surnames and the CRediT nomenclature should be used (<https://casrai.org/credit/>).

14) ORCID IDs: ORCID IDs are unique identifiers allowing researchers to create a record of their various scholarly contributions in a single place. At resubmission of your final files, please consider providing an ORCID ID for as many contributing authors as possible.

15) Materials and data sharing: As a condition of publication, authors must make protocols and unique materials (including, but not limited to, cloned DNAs; antibodies; bacterial, animal, or plant cells; and viruses) described in our published articles freely available upon request by researchers, who may use them in their own laboratory only. All materials must be made available on request and without undue delay.

All datasets included in the manuscript must be available from the date of online publication, and the source code for all custom computational methods, apart from commercial software programs, must be made available either in a publicly available database or as supplemental materials hosted on the journal website. Numerous resources exist for data storage and sharing (see Data Deposition: <https://rupress.org/jcb/pages/data-deposition>), and you should choose the most appropriate venue based on your data type and/or community standard. If no appropriate specific database exists, please deposit your data to an appropriate publicly available database.

16) Please note that JCB now requires authors to submit Source Data used to generate figures containing gels and Western blots with all revised manuscripts. This Source Data consists of fully uncropped and unprocessed images for each gel/blot

displayed in the main and supplemental figures. The Source Data files will be directly linked to specific figures in the published article.

We had already submitted the Source data files in the previous version of the manuscript.

As discussed in the cover letter, we have also made textual changes to address Reviewer #2's concerns.

Reviewer #2 (Comments to the Authors (Required)):

Jeyaprakash and his colleague have extensively revised this MS. They performed a lot of ITC experiments, which make this MS clearer. Then, this MS would be acceptable. However, it is still unclear about contribution of the 188-191 region of Sgo1 for interaction with CPC. Based on ITC results and LacO-LacI assays, authors concluded that the middle region of Sgo1 does not strongly contribute to CPC binding in vitro and posttranscriptional modification might be involved in the Sgo1-CPC interaction in vivo. While posttranscriptional modification of Sgo1 might be possible, authors clearly demonstrated that Sgo1₁₃₀₋₂₈₀ bound to CPC and Sgo1_{130-280_4A} did not (Figure S4). These data indicate that the Sgo1₁₃₀₋₂₈₀ region does bind CPC in vitro and 188-191 residues are involved in this interaction. It seems odd to explain posttranscriptional modification for these results. I trust biochemical data. Conformational changes of Sgo1 might happen, and then middle region does not appear to contribute to CPC binding. In any case, it would be better to slightly modify this point.

We thank this reviewer for their suggestion. However, there seems to be a slight confusion in interpreting the Sgo1₁₃₀₋₂₈₀-CPC_{ISB10-280} SEC data presented in Fig. S4A and S4B. We would like to reiterate that as observed in Fig S4A and S4B, both Sgo1_{130-280_4A} and Sgo1_{130-280 WT} bind CPC_{ISB10-280} similarly, albeit weakly as suggested by the sub-stoichiometric SDS-PAGE band intensities observed for the CPC subunits and Sgo1 (Fig. S4A and 4B). This observation together with the fact that perturbing the Sgo1 central region (via Sgo1_{130-280_4A}) did not significantly reduce Sgo1's affinity for CPC (as measured by the ITC experiments), suggest that the contribution of the Sgo1 central region for CPC binding is weak *in vitro*. We have now elaborated the corresponding text (page 11 and 12) by explicitly describing the above-mentioned observations: '*Considering the sub-stoichiometric amounts of Sgo1₁₃₀₋₂₈₀ observed to co-elute with CPC_{ISB10-280} in SEC (based on the SDS-PAGE band intensities observed for the corresponding SEC fractions, Fig. S4A and S4B) and that perturbing the central region interaction did not significantly reduce the measured CPC-binding affinity of Sgo1₁₋₄₁₅ by ITC (Fig. S4D and S4E), we conclude that the Sgo1 central region does not make a significant contribution to CPC binding in vitro. However, as the same Sgo1 mutant (Sgo1_{130-280_4A}) is sufficient to perturb CPC-Sgo1 interaction in cells, we propose that Sgo1 central region requires a yet unidentified post-translational modification/s to facilitate its interaction with the CPC, either in the Sgo1 region and/or in the CPC.*'

We really hope this improves the overall clarity of this section and the revised manuscript fully meets the formatting guidelines.

Yours sincerely,